# Provable Adversarial Robustness for Group Equivariant Tasks: Graphs, Point Clouds, Molecules, and More

**Jan Schuchardt, Yan Scholten, Stephan Günnemann**
{j.schuchardt, y.scholten, s.guennemann}@tum.de
Department of Computer Science & Munich Data Science Institute
Technical University of Munich

## Abstract

A machine learning model is traditionally considered robust if its prediction remains (almost) constant under input perturbations with small norm. However, real-world tasks like molecular property prediction or point cloud segmentation have inherent equivariances, such as rotation or permutation equivariance. In such tasks, even perturbations with large norm do not necessarily change an input's semantic content. Furthermore, there are perturbations for which a model's prediction explicitly needs to change. For the first time, we propose a sound notion of adversarial robustness that accounts for task equivariance. We then demonstrate that provable robustness can be achieved by (1) choosing a model that matches the task's equivariances (2) certifying traditional adversarial robustness. Certification methods are, however, unavailable for many models, such as those with continuous equivariances. We close this gap by developing the framework of equivariance-preserving randomized smoothing, which enables architecture-agnostic certification. We additionally derive the first architecture-specific graph edit distance certificates, i.e. sound robustness guarantees for isomorphism equivariant tasks like node classification. Overall, a sound notion of robustness is an important prerequisite for future work at the intersection of robust and geometric machine learning.

## 1 Introduction

Group equivariance and adversarial robustness are two important model properties when applying machine learning to real-world tasks involving images, graphs, point clouds and other data types:

Group equivariance is an ubiquitous form of task symmetry [1]. For instance, we do not know how to optimally classify point clouds, but know that the label is not affected by permutation. We cannot calculate molecular forces in closed form, but know that the force vectors rotate as the molecule rotates. Directly enforcing such equivariances in models is an effective inductive bias, as demonstrated by the success of convolutional layers [2], transformers [3] and graph neural networks [4].

Adversarial robustness [5–7] is a generalized notion of model Lipschitzness: *A small change to a model's input $x$ should only cause a small change to its prediction $f(x)$*. If a model is adversarially robust, then its accuracy will not be negatively affected by sensor noise, measurement errors or other small perturbations that are unavoidable when working with real-world data.

For the first time, we address the following question: *What is adversarial robustness in tasks that are group equivariant?* This is necessary because the notions of input and output similarity used in prior work on adversarial robustness (e.g. $\ell_p$ distances) are not suitable for group equivariant tasks.

37th Conference on Neural Information Processing Systems (NeurIPS 2023).

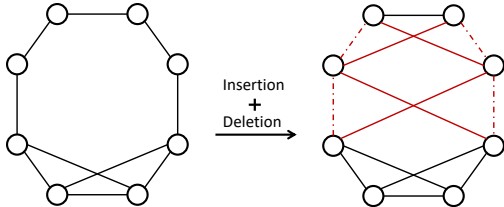 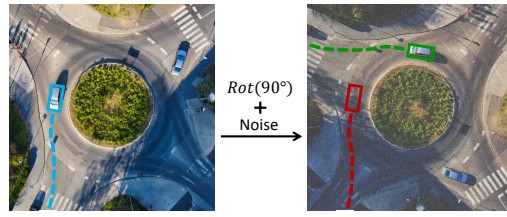

Figure 1: The right graph is constructed by inserting and deleting four edges. While their $\ell_0$ distance is large, the graphs are isomorphic and should thus have the same set of node labels.

Figure 2: The predicted trajectory (blue) should rotate as the image rotates (green), even in the presence of camera noise and other perturbations. It should not remain constant (red).

Fig. 1 illustrates why group equivariant tasks require rethinking input similarity. The right graph is constructed by modifying a large fraction of edges, meaning the perturbation of the adjacency matrix has a large $\ell_0$ norm. Prior work [8–17] may deem it too large for a node classifiers' prediction to be robust. However, we know the graphs are isomorphic, meaning they are the same geometric object and should have the same set of labels. Fig. 2 illustrates why group equivariant tasks also require rethinking output similarity. Prior work considers a prediction robust if it remains (almost) constant. But when predicting trajectories from drone footage, we know that they should rotate as the drone rotates – even in the presence of camera noise. The predictions should explicitly not remain constant.

To address these two issues we propose a sound notion of adversarial robustness for group equivariant tasks that a) measures input similarity by lifting distance functions to group invariant distance functions and b) jointly considers transformations of the input and output space. This notion of adversarial robustness applies to arbitrary tasks, groups and distance functions.

A natural goal after introducing a novel notion of robustness is developing (provably) robust models. We show that robustness can be guaranteed by being robust under the traditional notion of adversarial robustness – if the model's equivariances match the equivariances of the task it is used for. Importantly, this implies that existing robustness guarantees may actually hold for significantly larger sets of perturbed inputs. For instance, proving the robustness of a graph neural network w.r.t. $\ell_0$ distance is in fact proving robustness w.r.t graph edit distance with uniform cost for insertion and deletion.

Although equivariant models make provable robustness more attainable, there are no certification procedures for many architectures. For instance, there is no prior work on proving robustness for rotation equivariant models. To close this gap, we develop the framework of equivariance-preserving randomized smoothing. It specifies sufficient conditions under which models retains their equivariances when undergoing randomized smoothing – a state-of-the-art approach for simultaneously increasing and proving the robustness of arbitrary models [18–20]. In addition to that, we generalize the aforementioned graph edit guarantees to arbitrary, user specified costs. Varying these costs allows for a fine-grained analysis of a model's robustness to graph perturbations.

To summarize, our core contributions are that we

- propose a sound notion of adversarial robustness for group equivariant tasks,
- show that using equivariant models facilitates achieving (provable) robustness,
- develop the framework of equivariance-preserving randomized smoothing,
- and generalize each existing graph and node classification robustness certificate for $\ell_0$ perturbations to graph edit distance perturbations with user-specified costs.

Overall, reconsidering what adversarial robustness means in equivariant tasks is an important prerequisite for future work at the intersection of robust and geometric machine learning.

## 2 Related work

There is little prior work that studies equivariance and adversarial robustness jointly. The work that exists [21–31] only considers group invariant classification (a special case of equivariance) and does not consider how a task's equivariances should influence what we consider robust.

**Model invariance and robustness.** Prior work mostly focuses on a trade-off between invariance and robustness in image classification, i.e., whether increasing robustness to rotation or translation

decreases robustness to $\ell_p$ perturbations and vice-versa [21–25]. Schuchardt and Günnemann [26] used knowledge about the invariances of point cloud classifiers to prove that they are constant within larger regions than could be shown using previous approaches.

**Group invariant distances.** Recently, stability results for graph classifiers under isomorphism invariant optimal transport distances have been derived [27–29]. For point cloud classifiers, using the permutation invariant Chamfer or Hausdorff distance to craft attacks has been proposed [30, 31]. These works only focus on invariance and specific domains and do not consider that distances should be task-dependent: A rotation invariant distance for images may be desirable when segmenting cell nuclei, but not when classifying hand-written digits, since it would fail to distinguish 6 and 9.

**String edit distance.** In concurrent work, Huang et al. [32] use randomized smoothing to prove robustness of classifiers w.r.t. string edit distance, i.e., the number of substitutions that are needed to convert one string from alphabet $\Sigma \cup \{\perp\}$ into another, up to insertion of alignment tokens $\perp$. Their work further emphasizes the need for invariant distance functions in domains with symmetries, and the usefulness of randomized smoothing for proving robustness w.r.t. such distances.

**Robustness of models with equivariances.** Aside from work that studies invariance and adversarial robustness jointly, there is a rich literature investigating the robustness of models that happen to have equivariances. This includes convolutions [5–7, 33], transformers[34–37], point cloud models [30, 31, 38–46] and graph neural networks [8–17, 47–55]. The models are however treated as a series of matrix multiplications and nonlinearities, without accounting for their equivariances or the equivariances of the tasks they are used for. Nevertheless, many methods can actually be reused for proving (non-)robustness under our proposed notion of adversarial robustness (see Section 5).

**Transformation-specific robustness.** A subfield of robust machine learning focuses on robustness to unnoticeable parametric transformations (e.g. small rotations) [42, 46, 56–65]. These works implicitly assume that large transformations lead to easily identifiable out-of-distribution samples. This is not the case with equivariant tasks: For instance, a molecule rotated by $180°$ is still the same geometric object. Furthermore, they do not consider unstructured perturbations. Nevertheless, transformation-specific robustness can be framed as a special case of our proposed notion (see Appendix J).

**Semantics-aware robustness.** Our work is closely related to different proposals to include ground truth labels in the definition of adversarial robustness [22, 52, 66–71]. A problem is that the ground truth is usually unknown, which limits experimental evaluation to simple data generating distributions [52, 70] or using human study participants [22, 67, 71]. Geisler et al. [72] overcome this problem in the context of neural combinatorial optimization by using adversarial perturbations that are known to change the ground truth of a decision problem. Group equivariant tasks admit a similar approach, since we know how the ground truth changes for specific input transformations.

## 3 Background

**Group theory and equivariance.** Discussing group equivariance requires a few algebraic concepts. A group is a set $\mathbb{G}$ with identity element $e$ and associative operator $\cdot : \mathbb{G} \times \mathbb{G} \rightarrow \mathbb{G}$ such that $\forall g \in \mathbb{G} : e \cdot g = g \cdot e = g$ and each $g \in \mathbb{G}$ has an inverse element $g^{-1}$ with $g^{-1} \cdot g = g \cdot g^{-1} = e$. We are interested in transformations with a group structure, such as rotations. Given set $\mathbb{X}$, a (left) group action is a function $\bullet_{\mathbb{X}} : \mathbb{G} \times \mathbb{X} \rightarrow \mathbb{X}$ that transforms elements of $\mathbb{X}$ and preserves the structure of $\mathbb{G}$, i.e. $g \bullet_{\mathbb{X}} h \bullet_{\mathbb{X}} x = (g \cdot h) \bullet_{\mathbb{X}} x$. For instance, rotation by $\phi°$ and then by $\psi°$ is the same as rotation by $(\psi + \phi)°$. A group may act differently on different sets. For example, rotation group $SO(3)$ may act on point clouds via matrix multipliciation while acting on class labels via the identity function. When clear from context, we drop the subscripts. A function $f : \mathbb{X} \rightarrow \mathbb{Y}$ is equivariant if each action on its input is equivalent to an action on its output, i.e. $\forall x \in \mathbb{X}, g \in \mathbb{G} : f(g \bullet_{\mathbb{X}} x) = g \bullet_{\mathbb{Y}} f(x)$.

**Adversarial robustness** means that any small change to clean input $x$ only causes a small change to prediction $f(x)$, i.e. $\max_{x' \in \mathbb{B}_x} d_{\text{out}}(f(x), f(x')) \leq \delta$ with $\mathbb{B}_x = \{x' \mid d_{\text{in}}(x, x') \leq \epsilon\}$, and $d_{\text{in}}$ and $d_{\text{out}}$ quantifying input and output distance. We refer to the set of admissible perturbed inputs $\mathbb{B}_x$ as the *perturbation model*. A common special case is $d_{\text{out}}(y, y') = \mathbb{1}[y \neq y']$ and $\delta = 0$ [5–7]. Other forms of robustness involve training data [73, 74], but we focus on test-time perturbations.

**Randomized smoothing** [18–20] is a paradigm for increasing and proving the robustness of models in an architecture-agnostic manner. It works by randomizing the inputs of a *base model* $h$ to construct a more robust *smoothed model* $f$. While originally proposed for provably robust image classification,

it has evolved into a much more general framework that can be applied to various domains and tasks [14, 20, 51, 75–81]. Consider a measurable input space $(\mathbb{X}, \mathbb{D})$ and a measurable base model $h : \mathbb{X} \to \mathbb{V}$ that maps to a measurable intermediate space $(\mathbb{V}, \mathbb{F})$ (e.g. logits). Given the base model $h$ and a family of probability measures $(\mu_x)_{x \in \mathbb{X}}$ on $(\mathbb{X}, \mathbb{D})$ indexed by $\mathbb{X}$ (e.g. Gaussians with mean $x$), we can define the input-dependent pushforward measure $\mu_x \circ h^{-1}$ for any input $x \in \mathbb{X}$.[1] We can further define a *smoothing scheme* $\xi : \Delta(\mathbb{V}, \mathbb{F}) \to \mathbb{Y}$ that maps from probability measures $\Delta(\mathbb{V}, \mathbb{F})$ on the intermediate space to an output space $\mathbb{Y}$ (e.g. logit distributions to labels).[2] This lets us construct the smoothed model $f(x) = \xi(\mu_x \circ h^{-1})$, which makes a prediction for input $x \in \mathbb{X}$ based on some quantity of the input-dependent pushforward measure $\mu_x \circ h^{-1}$ (e.g. the expected value). Intuitively, if two inputs $x$ and $x'$ are sufficiently similar, then the smoothing measures $\mu_x$ and $\mu'_x$ will have a large overlap, thus leading to similar smoothed predictions. The combination of measures $(\mu_x)_{x \in \mathbb{X}}$ and smoothing scheme $\xi$ determines for which $d_{\text{in}}$ and $d_{\text{out}}$ robustness is guaranteed. We discuss these combinations in more detail in Appendix D.

## 4 Redefining robustness for group equivariant tasks

For the first time, we seek to define adversarial robustness for group equivariant tasks. By task we mean an unknown function $y : \mathbb{X} \to \mathbb{Y}$ that is equivariant with respect to the action of a group $\mathbb{G}$. We assume that it is approximated by a (learned) model $f : \mathbb{X} \to \mathbb{Y}$ whose robustness we want to determine. The model does not have to be equivariant. Like traditional adversarial robustness, we assume there are functions $d_{\text{in}} : \mathbb{X} \times \mathbb{X} \to \mathbb{R}_+$ and $d_{\text{out}} : \mathbb{Y} \times \mathbb{Y} \to \mathbb{R}_+$, which define what constitutes a small change in domain $\mathbb{X}$ and co-domain $\mathbb{Y}$, if we are not concerned with group symmetries. For instance, $\ell_2$ distance is a natural notion of similarity between Euclidean coordinates.

For ease of exposition, we further assume that all considered optimization domains are compact (so that minima and maxima exist) and that group $\mathbb{G}$ acts isometrically on $\mathbb{X}$, i.e. $\forall x, x' \in \mathbb{X}, \forall g \in \mathbb{G} :$ $d_{\text{in}}(g \bullet x, g \bullet x') = d_{\text{in}}(x, x')$. This covers most practically relevant use cases. For completeness, we discuss non-compact sets and non-isometric actions in Appendices H and I.

### 4.1 Perturbation model for group equivariant tasks

Our first goal is to resolve the shortcomings of using typical input distance functions $d_{\text{in}}$ to define what constitutes small input perturbations in group equivariant tasks. We seek some improved function $\hat{d}_{\text{in}}$ that accounts for the equivariance of $y$ and simultaneously captures the similarity of objects in domain $\mathbb{X}$, as defined by original distance function $d_{\text{in}}$. To this end, we define three desiderata.

**Desideratum 1.** We know that any perturbed $x' \in \mathbb{X}$ and $g \bullet x'$ with $g \in \mathbb{G}$ are up to symmetry the same geometric object with the same semantic content, i.e. $y(g \bullet x') = g \bullet y(x')$. But as illustrated in Fig. 1, group actions may cause a drastic change w.r.t. distance $d_{\text{in}}$. Thus, even if $f(x)$ is "robust" within $\mathbb{B}_x = \{x' \mid d_{\text{in}}(x, x') \leq \epsilon\}$ for some $\epsilon$, there may still be a $x' \in \mathbb{B}_x$ and $g \bullet x' \notin \mathbb{B}_x$ that lead to two completely different predictions. If a prediction can be altered without changing the semantic content of an input, it can hardly be considered robust. This problem cannot be resolved by requiring robustness for very large $\epsilon$ so that $\mathbb{B}_x$ covers all symmetric objects. Doing so would also include objects with substantially different semantic content from clean input $x$, which should actually lead to different predictions. Instead, we need a $\hat{d}_{\text{in}}$ that is constant for all symmetric forms of $x'$, so that robustness to one implies robustness to all: $\forall x, x' \in \mathbb{X}, g \in \mathbb{G} : \hat{d}_{\text{in}}(x, g \bullet x') = \hat{d}_{\text{in}}(x, x')$.

**Desideratum 2.** While the first desideratum accounts for equivariance, a model should also be robust to sensor noise, measurement errors and other small perturbations that are not necessarily group actions. Therefore, elements that are close with respect to the original distance function $d_{\text{in}}$ should remain close under our new distance function $\hat{d}_{\text{in}}$, i.e. $\forall x, x' : \hat{d}_{\text{in}}(x, x') \leq d_{\text{in}}(x, x')$.

**Desideratum 3.** The first two desiderata could be fulfilled by $\hat{d}_{\text{in}}(x, x') = 0$ or a function that arbitrarily changes the ordering of inputs w.r.t. distance. To appropriately capture similarity, $\hat{d}_{\text{in}}$ should not only underapproximate $d_{\text{in}}$, but preserve it as best as possible. Let $\mathbb{D}$ be the set of functions from $\mathbb{X} \times \mathbb{X}$ to $\mathbb{R}_+$ that fulfill the first two desiderata. We require that $\hat{d}_{\text{in}}(x, x') = \max_{\gamma \in \mathbb{D}} \gamma(x, x')$.

---

[1] Note that $h^{-1}$ is not the inverse but the pre-image. The base model need not be invertible.

[2] $\mathbb{V}$ and $\mathbb{Y}$ can also be the same set, as is the case with majority voting for classifiers.

**Proposition 1.** *A function $\hat{d}_{\text{in}} : \mathbb{X} \times \mathbb{X} \to \mathbb{R}_+$ that fulfills all three desiderata for any original distance function $d_{\text{in}} : \mathbb{X} \times \mathbb{X} \to \mathbb{R}_+$ exists and is uniquely defined: $\hat{d}_{\text{in}}(x, x') = \min_{g \in \mathbb{G}} d_{\text{in}}(x, g \bullet x')$.*

We prove this result in Appendix C. We refer to $\hat{d}_{\text{in}}$ as the *action-induced distance*. It is the distance after optimally aligning the perturbed input $x'$ with the clean input $x$ via a group action.

**Example: Point cloud registration distance.** Let $\mathbb{X} = \mathbb{R}^{N \times D}$. Consider $\mathbb{G} = S_N \times SE(D)$, the set of permutation matrices, rotation matrices and translation vectors. Let $\mathbb{G}$ act via $(\boldsymbol{P}, \boldsymbol{R}, \boldsymbol{t}) \bullet \boldsymbol{X} = \boldsymbol{P}\left(\boldsymbol{X}\boldsymbol{R}^T + \boldsymbol{1}_N \boldsymbol{t}^T\right)$ and let $d_{\text{in}}$ be the Frobenius distance. Then $\hat{d}_{\text{in}}$ is the *registration distance* [82], i.e. the distance after finding an optimal correspondence and applying an optimal rigid transformation.

**Example: Graph edit distance.** Let $\mathbb{X}$ be the set of all adjacency matrices $\{0, 1\}^{N \times N}$ with $d_{\text{in}}(\boldsymbol{A}, \boldsymbol{A}') = ||\boldsymbol{A} - \boldsymbol{A}'||_0$. Consider $\mathbb{G} = S_N$, the set of permutation matrices. Let $\mathbb{G}$ act via $\boldsymbol{P} \bullet \boldsymbol{A} = \boldsymbol{P}\boldsymbol{A}\boldsymbol{P}^T$. Then $\hat{d}_{\text{in}}(\boldsymbol{A}, \boldsymbol{A}')$ is the *graph edit distance* with uniform cost [83–85]. That is, the number edges that have to be inserted or deleted to transform $\boldsymbol{A}$ into a graph isomorphic to $\boldsymbol{A}'$.

As demonstrated by these examples, we do not claim to have invented this notion of distance for equivariant domains. Our contribution is justifying, in a principled manner, why this notion of distance should be used to define adversarial robustness for group equivariant tasks.

**Perturbation model.** Now that we have an appropriate input distance for group equivariant tasks, we can use it to define the set of admissible perturbed inputs $\mathbb{B}_x$ as $\{x' \mid \min_{g \in \mathbb{G}} d_{\text{in}}(x, g \bullet x') \leq \epsilon\}$ with some small $\epsilon \in \mathbb{R}_+$. Because every $g \in \mathbb{G}$ has an inverse element $g^{-1}$, this set is identical to $\{g \bullet x' \mid g \in \mathbb{G}, d_{\text{in}}(x, x') \leq \epsilon\}$. We use this equivalent representation because it lets us disentangle group actions from other perturbations. That is, $f(x)$ should be robust to inputs that can be generated via a small perturbation w.r.t. the original distance $d_{\text{in}}$ followed by a group action.

## 4.2 Output distance for group equivariant tasks

Fig. 2 illustrates that we also need to reconsider what a small change to a model's prediction is. Letting a group element act on a prediction may cause a large change w.r.t. $d_{\text{out}}$. In particular, we may have $d_{\text{out}}(y(x), y(g \bullet x)) = d_{\text{out}}(y(x), g \bullet y(x)) \gg 0$, even though $x$ and $g \bullet x$ have a distance of zero w.r.t. the action-induced distance. Thus, we would consider even the ground truth $y$ itself to be non-robust. Using action-induced distances, i.e. $\min_{g \in \mathbb{G}} d_{\text{out}}(y, g \bullet y')$, is not appropriate either. Action-induced distances cannot distinguish between a model that transforms its predictions in compliance with the ground truth and one that applies arbitrary group actions.

To appropriately measure output distance, we need to account for differences between clean prediction $f(x)$ and perturbed prediction $f(g \bullet x')$ that are caused by the specific group element $g \in \mathbb{G}$ acting on the model's input. To do so, we need to first revert the effect of the group action before comparing the predictions. That is, we need to measure output distance using $d_{\text{out}}(f(x), g^{-1} \bullet f(g \bullet x'))$.

## 4.3 Proposed definition of robustness for group equivariant tasks

Combining the perturbation model from Section 4.1 with the output distance from Section 4.2 leads to the following definition of adversarial robustness for group equivariant tasks:

**Definition 1.** *Assume that ground truth function $y : \mathbb{X} \to \mathbb{Y}$ is equivariant with respect to the action of group $\mathbb{G}$. Then, a prediction $f(x)$ for clean input $x \in \mathbb{X}$ is $(\mathbb{G}, d_{\text{in}}, d_{\text{out}}, \epsilon, \delta)$-equivariant-robust if*

$$\left(\max_{x' \in \mathbb{X}} \max_{g \in \mathbb{G}} d_{\text{out}}(f(x), g^{-1} \bullet f(g \bullet x'))\right) \text{ s.t. } d_{\text{in}}(x, x') \leq \epsilon) \leq \delta. \tag{1}$$

Simply speaking, prediction $f(x)$ should be considered robust if it is robust to unnoticeable perturbations and is approximately equivariant around $x$. For perturbations of size $\epsilon$ w.r.t. the original $d_{\text{in}}$, the prediction should not change by more than $\delta$ and no group action should cause this error to increase beyond $\delta$. Note that Definition 1 depends on the equivariances of the task $y$, not those of the model $f$. Further note that the constraint involves original distance $d_{\text{in}}$, not action-induced distance $\hat{d}_{\text{in}}$.

**Special cases.** If the task is not equivariant, i.e. $\mathbb{G} = \{e\}$, we recover the traditional notion of adversarial robustness because $e$ acts via identity. For $\epsilon = 0$, constant $\delta$ is a bound on the equivariance error, a common evaluation metric in geometric machine learning (see, e.g. [86–90]).

**Other invariant distances.** As discussed in Section 2, alternative invariant distances have been proposed for certain group invariant classification tasks. This includes Hausdorff and Chamfer

distance [30, 31], as well as optimal transport for graphs [27–29]. Any group invariant function $d_{\text{in}}$ is preserved by the action-induced distance: $\min_{g \in \mathbb{G}} d_{\text{in}}(x, g \bullet x') = \min_{g \in \mathbb{G}} d_{\text{in}}(x, x') = d_{\text{in}}(x, x')$. Thus, all previous results for invariant classification are compatible with our notion of robustness.

**Local budgets and local robustness.** For data that is composed of $N$ distinct elements, such as point clouds, one may want local budgets $\epsilon_1, \ldots, \epsilon_N$. For tasks that involve $M$ distinct predictions, such as segmentation, one may only be interested in the robustness of some subset of predictions. Definition 1 can be extended to accommodate local budgets and robustness, see Appendix G.

## 5 Achieving provable robustness for group equivariant tasks

Now we have a sound notion of robustness that overcomes the limitations discussed in Section 1. But it is not clear how to achieve provable robustness. Given a task $y : \mathbb{X} \to \mathbb{Y}$ that is equivariant with respect to the action of a group $\mathbb{G}$, we want a model $f : \mathbb{X} \to \mathbb{Y}$ and a corresponding algorithm that can verify the $(\mathbb{G}, d_{\text{in}}, d_{\text{out}}, \epsilon, \delta)$-equivariant-robustness of a prediction $f(x)$.

A challenge is that the action-induced distance $\hat{d}_{\text{in}}$, which defines our perturbation model and thus underlies the optimization domain in Eq. (1), is generally not tractable. For example, deciding whether the graph edit distance is smaller than some $\epsilon \in \mathbb{R}$ is NP-hard [91]. A solution would be to relax the optimization domain in order to pessimistically bound the model's actual robustness. This is, in essence, the approach taken by works on robustness to graph optimal transport perturbation models (e.g. [27, 28]). Instead of optimizing over discrete correspondences between objects, they optimize over couplings that define soft correspondences. There is however a more straight-forward solution that lets us take advantage of years of research in geometric and robust machine learning: Applying the principles of geometric machine learning and using a model with matching equivariances.

**Proposition 2.** *Consider a model $f : \mathbb{X} \to \mathbb{Y}$ that is equivariant with respect to the action of a group $\mathbb{G}$. Any prediction $f(x)$ is $(\mathbb{G}, d_{\text{in}}, d_{\text{out}}, \epsilon, \delta)$-equivariant-robust if and only if it is $(\{e\}, d_{\text{in}}, d_{\text{out}}, \epsilon, \delta)$-equivariant-robust, i.e. fulfills traditional adversarial robustness.*

*Proof.* Because $f$ is equivariant, we have $\forall g : g^{-1} \bullet f(g \bullet x') = g^{-1} \bullet g \bullet f(x') = f(x') = e^{-1} \bullet f(e \bullet x')$ and thus $d_{\text{out}}(f(x), g^{-1} \bullet f(g \bullet x')) = d_{\text{out}}(f(x), e^{-1} \bullet f(e \bullet x'))$. $\square$

By using model $f$ with the same equivariances as ground truth $y$, we reduce the problem of proving robustness to that of proving traditional adversarial robustness i.e. $\mathbb{G} = \{e\}$. Again, note that robustness does not mean that a prediction remains constant, but that it transforms in compliance with semantics-preserving transformations of the input, even under small perturbations (see Eq. (1)).

**Discussion.** Proposition 2 provides another strong argument for the use of geometric machine learning models. These models facilitate the problem of achieving provable robustness for group equivariant tasks — without relaxing the action-induced distance and thus weakening our guarantees. We can instead build upon existing robustness certification procedures for equivariant models, such as transformers, PointNet and graph convolutional networks, under traditional perturbation models, like $\ell_2$ or $\ell_0$ perturbations [8–17, 34–46]. We just need to use our knowledge about the equivariances of tasks and models to reinterpret what is actually certified by these procedures.

**Relation to orbit-based certificates.** A related result was discussed for group *invariant* point cloud classifiers in [26]: If $\mathbb{X} = \mathbb{R}^{N \times D}$ and classifier $f$ is constant within Frobenius norm ball $\mathbb{B}$ and invariant w.r.t. the action of group $\mathbb{G}$, then it is also constant within $\{g \bullet X' \mid X' \in \mathbb{B}, g \in \mathbb{G}\}$. However, this work did not discuss whether being constant within this set is desirable, how it relates to task invariance and what this result tells us about the adversarial robustness of the classifier.

**Adversarial attacks.** While we focus on provable robustness, Proposition 2 also applies to adversarial attacks, i.e. proving non-robustness via counterexample. Even when the equivariances of $y$ and $f$ do not match, traditional attacks are feasible solutions to Eq. (1) since they amount to constraining $g$ to set $\{e\}$. For completeness, we perform experiments with adversarial attacks in Appendix A.4.

### 5.1 Equivariance-preserving randomized smoothing

Equivariant models reduce the problem of proving robustness for group equivariant tasks to that of proving traditional robustness. However, specialized procedures to make these proofs are only available for a limited range of architectures. For example, specialized certification procedures for

point cloud models are limited to the PointNet architecture [42, 92], and there are no procedures to prove the robustness of models with continuous equivariances, such as rotation equivariance.

For such models, one could try to apply randomized smoothing (recall Section 3). By choosing a suitable smoothing scheme $\xi$ and measures $(\mu_x)_{x \in \mathbb{X}}$ for distances $d_{\text{in}}$ and $d_{\text{out}}$, one can transform any base model $h$ into smoothed model $f$ and prove its $(\{e\}, d_{\text{in}}, d_{\text{out}}, \epsilon, \delta)$-equivariant-robustness. However, base model $h$ having the same equivariances as task $y$ does not guarantee that smoothed model $f$ has the same equivariances. Thus, one can generally not use Proposition 2 to prove $(\mathbb{G}, d_{\text{in}}, d_{\text{out}}, \epsilon, \delta)$-equivariant-robustness. We propose the following sufficient condition for verifying that a specific smoothing scheme and family of measures are equivariance-preserving (proof in Appendix E.1).

**Proposition 3.** *Assume two measurable spaces $(\mathbb{X}, \mathbb{D})$, $(\mathbb{V}, \mathbb{F})$, an output space $\mathbb{Y}$ and a measurable base model $h : \mathbb{X} \to \mathbb{V}$ that is equivariant with respect to the action of group $\mathbb{G}$. Further assume that $\mathbb{G}$ acts on $\mathbb{X}$ and $\mathbb{V}$ via measurable functions. Let $\xi : \Delta(\mathbb{V}, \mathbb{F}) \to \mathbb{Y}$ be a smoothing scheme that maps from the set of probability measures $\Delta(\mathbb{V}, \mathbb{F})$ on intermediate space $(\mathbb{V}, \mathbb{F})$ to the output space. Define $T_{\mathbb{X}, g}(\cdot)$ to be the group action on set $\mathbb{X}$ for a fixed $g$, i.e. $T_{\mathbb{X}, g}(x) = g \bullet_{\mathbb{X}} x$. Then, the smoothed model $f(x) = \xi(\mu_x \circ h^{-1})$ is equivariant with respect to the action of group $\mathbb{G}$ if both*

- *the family of measures $(\mu_x)_{x \in \mathbb{X}}$ is equivariant, i.e. $\forall x \in \mathbb{X}, g \in \mathbb{G} : \mu_{g \bullet x} = \mu_x \circ T_{\mathbb{X}, g}^{-1}$,*

- *and smoothing scheme $\xi$ is equivariant, i.e. $\forall \nu \in \Delta(\mathbb{V}, \mathbb{F}), g \in \mathbb{G} : \xi(\nu \circ T_{\mathbb{V}, g}^{-1}) = g \bullet \xi(\nu)$.*

Note that $\circ$ is a composition of functions, whereas $\bullet$ is a group action. The intuition behind Proposition 3 is that, if family of measures $(\mu_x)_{x \in \mathbb{X}}$, and base model $h$, and smoothing scheme $\xi$ are equivariant, then we have a chain of equivariant functions which is overall equivariant. We provide a visual example of such an equivariant chain of functions in Fig. 31.

In the following, we show that various schemes and measures preserve practically relevant equivariances and can thus be used in conjunction with Proposition 2 to prove $(\mathbb{G}, d_{\text{in}}, d_{\text{out}}, \epsilon, \delta)$-equivariant-robustness. For the sake of readability, we provide a high-level discussion, leaving the formal propositions in Appendix E.2. We summarize our results in Tables 1 and 2. In Appendix D, we discuss how to derive robustness guarantees for arbitrary combinations of these schemes and measures.

**Componentwise smoothing schemes.** The most common type of smoothing scheme can be applied whenever the intermediate and output space have $M$ distinct components, i.e. $\mathbb{V} = \mathbb{A}^M$ and $\mathbb{Y} = \mathbb{B}^M$ for some $\mathbb{A}, \mathbb{B}$. It smooths each of the $M$ base model outputs independently. This includes majority voting [20], expected value smoothing [75, 76] and median smoothing [77], which can be used for tasks like classification, segmentation, node classification, regression, uncertainty estimation and object detection [14, 20, 51, 75–81]. Such schemes preserve equivariance to groups acting on $\mathbb{V}$ and $\mathbb{Y}$ via permutation. Note that $\mathbb{G}$ need not be the symmetric group $S_N$ to act via permutation. For instance, the identity function is a permutation, meaning componentwise smoothing schemes can be used to prove robustness for arbitrary group invariant tasks. See Proposition 4.

**Expected value smoothing scheme.** When the intermediate and output space are real-valued, i.e. $\mathbb{V} = \mathbb{Y} = \mathbb{R}^M$, one can make predictions via the expected value [75, 76]. Due to linearity of expectation, this scheme does not only preserve permutation equivariance but also equivariance to affine transformations. However, certifying the smoothed predictions requires that the support of the output distribution is bounded by a hyperrectangle, i.e. $\text{supp}(\mu_x \circ h^{-1}) \subseteq \{\boldsymbol{y} \in \mathbb{R}^M \mid \forall m : a_m \leq y_m \leq b_m\}$ for some $\boldsymbol{a}, \boldsymbol{b} \in \mathbb{R}^M$ [75, 76]. See Proposition 5.

**Median smoothing scheme.** When the support of the model's output distribution is real-valued and potentially unbounded, one can make smoothed predictions via the elementwise median [77]. This scheme does not only preserve permutation equivariance but also equivariance to elementwise linear transformations, such as scaling. See Proposition 6.

**Center smoothing scheme.** Center smoothing [93] is a flexible scheme that can be used whenever $d_{\text{out}}$ fulfills a relaxed triangle inequality. It predicts the center of the smallest $d_{\text{out}}$ ball with measure of at least $\frac{1}{2}$. Center smoothing has been applied to challenging tasks like image reconstruction, dimensionality reduction, object detection and image segmentation [93, 51]. We prove that center smoothing is equivariant to any group acting isometrically w.r.t. $d_{\text{out}}$. For example, when $\mathbb{V} = \mathbb{Y} = \mathbb{R}^{M \times D}$ and $d_{\text{out}}$ is the Frobenius norm, center smoothing guarantees $(\mathbb{G}, d_{\text{in}}, d_{\text{out}}, \epsilon, \delta)$-equivariant-robustness for any group acting on $\mathbb{R}^{N \times D}$ via permutation, rotation, translation or reflection. See Proposition 7.

**Product measures.** Many randomized smoothing methods use product measures. That is, they use independent noise to randomize elements of an $N$-dimensional input space $\mathbb{X} = \mathbb{A}^N$. The most

popular example are exponential family distributions, such as Gaussian, uniform and Laplacian noise, which can be used to prove robustness for various $\ell_p$ distances [19, 20, 94, 95], Mahalanobis distance [60, 96] and Wasserstein distance [97]. Product measures preserve equivariance to groups acting via permutation. Again, group $\mathbb{G}$ need not be symmetric group $S_n$ to act via permutation. For instance, rotating a square image by $90°$ (see Fig. 2) is a permutation of its pixels. See Proposition 8.

**Isotropic Gaussian measures.** Isotropic Gaussian measures are particularly useful for $\mathbb{X} = \mathbb{R}^{N \times D}$. They guarantee robustness when $d_{\text{in}}$ is the Frobenius distance [43, 46] and preserve equivariance to isometries, i.e. permutation, rotation, translation and reflection. Combined with Proposition 2, this guarantees robustness w.r.t. the aforementioned point cloud registration distance. See Proposition 9.

**Transformation-specific measures.** A standard tool for proving transformation-specific robustness (see Section 2) is transformation-specific smoothing [46, 60, 63–65], i.e. applying randomly sampled transformations from a parametric family $(\psi_\theta)_{\theta \in \Theta}$ with $\psi_\theta : \mathbb{X} \to \mathbb{X}$ to the inputs of a model. If all $\psi_\theta$ with $\theta \in \Theta$ are equivariant to the actions of group $\mathbb{G}$, then transformation-specific smoothing preserves this equivariance. For example, random scaling [65] preserves rotation equivariance and additive noise (e.g. Gaussian) preserves translation equivariance for $\mathbb{X} = \mathbb{R}^{N \times D}$. See Proposition 10.

**Sparsity-aware measures.** Sparsity-aware noise [14] can be applied to discrete graph-structured data to guarantee robustness to edge and attribute insertions and deletions. We prove that sparsity-aware measures preserve equivariance to groups acting via graph isomorphisms. As we discuss in the next section, this guarantees robustness w.r.t. the graph edit distance. See Proposition 11.

## 5.2 Deterministic edit distance certificates for graph and node classification

Besides sparsity-aware smoothing, there are also deterministic procedures for proving robustness for specific graph neural networks, namely graph convolutional networks [98] and APPNP [99]. They are however limited to uniform costs. To be more specific, let $\mathbb{X}$ be the set of all graphs $\{0,1\}^{N \times D} \times \{0,1\}^{N \times N}$ with $N$ nodes and $D$ binary attributes and let $(\boldsymbol{A})_+ = \max(\boldsymbol{A}, 0)$ and $(\boldsymbol{A})_- = \min(\boldsymbol{A}, 0)$ with elementwise maximum and minimum. Define $d_{\text{in}}((\boldsymbol{X}, \boldsymbol{A}), (\boldsymbol{X}', \boldsymbol{A}'))$ as

$$c_{\boldsymbol{X}}^+ \cdot ||(\boldsymbol{X}' - \boldsymbol{X})_+||_0 + c_{\boldsymbol{X}}^- \cdot ||(\boldsymbol{X}' - \boldsymbol{X})_-||_0 + c_{\boldsymbol{A}}^+ \cdot ||(\boldsymbol{A}' - \boldsymbol{A})_+||_0 + c_{\boldsymbol{A}}^- \cdot ||(\boldsymbol{A}' - \boldsymbol{A})_-||_0, \quad (2)$$

with costs $c_{\boldsymbol{X}}^+, c_{\boldsymbol{X}}^-, c_{\boldsymbol{A}}^+, c_{\boldsymbol{A}}^-$ for insertion and deletion of attributes and edges. Prior work can only prove robustness for costs in $\{\infty, 1\}$, i.e. disallow certain types of perturbations and use uniform cost for the remaining ones. In Appendix F we generalize each existing deterministic graph and node classification guarantee to non-uniform costs. This includes procedures based on convex outer adversarial polytopes [10], policy iteration [11], interval bound propagation [12], bilinear programming [13], and simultaneous linearization and dualization [15]. Our proofs mostly require solving different knapsack problems with local constraints and only two distinct costs – which can be done efficiently via dynamic programming or linear relaxations with analytic solutions (see Appendix F.1). As per Proposition 2, proving $(\{e\}, d_{\text{in}}, d_{\text{out}}, \epsilon, \delta)$-equivariant-robustness of isomorphism equivariant models in this way proves $(S_N, d_{\text{in}}, d_{\text{out}}, \epsilon, \delta)$-equivariant-robustness. Thus, these procedures guarantee robustness w.r.t graph edit distance $d_{\text{in}}((\boldsymbol{X}, \boldsymbol{A}), (\boldsymbol{X}', \boldsymbol{A}')) = \min_{\boldsymbol{P} \in S_N} d_{\text{in}}((\boldsymbol{X}, \boldsymbol{A}), (\boldsymbol{P}\boldsymbol{X}', \boldsymbol{P}\boldsymbol{A}'\boldsymbol{P}^T))$.

## 5.3 Limitations

Group equivariance covers various common task symmetries. However, there are symmetries that do not fit into this framework, such as local gauge equivariance [100–102] or wave function symmetries [103–112]. A limitation of relying on group equivariant models for provable robustness is that there are tasks where non-equivariant models have better empirical performance, for example vision transformers [113]. Models that are in principle equivariant may also lose their equivariance due to domain-specific artifacts like image interpolation. Finally, it should be noted that providing guarantees of the form $(\mathbb{G}, d_{\text{in}}, d_{\text{out}}, \epsilon, \delta)$ requires a-priori knowledge that the task is equivariant to group $\mathbb{G}$. These limitations are however not relevant for many important domains in which equivariant models are the de-facto standard (e.g. graphs, point clouds and molecules).

# 6 Experimental evaluation

In the following, we demonstrate how sound robustness guarantees for tasks with discrete and continuous domains and equivariances can be obtained via equivariance-preserving randomized smoothing.

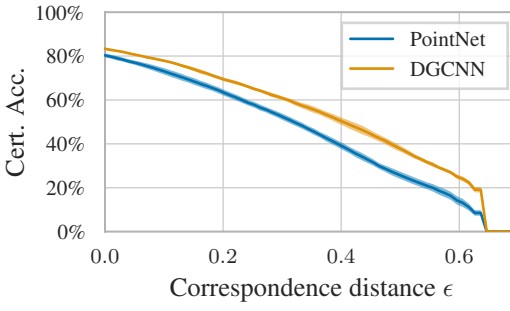
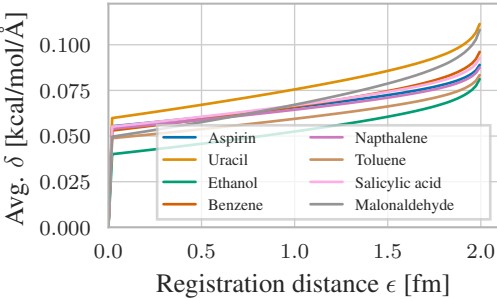

Figure 3: Provable robustness of smoothed ($\sigma = 0.2$) PointNet and DGCNN point cloud classifiers on ModelNet40. Correspondence distance $\epsilon$ is the Frobenius distance between point clouds after finding an optimal matching via permutation.

Figure 4: Provable robustness of smoothed ($\sigma = 1\,\mathrm{fm}$) DimeNet++ force predictions on MD17. The average provable bounds $\delta$ on prediction changes are 2 to 13 times smaller than the average test errors ($0.19$ to $0.74\,\mathrm{kcal/mol/\mathring{A}}$).

We additionally evaluate our graph edit distance certificates and how the cost of edit operations affects provable robustness. All experimental details are specified in Appendix B. Certificates are evaluated on a separate test set. Randomized smoothing methods are evaluated using sampling and hold with high probability. Each experiment is repeated 5 times. We visualize the standard deviation using shaded areas. When the shaded area is too small, we report its maximum value. An implementaton will be made available at https://cs.cit.tum.de/daml/equivariance-robustness.

**Point cloud classification.** The first task we consider is point cloud classification, i.e. $\mathbb{X} = \mathbb{R}^{N \times 3}$ and $\mathbb{Y} = \{1, \dots, K\}$. Natural distances $d_{\mathrm{in}}$ and $d_{\mathrm{out}}$ are the Frobenius distance and 0-1-loss. The task is permutation invariant, i.e. symmetric group $\mathbb{G} = S_N$ acts on $\mathbb{X}$ via permutation and on $\mathbb{Y}$ via identity. Thus, we need to prove robustness w.r.t. to the correspondence distance $\hat{d}_{\mathrm{in}}$, i.e. the Frobenius distance after finding an optimal matching between rows via permutation. To preserve invariance and use Proposition 2 while randomly smoothing, we use Gaussian measures and majority voting, i.e. we predict the most likely label under Gaussian input perturbations. As invariant base models $h$ we choose PointNet [114] and DGCNN [115], which are used in prior work on robust point cloud classification [30, 31]. We evaluate the robustness guarantees for smoothing standard deviation $\sigma = 0.2$ on ModelNet40 [116]. This dataset consists of point cloud representations of CAD models from 40 different classes. Fig. 3 shows the certified accuracy, i.e. the percentage of predictions that are correct and provably robust. Both smoothed models have a high accuracy above 80%, but DGCNN has significantly higher provable robustness. In Appendix A.1 we repeat the experiment for different values of $\sigma$ and compare our guarantees to those of 3DVerifier [92].

**Molecular force prediction.** Next, we consider a task with continuous equivariances: Predicting the forces acting on each atom in a molecule, which can for instance be used to simulate molecular dynamics. We have $\mathbb{X} = \mathbb{R}^{N \times 3}$ (atomic numbers are treated as constant) and $\mathbb{Y} = \mathbb{R}^{N \times 3}$. Suitable distances $d_{\mathrm{in}}$ and $d_{\mathrm{out}}$ are the Frobenius distance and the average $\ell_2$ error $\sum_{n=1}^{N} ||\boldsymbol{Y}_n - \boldsymbol{Y}'_n||_2 \,/\, N$. The task is permutation, rotation and translation equivariant, i.e. $\mathbb{G} = S_N \times SE(3)$. Thus, $\hat{d}_{\mathrm{in}}$ is the point cloud registration distance. To preserve equivariance, we use Gaussian measures and center smoothing. We choose DimeNet++ [117] as equivariant base model $h$. We evaluate the provable robustness for smoothing standard deviation $\sigma = 1\,\mathrm{fm}$ on MD17 [118], a collection of 8 datasets, each consisting of a large number of configurations of a specific molecule. With just 1000 training samples, the models achieve low average test errors between $0.19$ and $0.74\,\mathrm{kcal/mol/\mathring{A}}$. We use 1000 samples per test set to evaluate the robustness guarantees. Fig. 4 shows the average upper bounds $\delta$ on the change of the predicted force vectors for small perturbations between 0 and $2\,\mathrm{fm}$. These average $\delta$ are smaller than the average test errors by factors between 2 and 13. The standard deviation across seeds is below $3 \times 10^{-4}\,\mathrm{kcal/mol/\mathring{A}}$ for all $\epsilon$. In Appendix A.2 we show that the maximum $\epsilon$ and $\delta$ grow approximately linearly with $\sigma$ and repeat our experiments with SchNet [119] and SphereNet [120] base models. Note that we are not (primarily) proving robustness to malicious perturbations, but robustness to measurement errors or errors in previous simulation steps.

**Node classification.** Finally, we consider a task with discrete domain and co-domain: Node classification, i.e. $\mathbb{X} = \{0, 1\}^{N \times D} \times \{0, 1\}^{N \times N}$ and $\mathbb{Y} = \{1, \dots, K\}^N$. Distance $d_{\mathrm{in}}$ is naturally defined via a weighted sum of inserted and deleted bits (see Eq. (2)). Output distance $d_{\mathrm{out}}$ is the

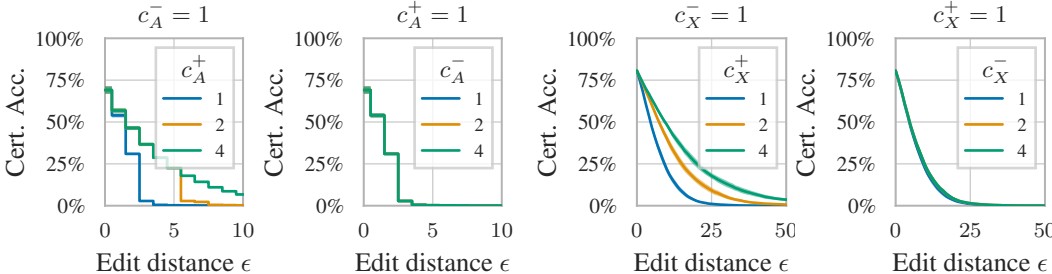

Figure 5: Randomized smoothing guarantees for GCNs on Cora-ML. Increasing the cost of adversarial edge insertions increases the provable robustness for the same perturbation budgets $\epsilon$.

Figure 6: Convex adversarial polytope guarantees for GCNs on Cora-ML. Increasing the cost of attribute insertions increases the provable robustness for the same perturbation budgets $\epsilon$.

$\ell_0$ distance, i.e. the number of changed predictions. The task is equivariant to symmetric group $S_n$ acting on $\mathbb{X}$ and $\mathbb{Y}$ via isomorphisms and permutations, respectively. To preserve equivariance, we use sparsity-aware measures and majority voting. The flip probabilites (see Appendix F.7) are set to $p_X^+ = 0, p_X^- = 0, p_A^+ = 0.001, p_A^- = 0.8$. We use a 2-layer graph convolutional network [98] as our isomorphism equivariant base model $h$. Fig. 5 shows the resulting guarantees on Cora ML [121, 122] for graph edit perturbations of the adjacency, i.e. $c_X^+ = c_X^- = \infty$, for varying costs $c_A^+$ and $c_A^-$. We observe that increasing the cost for edge insertions significantly increases the provable robustness for the same budgets $\epsilon$, whereas the cost for edge deletions has virtually no effect. This suggests that insertions are much more effective at changing a model's predictions, which was empirically observed in prior work [123]. We repeat the experiment with other flip probabilities in Appendix A.3. The standard deviation of certified accuracies across all seeds was below $2.1\,\mathrm{p.p.}$ everywhere. Note that here, certified accuracy does not refer to the percentage of predictions that are correct and constant, but those that are correct and permute in compliance with isomorphisms of the input graph.

**Deterministic edit distance certificates.** In Fig. 6 we repeat the node classification experiment with our generalization of the convex polytope method from [10]. We consider feature perturbations, i.e. $c_A^+ = c_A^- = \infty$, for varying $c_X^+$ and $c_X^-$. Again, the cost of insertions has a larger effect, suggesting that the model is less robust to them. In Appendix A.3 we repeat the experiments with the four other graph edit distance certificates and also evaluate them on Citeseer [124] and the TUDataset [125]. Overall, our generalizations of existing graph robustness guarantees let us prove robustness to more complex threat models, even though evaluating the edit distance is computationally hard.

# 7   Conclusion

The main goal of this paper is to define adversarial robustness for group equivariant tasks. To this end, we introduce action-induced distances as the appropriate notion of input distance and consider how predictions should transform for semantics-preserving transformations of inputs. If the equivariances of a task and model match, the proposed notion of robustness can be guaranteed by proving traditional adversarial robustness. This has two consequences: Firstly, specialized certification procedures for equivariant architectures can be reused. One just has to reinterpret what is actually guaranteed by these procedures, e.g. robustness to perturbations bounded by graph edit distance. Secondly, randomized smoothing can be used to provide architecture-agnostic guarantees — but only if the smoothing scheme and measures preserve the model's equivariances. We experimentally demonstrated the generality of this equivariance-preserving randomized smoothing approach by certifying robustness for graph, point cloud and molecule models. Overall, our work provides a sound foundation for future work at the intersection of robust and geometric machine learning.

**Future work.** Based on Proposition 2, a direction for future work is to continue making equivariant models more robust to classic threat models, without caring about equivariance. A more interesting direction is to make attacks, defenses and certificates equivariance-aware. For instance, knowledge about model equivariances could be used to reduce the search space for attacks, disrupt gradient-based attacks (similar to [126]), or derive stronger randomized smoothing guarantees (as proposed for invariant models in [26]). Finally, developing procedures to certify non-equivariant models (e.g. vision transformers) or models that are only "almost equivariant" (e.g. due to interpolation artifacts) under the proposed notion of robustness would be desirable for equivariant computer vision tasks.

# 8 Acknowledgments and disclosure of funding

The authors would like to thank Hongwei Jin for assistance with the implementation of their topology attack certificates, Tom Wollschläger for providing access to code for training molecular force models, and Aman Saxena for valuable discussions concerning non-compact sets and non-isometric actions. This work has been funded by the Munich Center for Machine Learning, by the DAAD program Konrad Zuse Schools of Excellence in Artificial Intelligence (sponsored by the Federal Ministry of Education and Research), and by the German Research Foundation, grant GU 1409/4-1. The authors of this work take full responsibility for its content.

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

# A   Additional experiments

## A.1   Point cloud classification

**Different smoothing standard deviations.** In Fig. 7 we repeat our experiments on permutation invariant point cloud classification for different smoothing standard deviations $\sigma \in \{0.05, 0.1, 0.15, 0.25\}$. The maximum certifiable correspondence distance $\epsilon$ increases linearly with $\sigma$. For $\sigma = 0.05$, the average natural accuracy (i.e. certified accuracy at $\epsilon = 0$) of the smoothed PointNet and DGCNN models is $88\%$ and $90.4\%$, respectively. Their certified accuracies are almost identical for most $\epsilon$. For the larger $\sigma = 0.25$, their accuracy decreases to $75.5\%$ and $79.9\%$, respectively, i.e. the difference in accuracy grows by $2\,\mathrm{p.p.}$. DGCNN also offers stronger robustness guarantees with certified accuracies that are up to $6.3\,\mathrm{p.p.}$ larger. The most important takeaway should however be that equivariance-preserving randomized smoothing provides sound robustness guarantees for equivariant tasks where discrete groups act on continuous data.

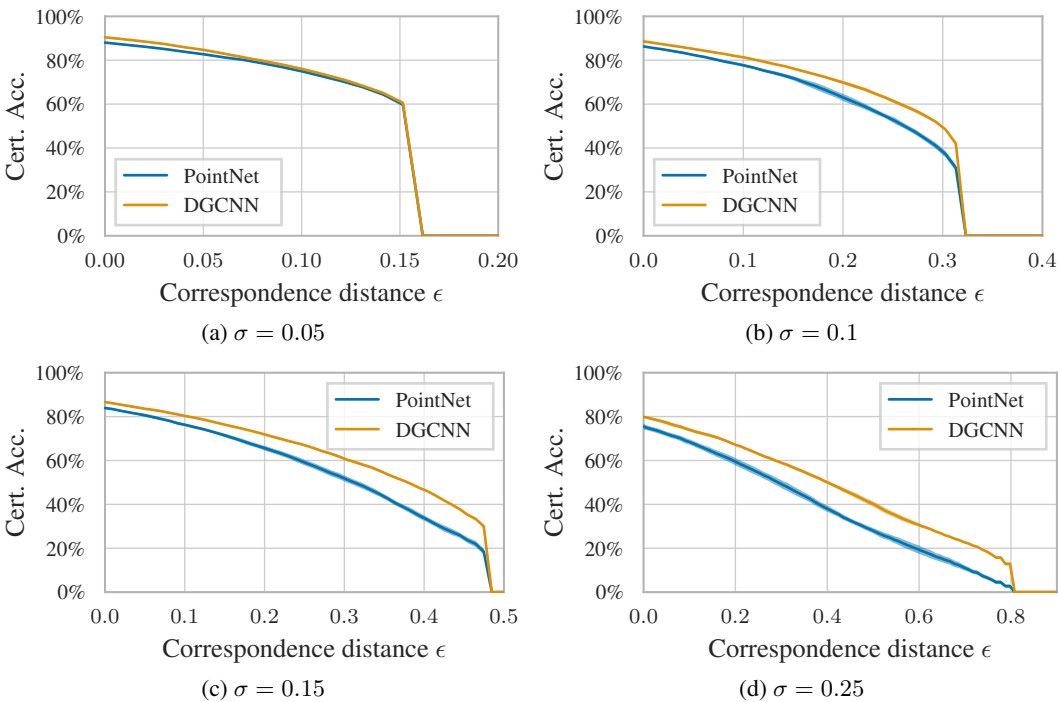

Figure 7: Provable robustness of smoothed PointNet and DGCNN point cloud classifiers on ModelNet40 for different smoothing standard deivations $\sigma$. Correspondence distance $\epsilon$ is the Frobenius distance between point clouds after finding an optimal matching via permutation. For larger $\sigma$, DGCNN offers better accuracy and provable robustness than PointNet.

**Comparison to 3DVerifier.** As shown in Fig. 7, Gaussian randomized smoothing with $\sigma \in [0.05, 0.25]$ lets us retain relatively high accuracy while certifying robustness to correspondence distance perturbations of sizes up to $\epsilon = 0.8$. However, as discussed in Section 5, Proposition 2 does not make any assumptions about how we prove traditional adversarial robustness. For example, 3DVerifier [92], which is a specialized certification procedure for PointNet, can on average prove robustness robustness to $\ell_2$ norm perturbations of size $\epsilon = 0.959$ (see Table 2 in [92]). Because both the point cloud classification task and the PointNet architecture are permutation invariant, this directly translates to correspondence distance perturbations with an average size of $0.959$. This experiment demonstrates that developing specialized certification procedures for a wider range of equivariant models is a promising direction for future work. Nevertheless, equivariance-preserving randomized smoothing remains valuable as a general-purpose certification procedure. For instance, 3DVerifier cannot prove robustness for DGCNN or other point cloud classification architectures.

### A.2  Molecular force prediction

**Different smoothing standard deviations.** In Fig. 8 we repeat our experiments on permutation and isometry equivariant force prediction on MD17 with different smoothing standard deviations $\sigma \in \{10\,\text{fm}, 100\,\text{fm}\}$. The maximum certifiable registration distance grows linearly with $\sigma$. This is inherent to Gaussian smoothing (see [20]). The average certified output distance $\delta$ also grows approximately linearly. This can potentially be explained as follows: In a sufficiently small region around clean input $X$, the model may have approximately linear behavior. If $\sigma$ is small, the output distribution will be similar to a normal distribution whose covariances grow linearly with $\sigma^2$. The certified $\delta$ of center smoothing are always between the median and maximum sampled distance from the smoothed prediction (see [93]), which will in this case also increase approximately linearly.

**Different base models.** In Appendix A.2 we repeat our experiment with $\sigma = 1\,\text{fm}$ and different base models, namely SchNet [119] and SphereNet [120]. SphereNet achieves lower average test errors between $0.45$ and $1.4\,\text{kcal/mol/Å}$ than SchNet, which has test errors between between $0.51$ and $1.89\,\text{kcal/mol/Å}$. However, the provable robustness guarantees obtained for SphereNet are worse than those for SchNet by more than an order of magnitude. For adversarial budgets of up to $2\,\text{fm}$, the average certified output distance $\delta$ for SphereNet varies between $0.35\,\text{kcal/mol/Å}$ (Benzene) and $1.61\,\text{kcal/mol/Å}$ (Malonaldehyde). For comparison, the average certified output $\delta$ for SchNet vary between $0.04\,\text{kcal/mol/Å}$ (Ethanol) and $0.12\,\text{kcal/mol/Å}$ (Uracil) and are quite similar (but not identical to) those for DimeNet++. We also observe a much higher variance across random seeds.

**Conclusion.** While these observations may be of interest to some readers that are specifically interested in machine learning for molecules, the most important takeaway should be that equivariance-preserving randomized smoothing can be used to obtain sound robustness guarantees for equivariant tasks in which continuous groups act on continuous data.

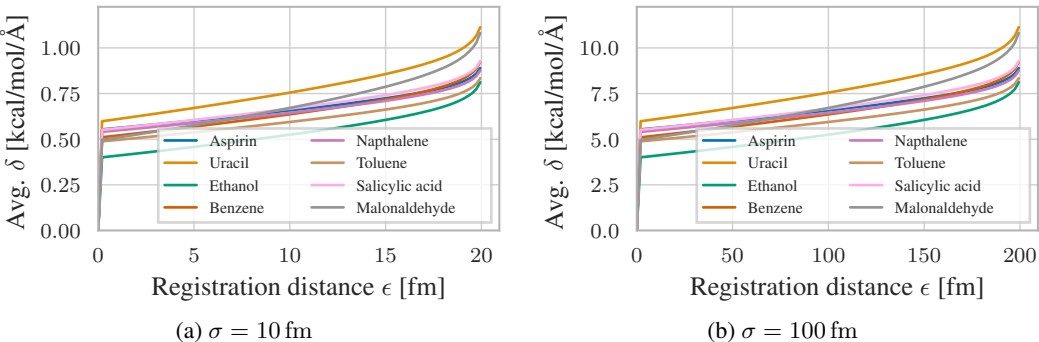

Figure 8: Provable robustness of smoothed DimeNet++ force predictions on MD17 with smoothing standard deviations $\sigma \in \{10\,\text{fm}, 100\,\text{fm}\}$. The average certified output distance $\delta$ and the maximum certifiable registration distance $\epsilon$ grow (approximately) linearly with $\sigma$.

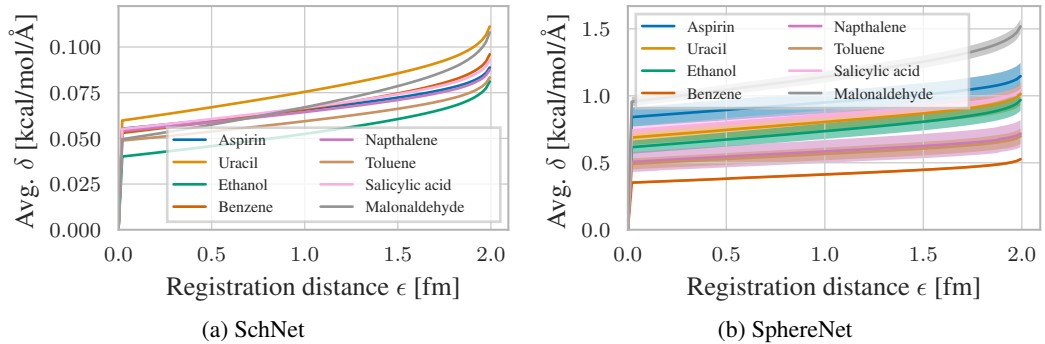

Figure 9: Provable robustness of smoothed SchNet and SphereNet force predictions on MD17 with a smoothing standard deviation $\sigma = 1\,\text{fm}$. While SphereNet is more accurate, its provable robustness guarantees are weaker and have higher variance across different random weight initializations.

### A.3 Node and graph classification

Next, we evaluate the different robustness guarantees for graph and node classification that we generalized to graph edit distance perturbations with user-specified costs (see Appendix F).

**Convex outer adversarial polytopes for attribute perturbations.** In Fig. 11 we repeat our experiment on proving the robustness of 2-layer graph convolutional networks to graph edit distance perturbations. We use our generalization of the convex relaxation approach (see Appendix F.3), which was originally proposed in [10], on the Citeseer [124] node classification dataset. As before, we set $c_A^+ = c_A^- = \infty$ and vary the cost of attribute insertion $c_X^+$ and deletion $c_X^-$. The result is qualitatively similar to our results on Cora-ML. Increasing the cost of attribute insertions leads to significantly higher certified accuracy. With $c_X^- = 1$, the average certified accuracy for $c_X^+ = 4$ at edit distance $\epsilon = 25$ is 23.4%, which is much larger than the 6.6% for $c_X^+ = 1$. Increasing the cost of deletions does however not have a noticeable effect, which matches the empirical observation that insertions are more effective when attacking graph neural networks [123]. The main difference to Cora-ML is that we have a larger standard deviation across seeds. This is not necessarily surprising, since many graph benchmarks are known to be sensitive to the choice of data split and model initialization [127]. Note that the split and initialization for each seed are the same in all experiments.

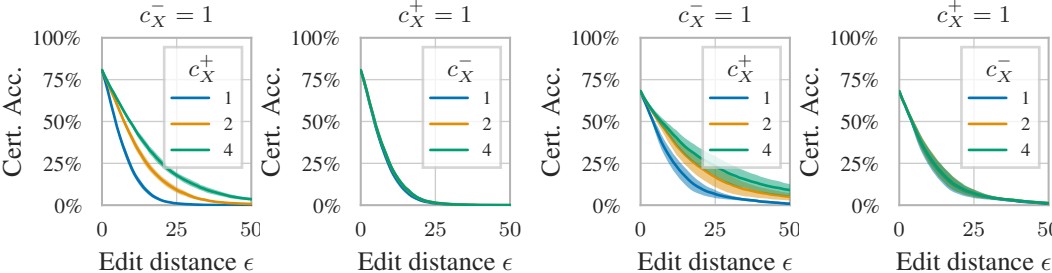

Figure 10: Convex outer adversarial polytope robustness guarantees for GCNs under attribute perturbations on Cora-ML.

Figure 11: Convex outer adversarial polytope robustness guarantees for GCNs under attribute perturbations on Citeseer.

**Interval bound propagation for attribute perturbations.** In Figs. 12 and 13 we repeat the same experiment using our generalization of the interval bound propagation approach (see Appendix F.2) originally proposed in [12]. Unlike before, increasing a single cost parameter while keeping the other one at 1 has no noticeable effect on the certified accuracy. A potential explanation is that interval bound propagation only offers very loose guarantees. For example, the maximum certifiable edit distance on Cora (see Fig. 12) is 10, compared to over 50 with the previously discussed convex relaxation approach. The relaxation might simply be too loose to accurately capture the effect of different perturbation types on the models.

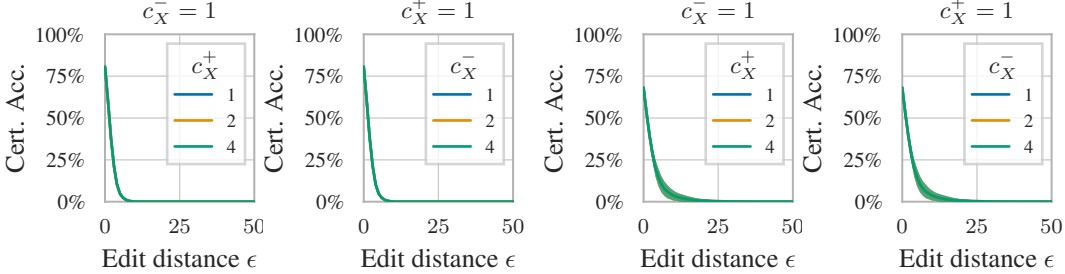

Figure 12: Interval bound propagation robustness guarantees for GCNs under attribute perturbations on Cora-ML.

Figure 13: Interval bound propagation robustness guarantees for GCNs under attribute perturbations on Citeseer.

**Policy iteration for adjacency perturbations.** Next, we consider perturbations of the adjacency matrix. We begin with our generalization of the policy iteration approach (see Appendix F.6)

originally proposed in [11]. This robustness guarantee is specifically designed for $\pi$-PPNP, which is APPNP [99] where the degree normalized adjacency matrix $\boldsymbol{D}^{-1/2}\boldsymbol{A}\boldsymbol{D}^{-1/2}$ is replaced with $\boldsymbol{D}^{-1}\boldsymbol{A}$. Different from the other guarantees, we do not have a single global budget $\epsilon$. Instead, we only have nodewise budgets $\rho_1, \ldots, \rho_N$ with $\rho_n = (D_{n,n} - c + s)$ with an arbitrarily chosen constant $c$ and attack strength $s$. Like in [11], we set $c = 11$ and vary attack strength $s$. Figs. 14 and 15 shows the resulting certified accuracies on Cora-ML and Citeseer for $c_{\boldsymbol{X}}^+ = c_{\boldsymbol{X}}^- = \infty$ and varying $c_{\boldsymbol{A}}^+, c_{\boldsymbol{A}}^-$. Again, increasing the cost of adversarial perturbations to $c_{\boldsymbol{A}}^+ = 4$ has a larger effect, in some cases almost tripling the certified accuracy. But there is also a small increase of up to $2.9\,\mathrm{p.p.}$ certified accuracy when increasing the deletion cost $c_{\boldsymbol{A}}^-$ from 1 to 4 on Citeseer.

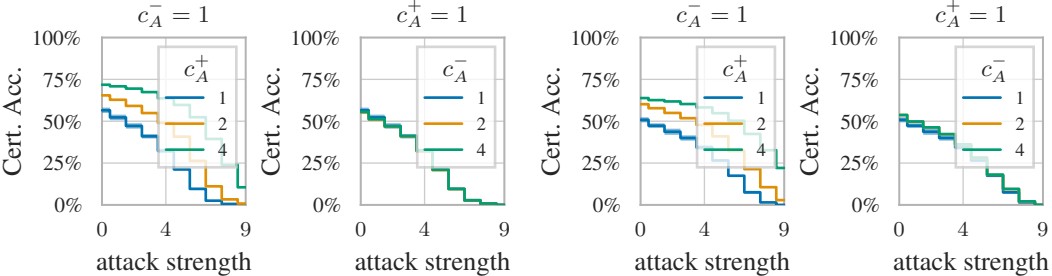

Figure 14: Robustness guarantees for $\pi$-PPNP under adjacency perturbations on Cora-ML. Note that we do not have a global budget $\epsilon$ and instead vary per-node budgets.

Figure 15: Robustness guarantees for $\pi$-PPNP under adjacency perturbations on Citeseer. Note that we do not have a global budget $\epsilon$ and instead vary per-node budgets.

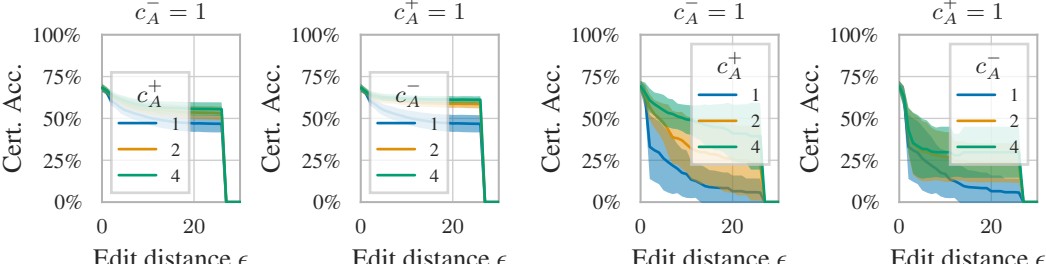

Figure 16: Linearization and dualization robustness guarantees for GCN graph classifiers under adjacency perturbations on PROTEINS.

Figure 17: Linearization and dualization robustness guarantees for GCN graph classifiers under adjacency perturbations on MUTAG.

**Bilinear programming for adjacency perturbations.** Generalizing the bilinear programming method for proving robustness of GCNs to adjacency perturbations proposed in [13] from $c_{\boldsymbol{X}}^+, c_{\boldsymbol{X}}^-, c_{\boldsymbol{A}}^+ = \infty, c_{\boldsymbol{X}}^- = 1$ to $c_{\boldsymbol{X}}^- \in \mathbb{R}_+$ does not require any modifications, since scaling the cost is equivalent to scaling the adversarial budgets (see Appendix F.4). We thus refer to [13] for experimental results.

**Linearization and dualization for adjacency perturbations.** Next, we consider GCN-based node classifiers and adjacency perturbations. We prove robustness via our generalization of the procedure proposed in [15]. This guarantee is specifically designed for single layer GCNs followed by a linear layer and max-pooling. Since Cora-ML and Citeseer are node classification datasets, we instead use the PROTEINS and MUTAG datasets from the TUDataset [125] collection, which were also used in [15]. Figs. 16 and 17 show the resulting certified accuracies for fixed $c_{\boldsymbol{X}}^+ = c_{\boldsymbol{X}}^- = \infty$ and varying adjacency perturbation costs $c_{\boldsymbol{A}}^+$ and $c_{\boldsymbol{A}}^-$. Unlike in all previous experiments, increasing the cost of deletions also leads to a significant increase for the same edit distance budget $\epsilon$. On PROTEINS, choosing $c_{\boldsymbol{A}}^+ = 1$ and $c_{\boldsymbol{A}}^- = 4$ even yields stronger robustness guarantees than $c_{\boldsymbol{A}}^+ = 4$ and $c_{\boldsymbol{A}}^- = 1$. For example, the average certified accuracies for budget $\epsilon$ are $61.2\%$ and $55.6\%$, respectively (see Fig. 16). This may be caused by one of multiple factory. Firstly, we have a different task – graph instead of node classificaton. Secondly, we have a different architecture that aggregates

information from the entire graph via max-pooling instead of just using local information. Thirdly, the considered graphs have a significantly different distribution than Cora-ML and Citeseer. For instance, Cora-ML and Citeser consist of 2708 and 3327 nodes, respectively, whereas the average number of nodes in PROTEINS and MUTAG is 17.9 and 39.1. Cora-ML and Citeseer also have much sparser adjacency matrices, with just $0.14\%$ and $0.08\%$ non-zero entries.

**Sparsity-aware randomized smoothing for attribute and adjacency perturbations.** Finally, we perform additional experiments with sparsity-aware randomized smoothing [14]. As we discussed in Section 5.1, this randomized smoothing method preserves isomorphism equivariance and can thus be used to prove adversarial robustness for arbitrary isomorphism equivariant models and tasks w.r.t. graph edit distance and both attribute and adjacency perturbations. Figs. 18 to 29 show the certified accuracy of graph convolutional networks, graph attention networks [128] (GAT), and APPNP on Cora-ML and Citeseer for varying costs of attribute and adjacency perturbations with smoothing parameters $p_X^+ = 0.001, p_X^- = 0.8, p_A^+ = 0, p_A^- = 0$ and $p_X^+ = 0, p_X^- = 0, p_A^+ = 0.001, p_A^- = 0.8$. While the different models and datasets differ slightly with respect to (certified) accuracy, we observe the same effect as in our previous node classification experiments: Only increasing the cost of insertions has a large effect on provable robustness for any graph edit distance $\epsilon$, which suggests that these perturbations are significantly more effective at changing the prediction of the models.

**Conclusion.** Overall, there are three main takeaways from evaluating the different graph edit distance robustness guarantees. Firstly, even though evaluating the graph edit distance is not tractable, it is possible to prove robust under it to provide sound robustness guarantees for isomorphism equivariant tasks. Secondly, unless they use very loose relaxations (like interval bound propagation), generalizing existing guarantees to non-uniform costs provides more fine-grained insights into the adversarial robustness of different graph and node classification architectures. Thirdly, equivariance-preserving randomized smoothing is an effective way of obtaining sound robustness guarantees for equivariant tasks where discrete groups act on discrete data.

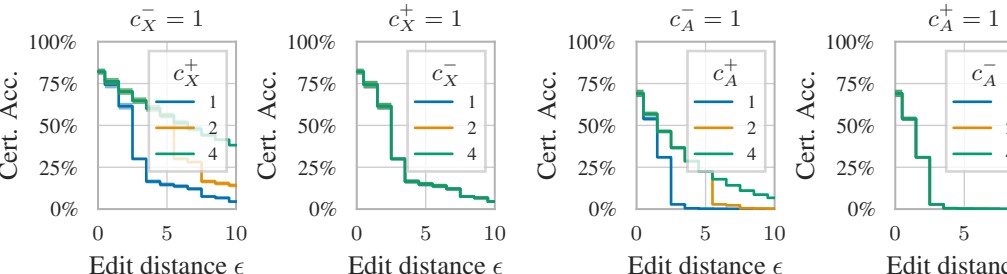

Figure 18: Randomized smoothing robustness guarantees for GCNs under attribute perturbations on Cora-ML. The Smoothing parameters are set to $p_X^+ = 0.001, p_X^- = 0.8, p_A^+ = 0, p_A^- = 0$.

Figure 19: Randomized smoothing robustness guarantees for GCNs under adjacency perturbations on Cora-ML. The moothing parameters are set to $p_X^+ = 0, p_X^- = 0, p_A^+ = 0.001, p_A^- = 0.8$.

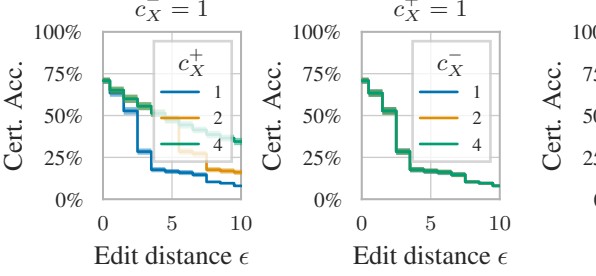

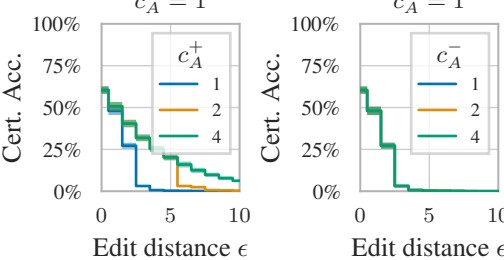

Figure 20: Randomized smoothing robustness guarantees for GCNs under attribute perturbations on Citeseer. The Smoothing parameters are set to $p_X^+ = 0.001, p_X^- = 0.8, p_A^+ = 0, p_A^- = 0$.

Figure 21: Randomized smoothing robustness guarantees for GCNs under adjacency perturbations on Citeseer. The moothing parameters are set to $p_X^+ = 0, p_X^- = 0, p_A^+ = 0.001, p_A^- = 0.8$.

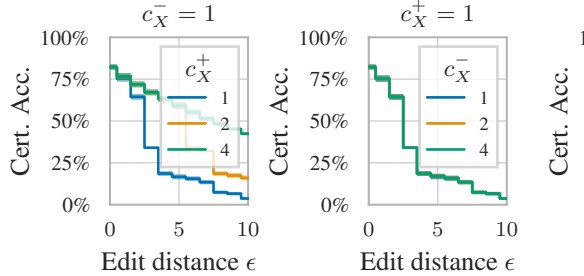
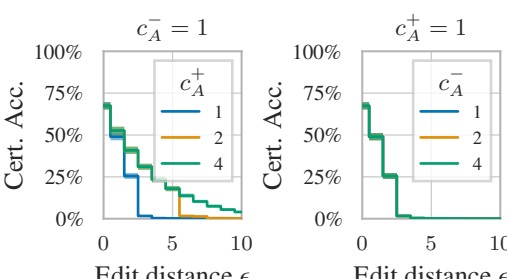

Figure 22: Randomized smoothing robustness guarantees for GATs under attribute perturbations on Cora-ML. The Smoothing parameters are set to $p_{\mathbf{X}}^+ = 0.001, p_{\mathbf{X}}^- = 0.8, p_{\mathbf{A}}^+ = 0, p_{\mathbf{A}}^- = 0$.

Figure 23: Randomized smoothing robustness guarantees for GATs under adjacency perturbations on Cora-ML. The moothing parameters are set to $p_{\mathbf{X}}^+ = 0, p_{\mathbf{X}}^- = 0, p_{\mathbf{A}}^+ = 0.001, p_{\mathbf{A}}^- = 0.8$.

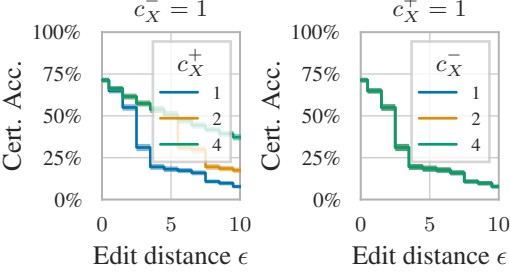
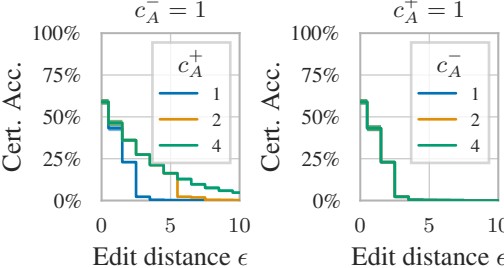

Figure 24: Randomized smoothing robustness guarantees for GATs under attribute perturbations on Citeseer. The Smoothing parameters are set to $p_{\mathbf{X}}^+ = 0.001, p_{\mathbf{X}}^- = 0.8, p_{\mathbf{A}}^+ = 0, p_{\mathbf{A}}^- = 0$.

Figure 25: Randomized smoothing robustness guarantees for GATs under adjacency perturbations on Citeseer. The moothing parameters are set to $p_{\mathbf{X}}^+ = 0, p_{\mathbf{X}}^- = 0, p_{\mathbf{A}}^+ = 0.001, p_{\mathbf{A}}^- = 0.8$.

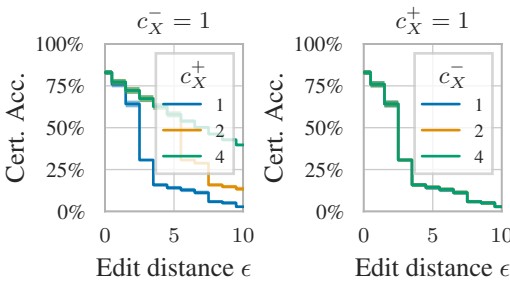
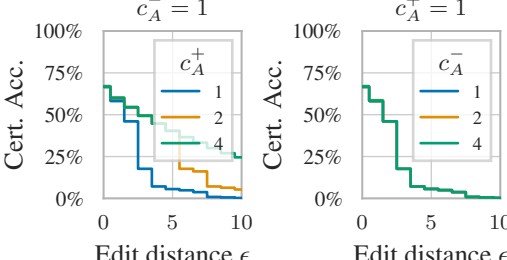

Figure 26: Randomized smoothing robustness guarantees for APPNP under attribute perturbations on Cora-ML. The Smoothing parameters are set to $p_{\mathbf{X}}^+ = 0.001, p_{\mathbf{X}}^- = 0.8, p_{\mathbf{A}}^+ = 0, p_{\mathbf{A}}^- = 0$.

Figure 27: Randomized smoothing robustness guarantees for APPNP under adjacency perturbations on Cora-ML. The moothing parameters are set to $p_{\mathbf{X}}^+ = 0, p_{\mathbf{X}}^- = 0, p_{\mathbf{A}}^+ = 0.001, p_{\mathbf{A}}^- = 0.8$.

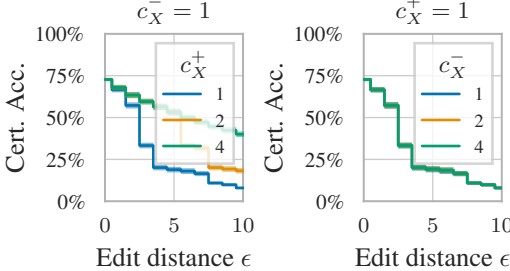
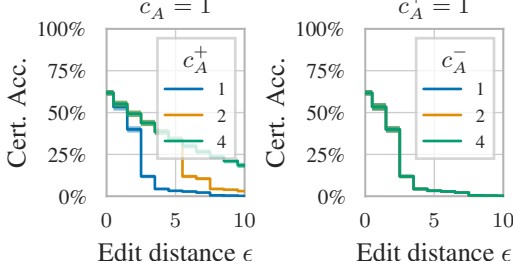

Figure 28: Randomized smoothing robustness guarantees for APPNP under attribute perturbations on Citeseer. The Smoothing parameters are set to $p_{\mathbf{X}}^+ = 0.001, p_{\mathbf{X}}^- = 0.8, p_{\mathbf{A}}^+ = 0, p_{\mathbf{A}}^- = 0$.

Figure 29: Randomized smoothing robustness guarantees for APPNP under adjacency perturbations on Citeseer. The moothing parameters are set to $p_{\mathbf{X}}^+ = 0, p_{\mathbf{X}}^- = 0, p_{\mathbf{A}}^+ = 0.001, p_{\mathbf{A}}^- = 0.8$.

## A.4 Adversarial attacks

As discussed in Section 5, any traditional adversarial attack is a valid adversarial attack under our proposed notion of adversarial robustness for equivariant tasks. This holds no matter if the equivariances of task $y$ and model $f$ match or not. As an example we conduct adversarial attacks on a PointNet classifier and a graph convolutional network. We use the same datasets, models and hyperparameters as for the experiments shown in Fig. 3 (without randomized smoothing) and Fig. 10. To attack the PointNet classifier, we use a single gradient step w.r.t. cross-entropy loss, which we scale to have an $\ell_2$ norm of $\epsilon$. To attack the graph convolutional network, we use the method from Section 4.4 of [10], which is directly derived from their certification procedure. As can be seen, future work is needed to further improve the robustness of equivariant models to our proposed perturbation model.

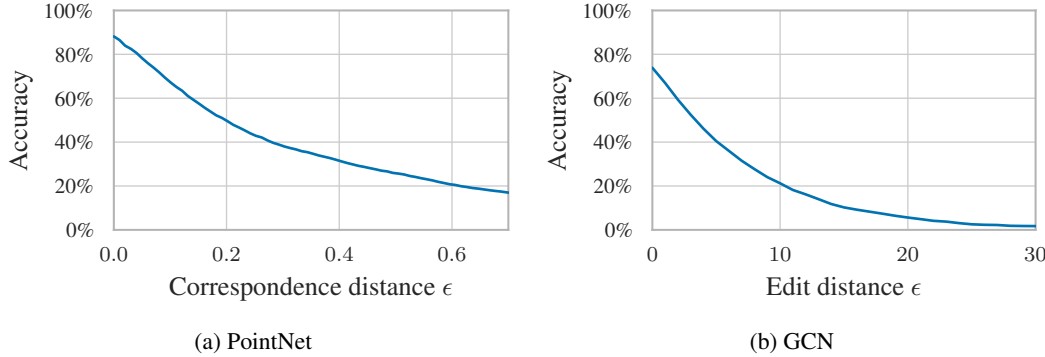

(a) PointNet                    (b) GCN

Figure 30: Adversarial attacks on (a) PointNet using a single gradient step with $\ell_2$ norm $\epsilon$ and (b) GCN (with perturbation costs $c_X^+ = c_X^- = 1$) using the method from Section 4.4 of [10].

# B  Full experimental setup

## B.1  Point cloud classification

For our experiments on point cloud classification, we replicate the experimental setup from [26], where randomized smoothing was also applied to point cloud classification. We just use different base models, which are not rotation and translation invariant. Each experiment (including training) is repeated 5 times using different random seeds. We report the mean and standard deviation in certificate strength across all seeds.

### B.1.1  Data

All experiments are performed on ModelNet[116] which consists of 9843 training samples and 2468 test samples from 40 classes. We apply the same preprocessing steps as in [114] to transform the CAD models into point clouds consisting of 1024 points, i.e. random sampling on the surface of the objects and normalization into the unit sphere. We use 20% of the training data for validation. All certificates are evaluated on the test data.

### B.1.2  Models

We use two different models, PointNet [114] and DGCNN [115].

For PointNet, we use three linear layers $(64, 128, 1024$ neurons) before pooling and three layers $(512, 256, 40$ neurons) after pooling. Before feeding input point clouds into the model, we multiply them with a matrix predicted by a T-Net, i.e. another PointNet with three linear layers $(64, 128, 1024$ neurons) before pooling and three linear layers $(512, 256, 3 \cdot 3$ neurons) after pooling. We use batch normalization $(\epsilon = 1e-5, \mathrm{momentum} = 0.1)$ for all layers, except the output layer. We use dropout $(p = 0.4)$ for the second-to-last layer.

For DGCNN, we use four dynamic graph convolution layers $(64, 64, 64, 128$ neurons) with k-nearest neighbor graph construction $(k = 20)$. The outputs of all four layers are concatenated, passed through a linear layer $(1024$ neurons), then max-pooled across the node dimension and finally passed through three more linear layers $(512, 256, 40$ neurons). We use batch normalization for all layers $(\epsilon = 1e-5, \mathrm{momentum} = 0.1)$. We use dropout $(p = 0.4)$ for the second-to-last and third-to-last layer. Before feeding input point clouds into the model, we multiply them with a $3 \times 3$ matrix predicted by the same T-Net architecture we described for PointNet.

### B.1.3  Training

We use the same training parameters as in [26]. We train for 200 (PointNet) or 400 (DGCNN) epochs using Adam $(\beta_1 = 0.9, \beta_2 = 0.99, \epsilon = 1e-8, \mathrm{weight\_decay} = 1e^-4)$ with a learning rate of $0.001$ and exponential weight decay with factor $0.7$ every 20 (PointNet) or 40 (DGCNN) epochs. We use a batch size of 128 for PointNet and 32 for DGCNN. We add the T-Net loss proposed by [114] to the cross entropy loss with a weight of $0.0001$. The data is randomly augmented via scaling by a factor uniformly sampled from $[0.8, 1.25]$. We additionally add isotropic Gaussian noise with the same standard deviation we use for randomized smoothing $(0.05, 0.1, 0.15, 0.2$ or $0.25)$.

### B.1.4  Certification parameters

We perform randomized smoothing with isotropic Gaussian noise $(\sigma \in \{0.05, 0.1, 0.15, 0.2$ or $0.25\})$ and majority voting [20]. We use $1,000$ Monte Carlo samples for prediction and $10,000$ Monte Carlo samples for certification. We compute the Clopper-Pearson confidence intervals for class probabilities with $\alpha = 0.001$, i.e. all guarantees hold with high probability $99.9\%$.

### B.1.5  Computational resources

All experiments involving PointNet were performed on a Xeon E5-2630 v4 CPU @ $2.20\,\mathrm{GHz}$ with a NVIDA GTX 1080TI GPU. The average time for training a model was $21.1\,\mathrm{min}$. The average time for certifying the robustness of a single model on the entire test set for 50 different budgets $\epsilon$ was $81.7\,\mathrm{min}$.

All experiments involving DGCNN were performed on an AMD EPYC 7543 CPU @ 2.80GHz with a NVIDA A100 (40 GB) GPU. The average time for training a model was $161.9\,\text{min}$. The average time for certifying the robustness of a single model on the entire test set for 50 different budgets $\epsilon$ was $601.7\,\text{min}$.

## B.2 Molecular force prediction

Like with point cloud classification, we train and certify 5 different models using different random seeds on a common benchmark dataset. We report the mean and standard deviation in certificate strength across all seeds.

### B.2.1 Data

We use the original (not the revised) MD17 [118] collection of datasets. It consists of 8 different datasets, each consisting of a large number (between $133\,770$ and $993\,237$) of spatial configurations of a specific molecule. For each dataset, we use 1000 randomly chosen configurations for training, 1000 for validation and 1000 for evaluating the robustness guarantees. We use the same data split for all experiments.

### B.2.2 Models

We use DimeNet++ [117], SchNet [119] and SphereNet [120] as our base model. All models predict atomic energies. Force are calculated via automatic differentiation w.r.t. the input coordinates.

For DimeNet++, we use the default model parameters for MD17 force predictions specified in [117].. There are 4 layers with 128 hidden channels. The triplet embedding size is 64. The basis embedding size is 8. The output embedding size is 256. The number of basis functions is 8 (bilinear), 7 (spherical), 6 (radial). The cutoff radius for graph construction is $5\,\text{Å}$. The number of residual layers before the skip connection is 1. The number of residual layers after the skip connection is 2.

For SchNet, we use 128 hidden channels and 128 filters. We set the number of interaction blocks to 6 and use 50 Gaussians. The cutoff distance is $10\,\text{Å}$, with a maximum number of 32 neighbors. We use addition as our global readout function.

For SphereNet, we use 4 layers with 128 hidden channels. The triplet embedding size is 64. The basis embedding size is 8 for distance, angle and torsion. The output embedding size is 256. The number of basis functions is 8 (bilinear), 7 (spherical), 6 (radial). The cutoff radius for graph construction is $5\,\text{Å}$. We use swish activation functions. The number of residual layers before the skip connection is 1. The number of residual layers after the skip connection is 2.

### B.2.3 Training parameters

To train DimeNet++ and Schnet, we use ADAM with a learning rate of 0.001. We use linear warmup with exponential decay as our learning rate scheduler (1000 warmup steps, 400,000 decay steps, decay rate of 0.01) and train until convergence with a patience of 50 epochs and a convergence threshold of $10^{-4}$.

To train SphereNet, we use ADAM with a learning rate of 0.001. We use "reduce on plateau' as our learning rate scheduler (decay factor of 0.8, patience of 80 epochs, convergence threshold of $10^{-4}$, cooldown of 10 epochs) and train until convergence with a patience of 50 epochs and a convergence threshold of $10^{-4}$.

We do not add randomized smoothing noise during the training process.

### B.2.4 Certification parameters

We perform randomized smoothing with isotropic Gaussian noise ($\sigma \in \{1\,\text{fm}, 10\,\text{fm}, 100\,\text{fm}\}$) and center smoothing [93] with default parameters $\alpha_1 = 0.005, \alpha_2 = 0.005, \Delta = 0.05$, i.e. all guarantees hold with high probability 99%. We use 10,000 samples for prediction, 10,000 samples to test for abstention and 10,000 samples for certification. In our experiments the smoothed model never abstained.

### B.2.5 Computational resources

All experiments for molecular force prediction were performed on a Xeon E5-2630 v4 CPU @ 2.20 GHz with a NVIDA GTX 1080TI GPU. We trained 4 models simultaneously. The average time for training a DimeNet++ model was 28.9 h. The average time for certifying the robustness of a single DimeNet++ model on 1000 molecule configurations for 1000 different budgets $\epsilon$ was 10.49 h.

### B.3 Node and graph classification

In Section 5.2 we generalized six approaches for proving the robustness of graph neural networks with respect to graph edit distances. While the approaches differ, we mostly use the same datasets and models. As before, we perform each experiment (including training) with 5 different random seeds and report the mean and standard deviation in certificate strength across all seeds.

### B.3.1 Data

We use two standard node classification datasets, Cora-ML [121, 122] (2,708 nodes, 10,556 edges, 1,433 features, 7 classes) and Citeseer [124](3,327 nodes, 9,104 edges, 3,703 features, 6 classes), and two graph classification sets, PROTEINS (1,113 graphs, 3 features, 2 classes) and MUTAG (188 graphs, 7 features, 2 classes), which are part of the TUDataset [125]. These datasets were also used in the papers that proposed the original certification procedures. With Cora-ML and Citeseer, we follow the procedure from [14] and use 20 nodes per class for training, 20 nodes per class for validation and the remainder for evaluating the certificates. With PROTEINS and MUTAG, we use 30% of graphs for training, 20% for validation and the remainder for evaluating the certificates.

### B.3.2 Models

The main model we use for node classification is a standard 2-layer graph convolutional network with ReLU nonlinearities and 32 hidden features. We insert self-loops and perform degree normalization via $\boldsymbol{D}^{-1/2}\boldsymbol{A}\boldsymbol{D}^{-1/2}$.

For node classification, we use the architecture described in [15], i.e. a single layer GCN with 64 neurons, followed by a linear layer and max-pooling. We also insert self-loops, but perform degree normalization via $\boldsymbol{D}^{-1}\boldsymbol{A}$.

With sparsity-aware randomized smoothing, we additionally use graph attention networks [128] and APPNP [122]. We implement all GNNs with two-layers and 32 hidden dimensions, except GAT for which we implement 8 hidden dimensions and 8 attention heads. For APPNP we further set k_hops=10 and teleport probability $\alpha = 0.15$.

For the experiments with certificates for the $\pi$-PPNP architecture we set teleport probability $\alpha = 0.15$ and use a hidden dimension of 64.

### B.3.3 Training parameters

We train models for a maximum of 1,000 ($\pi$-PPNP), 3,000 (other node classifiers) or 200 (graph classifiers) epochs with Adam ($lr = 1e-3$, $\beta_1 = 0.9$, $\beta_2 = 0.99$, $\epsilon = 1e-8$) and cross entropy loss. For node classification we additionally use weight decay $5e-4$ and early stopping after 50 epochs. For node-classification we perform full-batch gradient descent. For graph classification we use minibatches of size 20. For the sparsity-ware randomized smoothing experiments we use a dropout of 0.5 on the hidden node representations after the first graph convolution for GCN and GAT, and additionally on the attention coefficients for GAT. For randomized smoothing we additionally add sparsity-aware noise to the training data. We use the same flip probabilities as for certification.

### B.3.4 Certification parameters

For interval bound propagation and the convex outer adversarial polytope method, we set the nodewise local budgets $\rho_1, \ldots, \rho_N$ to 1% of input features, just like in [10].

For our generalization of the graph classification certificate from, we set the local budget $\rho_n$ of node $n$ with degree $d_n$ to $\min(0, d_n - \max_m d_m + 3)$, just like in [15]. We additionally enforce symmetry

of the adjacency matrix via Lagrange dualization (see Appendix A.3) and perform 200 alternating optimization steps using the default parameters from [15].

We perform randomized smoothing with sparse noise ($p_{\mathbf{X}}^+ = 0.001$, $p_{\mathbf{X}}^- = 0.8$, $p_{\mathbf{A}}^+ = 0$, $p_{\mathbf{A}}^- = 0$, or $p_{\mathbf{X}}^+ = 0$, $p_{\mathbf{X}}^- = 0$, $p_{\mathbf{A}}^+ = 0.001$, $p_{\mathbf{A}}^- = 0.8$) and majority voting [14]. For node classification we use $1,000$ Monte-Carlo samples for prediction and $1,000,000$ Monte-Carlo samples for certification. For graph classification we use $1,000$ Monte-Carlo samples for prediction and $10,000$ Monte-Carlo samples for certification. We compute the Clopper-Pearson confidence intervals for class probabilities with $\alpha = 0.01$, i.e. all guarantees hold with high probability $99\%$.

### B.3.5 Computational resources

All experiments were performed on a Xeon E5-2630 v4 CPU @ $2.20\,\mathrm{GHz}$ with a NVIDA GTX 1080TI GPU. With interval bound propagation, the average time of training and certifying a model for 100 budgets $\epsilon$ was $1.2\,\mathrm{min}$ on Cora-ML and $3.0\,\mathrm{min}$ on Citeseer. With the convex outer adversarial polytope method, the average time of training and certifying a model for 100 budgets $\epsilon$ was $9.7\,\mathrm{min}$ on Cora-ML and $14.4\,\mathrm{min}$ on Citeseer. With the linearization and dualization method for graph classifiers, the average time of the average time of training and certifying a model for 100 budgets $\epsilon$ on the entire test set was $1.2\,\mathrm{h}$ on MUTAG and $48.1\,\mathrm{h}$ on PROTEINS. With the policy iteration method for $\pi$-PPNP, the average time for training and certifying a model for 10 attack strengths was $126.1\,\mathrm{min}$. With sparsity-aware randomized smoothing, the average time for training and certifying a model for 1000 budgets $\epsilon$ was $267.7\,\mathrm{min}$.

### B.4 Third-party assets

Since we extend various existing methods for proving the robustness of graph neural networks, we naturally built upon their reference implementations. For interval bound propagation and convex outer adversarial polytopes, we extend the reference implementation from [10] (https://github.com/danielzuegner/robust-gcn). For the linearization and dualization method, we extend the reference implementation from [15] (https://github.com/RobustGraph/RoboGraph). For the policy iteration method, we extend the reference implementation from [11] (https://github.com/abojchevski/graph_cert). For sparsity-aware randomized smoothing on graphs, we use the reference implementation from [14] (https://github.com/abojchevski/sparse_smoothing).

For equivariance-preserving randomized smoothing on point clouds, we use the training and sampling procedure from [26] (https://github.com/jan-schuchardt/invariance-smoothing).

All of the above are available under MIT license.

To train models for molecular force prediction, we use a pre-release version of code from [129] (https://github.com/wollschl/uncertainty_for_molecules), which we will include in our reference implementation.

## C   Proof of Proposition 1

In the following, we show that the action-induced distance is the only function $\hat{d}_{\text{in}} : \mathbb{X} \times \mathbb{R} \to \mathbb{R}_+$ that fulfills all three desiderata defined in Section 4.1. These were

- $\forall x, x' \in \mathbb{X}, g \in \mathbb{G} : \hat{d}_{\text{in}}(x, g \bullet x') = \hat{d}_{\text{in}}(x, x')$,

- $\forall x, x' : \hat{d}_{\text{in}}(x, x') \leq d_{\text{in}}(x, x')$,

- $\hat{d}_{\text{in}}(x, x') = \max_{\gamma \in \mathbb{D}} \gamma(x, x')$,

where $\mathbb{D}$ is the set of functions from $\mathbb{X} \times \mathbb{X}$ to $\mathbb{R}_+$ that fulfill the first two desiderata

**Proposition 1.** *A function $\hat{d}_{\text{in}} : \mathbb{X} \times \mathbb{X} \to \mathbb{R}_+$ that fulfills all three desiderata for any original distance function $d_{\text{in}} : \mathbb{X} \times \mathbb{X} \to \mathbb{R}_+$ exists and is uniquely defined: $\hat{d}_{\text{in}}(x, x') = \min_{g \in \mathbb{G}} d_{\text{in}}(x, g \bullet x')$.*

*Proof.* Consider a specific pair $x, x' \in \mathbb{X}'$. The first desideratum states that we must have $\hat{d}_{\text{in}}(x, x') = \hat{d}_{\text{in}}(x, g \bullet x')$ for all $g \in \mathbb{G}$. The second desideratum states that we must have $\hat{d}_{\text{in}}(x, g \bullet x') \leq d_{\text{in}}(x, g \bullet x')$ for all $g \in \mathbb{G}$. Thus, by the transitive property, fulfilling both desiderata simultaneously for our specific $x$ and $x'$ is equivalent to $\forall g \in \mathbb{G} : \hat{d}_{\text{in}}(x, x') \leq d_{\text{in}}(x, g \bullet x')$. This is equivalent to $\hat{d}_{\text{in}}(x, x') \leq \min_{g \in \mathbb{G}} d_{\text{in}}(x, g \bullet x')$. The left side of the equality is naturally maximized to fulfill the third desideratum when we have a strict equality, i.e. $\hat{d}_{\text{in}}(x, x') = \min_{g \in \mathbb{G}} d_{\text{in}}(x, g \bullet x')$.  $\square$

# D  Combining smoothing measures and smoothing schemes

Most randomized smoothing literature can be divided into works that focus on investigating properties of different smoothing measures (e.g. [14, 20, 60, 64, 65, 94, 95] and works that propose new smoothing schemes (e.g. [20, 75, 77, 93]. Measure-focused works usually only consider classification tasks, while scheme-focused works usually only consider Gaussian measures. But, as we shall detail in the following, any of the smoothing schemes discussed in Section 5.1 can be used with arbitrary smoothing measures to enable certification for various input distances $d_{\text{in}}$ and output distances $d_{\text{out}}$.

## D.1  Smoothing measures

Measure-focused works usually consider a base classifier $h : \mathbb{X} \to \mathbb{Y}$ with $\mathbb{Y} = \{1, \ldots, K\}$. They define a family of smoothing measures $(\mu_x)_{x \in \mathbb{X}}$ to construct the smoothed classifier $f(x) = \max_{k \in \mathbb{Y}} \Pr_{z \sim \mu_x}[h(z) = k]$. To certify the predictions of such a smoothed classifier, they derive lower and upper bounds on the probability of classifying a perturbed input $x' \in \mathbb{X}$ as a specific class $k \in \{1, \ldots, k\}$, i.e. $\underline{p_{h,x',k}} \leq \Pr_{z \sim \mu_{x'}}[h(z) = k] \leq \overline{p_{h,x',k}}$. These bounds are usually obtained by finding the least / most robust model that has the same clean prediction probability as base classifier $h$, i.e.

$$\underline{p_{h,x',k}} = \min_{\tilde{h}:\mathbb{X} \to \mathbb{Y}} \Pr_{z \sim \mu_{x'}}\left[\tilde{h}(z) = k\right] \quad \text{s.t.} \quad \Pr_{z \sim \mu_x}\left[\tilde{h}(z) = k\right] = \Pr_{z \sim \mu_x}[h(z) = k],$$

$$\overline{p_{h,x',k}} = \max_{\tilde{h}:\mathbb{X} \to \mathbb{Y}} \Pr_{z \sim \mu_{x'}}\left[\tilde{h}(z) = k\right] \quad \text{s.t.} \quad \Pr_{z \sim \mu_x}\left[\tilde{h}(z) = k\right] = \Pr_{z \sim \mu_x}[h(z) = k]$$

As long as $\mu_x$ and $\mu_{x'}$ have a density, these problems[3] can always be solved exactly via the Neyman-Pearson lemma [130].

For appropriately chosen smoothing measures $(\mu_x)_{x \in \mathbb{X}}$, one can then identify that the perturbed probability bounds $\underline{p_{h,x',k}}$ and $\overline{p_{h,x',k}}$ only depend on a certain distance function $d_{\text{in}} : \mathbb{X} \times \mathbb{X} \to \mathbb{R}_+$. For example:

- For $\mathbb{X} = \mathbb{R}^N$, and $\mu_x = \mathcal{N}(\boldsymbol{x}, \sigma \cdot \boldsymbol{I})$, the bounds $\underline{p_{h,x',k}}$ and $\overline{p_{h,x',k}}$ only depend on $d_{\text{in}}(\boldsymbol{x}, \boldsymbol{x}') = ||\boldsymbol{x} - \boldsymbol{x}'||_2$ and are monotonically decreasing / increasing [20].
- For $\mathbb{X} = \mathbb{R}^N$, and $\mu_x = \text{Laplace}(\boldsymbol{x}, \sigma \cdot \boldsymbol{I})$, the bounds $\underline{p_{h,x',k}}$ and $\overline{p_{h,x',k}}$ only depend on $d_{\text{in}}(\boldsymbol{x}, \boldsymbol{x}') = ||\boldsymbol{x} - \boldsymbol{x}'||_1$ and are monotonically decreasing / increasing [95].
- For $\mathbb{X} = \{0, 1\}$, and $\mu_x$ being the measure associated with i.i.d. flipping of input bits, the bounds $\underline{p_{h,x',k}}$ and $\overline{p_{h,x',k}}$ only depend on $d_{\text{in}}(\boldsymbol{x}, \boldsymbol{x}') = ||\boldsymbol{x} - \boldsymbol{x}'||_0$ and are monotonically decreasing / increasing [94].

For an overview of various additive smoothing schemes and their associated input distances, see [95]. Note that works on non-additive smoothing measures, like ablation smoothing [131] or derandomized smoothing [132], also provide bounds $\underline{p_{h,x',k}}$ and $\overline{p_{h,x',k}}$ as a function of some input distance $d_{\text{in}}$, such as the number of perturbed pixels or the size of an adversarial image patch.

## D.2  Smoothing schemes

Scheme-focused works include majority voting [20], expected value smoothing [75], median smoothing [77] and center smoothing [93]. These works only consider Gaussian smoothing, i.e. $\mathbb{X} = \mathbb{R}^N$ and $\mu_x = \mathcal{N}(\boldsymbol{x}, \sigma \cdot \boldsymbol{I})$. Each one proposes a different prediction procedure and certification procedure. The prediction procedures provide a sampling-based approximations of the smoothed prediction $\xi(\mu \circ h^{-1})$ and does not depend on the choice of smoothing measure. The certification procedures can be adapted to arbitrary smoothing measures as follows.

**Majority voting.** Majority voting assumes a base classifier $h : \mathbb{X} \to \mathbb{Y}$ with $\mathbb{Y} = \{1, \ldots, K\}$. It constructs a smoothed classifier $f(x) = \max_{k \in \mathbb{Y}} \Pr_{z \sim \mu_x}[h(z) = k]$. The prediction $f(x) = k^*$ of this smoothed classifier is robust if $\underline{p_{h,x',k^*}} > \max_{k \neq k^*} \overline{p_{h,x',k}}$ or if $\underline{p_{h,x',k^*}} > 0.5$. This certification procedure can be adapted to any other smoothing measure by inserting the corresponding lower and

---

[3]or continuous relaxations thereof, if any constant likelihood region has non-zero measure

upper bounds. It can be generalized to multi-output classification tasks with $\mathbb{Y} = \{1, \ldots, K\}^M$ (e.g. segmentation) by applying the certification procedure independently to each output dimension.

**Expected value and median smoothing.** Median smoothing and expected value smoothing assume a base regression model $h : \mathbb{X} \to \mathbb{R}$. They construct a smoothed model $f(\boldsymbol{x})$ via the expected value or median of pushforward measure $\mu_x \circ h^{-1}$. To certify robustness, they require lower and upper bounds on the cumulative distribution function $\Pr_{z \sim \mu'_x} \mathbb{1}[h(z) \leq a_t]$ for a finite range of thresholds $-\infty \leq a_t \leq \cdots \leq a_T \leq \infty$. Since each of the indicator functions $\mathbb{1}[h(z) \leq a_t]$ can be thought of as a binary classifier $g_t : \mathbb{X} \to \{0, 1\}$, these lower and upper bounds are given by $\underline{p_{g_t, x', 1}}$ and $\overline{p_{g_t, x', 1}}$. These certification procedures can thus be adapted to any other smoothing measure by inserting the corresponding lower and upper bounds. They can be generalized to multi-output regression tasks with $\mathbb{Y} = \mathbb{R}^M$ by applying the certification procedure independently to each output dimension. Instead of proving that the smoothed prediction $f(\boldsymbol{x})$ remains constant, these smoothing schemes guarantee that the output remains in a certified hyperrectangle $\mathbb{H} = \{\boldsymbol{y} \in \mathbb{R}^M \mid l_m \leq x_m \leq u_m\}$. One can then prove that $d_{\text{out}}(f(\boldsymbol{x}), f(\boldsymbol{x}')) \leq \delta$ by verifying that $\{\boldsymbol{y} \in \mathbb{R}^M \mid d_{\text{out}}(f(\boldsymbol{x}), \boldsymbol{y}) \leq \delta\} \subseteq \mathbb{H}$, i.e. the hyperrectangle contains a $d_{\text{out}}$-ball of radius $\delta$.

**Center smoothing.** Center smoothing is compatible with any model $h : \mathbb{X} \to \mathbb{Y}$, as long as the output space $\mathbb{Y}$ fulfills a relaxed triangle inequality. It constructs a smoothed model $f(\boldsymbol{x})$ via the center of the smallest ball that has at least $50\%$ probability under pushforward measure $\mu_{\boldsymbol{x}} \circ h^{-1}$. To certify robustness, it requires bounds on the cumulative distribution function $\Pr_{z \sim \mu'_x} \mathbb{1}[d_{\text{out}}(f(\boldsymbol{x}), h(\boldsymbol{z}) \leq a_t]$. Thus, like with expected value and median smoothing, this certification procedure can be adapted to any other smoothing measure by inserting the corresponding bounds $\underline{p_{g_t, x', 1}}$ and $\overline{p_{g_t, x', 1}}$. This smoothing scheme directly provides an upper bound on $d_{\text{out}}(f(\boldsymbol{x}), f(\boldsymbol{x}'))$.

Informally speaking, any of these schemes can be adapted to non-Gaussian measures by replacing any occurence of $\Phi(\Phi^{-1}(\ldots) - \ldots)$ and $\Phi(\Phi^{-1}(\ldots) + \ldots)$, which are the formulae for $\underline{p_{h, x', k}}$ and $\overline{p_{h, x', k}}$ of Gaussian measures [20], with the bounds of another smoothing measure. For example, one can combine the bounds from derandomized smoothing with the center smoothing certification procedure to prove robustness of generative image reconstruction models to patch-based attacks.

# E   Equivariance-preserving randomized smoothing

In the following, we provide more formal justifications for our randomized smoothing framework, that we introduced in section Section 5.1 and that is visualized in Fig. 31. We first verify the correctness of our sufficient condition for equivariance-preserving randomized smoothing. We then consider the different equivariances preserved by different smoothing schemes and measures. We provide a tabular overview of different schemes, measures and their properties in Tables 1 and 2.

**Concerning notation for pushforward measures.** Consider measurable spaces $(\mathbb{X}, \mathbb{D})$ and $(\mathbb{V}, \mathbb{F})$, and a measurable function $h : \mathbb{X} \to \mathbb{V}$. We write $h^{-1}$ to refer to the preimage of function $h$, not its inverse. That is, for any $\mathbb{A} \subseteq \mathbb{V}$ we have $h^{-1}(\mathbb{A}) = \{x \in \mathbb{X} \mid h(x) \in \mathbb{A}\}$. Because $h$ is measurable, we have $h^{-1}(\mathbb{A}) \in \mathbb{D}$ whenever $\mathbb{A} \in \mathbb{F}$. Thus, the expression $\mu \circ h^{-1}(\mathbb{A})$ refers to the measure of all elements of $\mathbb{X}$ that get mapped into $\mathbb{A} \in \mathbb{F}$, i.e. it is the pushforward measure of $\mu$.

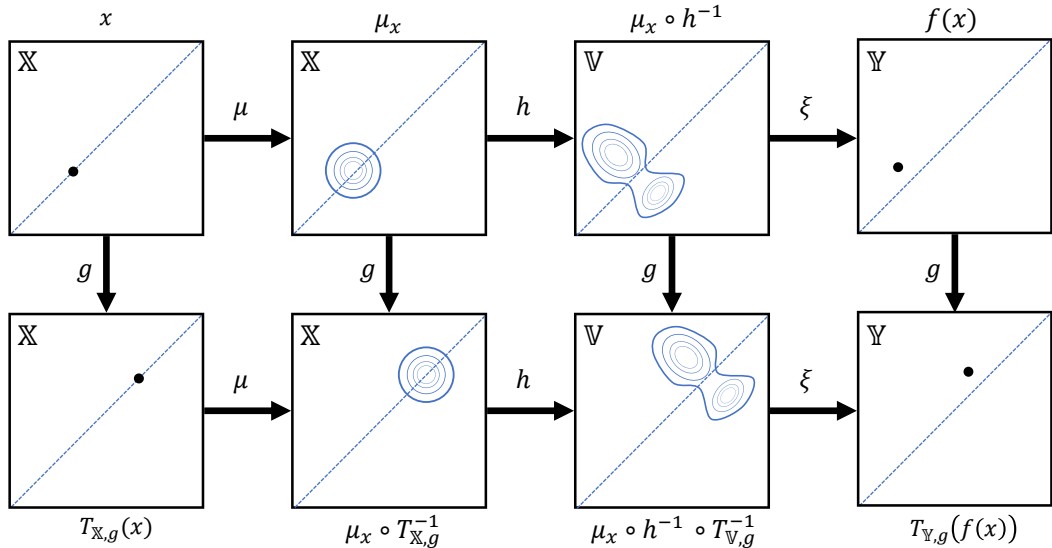

Figure 31: Example of equivariance-preserving randomized smoothing for $\mathbb{X} = \mathbb{V} = \mathbb{Y} = \mathbb{R}^2$ and diagonal translation group $\mathbb{G} = (\mathbb{R}, +)$, which acts via $T_{\cdot,g}(\boldsymbol{x}) = \boldsymbol{x} + g \cdot \boldsymbol{1}$. We construct our smoothed model using Gaussian measures $\mu_{\boldsymbol{x}} = \mathcal{N}(\boldsymbol{x}, \sigma \cdot \boldsymbol{I})$, base model $h : \mathbb{X} \to \mathbb{V}$ and expected value smoothing scheme $\xi : \Delta(\mathbb{V}, \mathbb{F}) \to \mathbb{Y}$, which maps from measures on intermediate space $(\mathbb{V}, \mathbb{F})$ to output space $\mathbb{Y}$. Measures are visualized via isocontours of their densities. Because the family of measures $(\mu_{\boldsymbol{x}})_{\boldsymbol{x} \in \mathbb{X}}$ is equivariant (left cycle), the base model $h$ is equivariant (central cycle), and the smoothing scheme $\xi$ is equivariant (right cycle), the smoothed model $f(\boldsymbol{x}) = \xi(\mu_{\boldsymbol{x}} \circ h^{-1})$ is also equivariant to diagonal translation.

## E.1   Proof of Proposition 3

**Proposition 3.** *Assume two measurable spaces $(\mathbb{X}, \mathbb{D})$, $(\mathbb{V}, \mathbb{F})$, an output space $\mathbb{Y}$ and a measurable base model $h : \mathbb{X} \to \mathbb{V}$ that is equivariant with respect to the action of group $\mathbb{G}$. Further assume that $\mathbb{G}$ acts on $\mathbb{X}$ and $\mathbb{V}$ via measurable functions. Let $\xi : \Delta(\mathbb{V}, \mathbb{F}) \to \mathbb{Y}$ be a smoothing scheme that maps from the set of probability measures $\Delta(\mathbb{V}, \mathbb{F})$ on intermediate space $(\mathbb{V}, \mathbb{F})$ to the output space. Define $T_{\mathbb{X},g}(\cdot)$ to be the group action on set $\mathbb{X}$ for a fixed $g$, i.e. $T_{\mathbb{X},g}(x) = g \bullet_{\mathbb{X}} x$. Then, the smoothed model $f(x) = \xi(\mu_x \circ h^{-1})$ is equivariant with respect to the action of group $\mathbb{G}$ if both*

- *the family of measures $(\mu_x)_{x \in \mathbb{X}}$ is equivariant, i.e. $\forall x \in \mathbb{X}, g \in \mathbb{G} : \mu_{g \bullet x} = \mu_x \circ T_{\mathbb{X},g}^{-1}$,*

- *and smoothing scheme $\xi$ is equivariant, i.e. $\forall \nu \in \Delta(\mathbb{V}, \mathbb{F}), g \in \mathbb{G} : \xi(\nu \circ T_{\mathbb{V},g}^{-1}) = g \bullet \xi(\nu)$.*

*Proof.* By definition of $f$, verifying that $\forall x \in \mathbb{X}, \forall g \in \mathbb{G} : f(g \bullet x) = g \bullet f(x)$ is equivalent to verifying that $\xi(\mu_{g \bullet x} \circ h^{-1}) = g \bullet \xi(\mu_g \circ h^{-1})$. We can do so by first using the equivariance of the family of measures $(\mu_x)_{x \in \mathbb{X}}$, then the equivariance of base model $h$ and then the equivariance of

smoothing scheme $\xi$:

$$
\begin{aligned}
\xi(\mu_{g\bullet x} \circ h^{-1}) &= \xi\left(\left(\mu_x \circ T_{\mathbb{X},g}^{-1}\right) \circ h^{-1}\right) \\
&= \xi\left(\mu_x \circ \left(T_{\mathbb{X},g}^{-1} \circ h^{-1}\right)\right) \\
&= \xi\left(\mu_x \circ (h \circ T_{\mathbb{X},g})^{-1}\right) \\
&= \xi\left(\mu_x \circ (T_{\mathbb{V},g} \circ h)^{-1}\right) \\
&= \xi\left(\mu_x \circ \left(h^{-1} \circ T_{\mathbb{V},g}^{-1}\right)\right) \\
&= \xi\left(\left(\mu_x \circ h^{-1}\right) \circ T_{\mathbb{V},g}^{-1}\right) = g \bullet \xi(\mu_x \circ h^{-1}).
\end{aligned}
$$

In the second and sixth equality we used the associativity of function composition. In the third and fifth equality we used that the preimage of a composition of functions is equivalent to the composition of the individual preimages in reverse order. $\qquad\square$

## E.2 Equivariance-preserving schemes and measures

In the following, $\Delta(\mathbb{X}, \mathbb{D})$ refers to the set of all probability measures on measurable space $(\mathbb{X}, \mathbb{D})$.

### E.2.1 Componentwise smoothing schemes.

For this type of smoothing scheme, we assume that the intermediate space and output space consist of $M$ distinct components, i.e. $\mathbb{V} = \mathbb{A}^M$ and $\mathbb{Y} = \mathbb{B}^M$ for some sets $\mathbb{A}, \mathbb{B}$. We write $\mathrm{proj}_m : \mathbb{A}^M \to \mathbb{A}$ for the function that maps any vector $\boldsymbol{a} \in \mathbb{A}^M$ to its $m$th component $a_m$. Thus, for any measure $\nu \in \Delta(\mathbb{A}^M, \mathbb{F}^M)$ the function $\nu \circ \mathrm{proj}_m^{-1}$ is the $m$th marginal measure, which is a measure on the single-dimensional space $(\mathbb{A}, \mathbb{F})$. Componentwise smoothing means generating the $m$th prediction based on some quantity of the $m$th marginal measure.

**Proposition 4.** *Assume an $M$-dimensional measurable product space $(\mathbb{A}^M, \mathbb{F}^M)$ and an output space $\mathbb{B}^M$. Further assume that there is a function $\kappa : \Delta(\mathbb{A}, \mathbb{F}) \to \mathbb{B}$ such that $\forall \nu \in \Delta(\mathbb{A}^M, \mathbb{F}^M)$ : $\xi(\nu)_m = \kappa(\nu \circ \mathrm{proj}_m^{-1})$. If a group $\mathbb{G}$ acts on $\mathbb{A}^N$ and $\mathbb{B}^N$ via the same permutation, then the smoothing scheme $\xi$ is equivariant.*

*Proof.* Consider an arbitrary $g \in \mathbb{G}$ with corresponding actions $T_{\mathbb{V},g} : \mathbb{A}^M \to \mathbb{A}^M$ with $T_{\mathbb{V},g}(\boldsymbol{v})_m = v_{\pi^{-1}(m)}$ and $T_{\mathbb{Y},g} : \mathbb{A}^M \to \mathbb{A}^M$ with $T_{\mathbb{Y},g}(\boldsymbol{y})_m = y_{\pi^{-1}(m)}$, where $\pi : \{1, \ldots, M\} \to \{1, \ldots, M\}$ is a permutation. For any $m$, we can use the definition $\xi$, the associativity of function composition and the fact that the preimage of a composition is equivalent to the composition of preimages in reverse order to show that

$$
\begin{aligned}
\xi\left(\nu \circ T_{\mathbb{V},g}^{-1}\right)_m &= \kappa\left((\nu \circ T_{\mathbb{V},g}^{-1}) \circ \mathrm{proj}_m^{-1}\right) \\
&= \kappa\left(\nu \circ (T_{\mathbb{V},g}^{-1} \circ \mathrm{proj}_m^{-1})\right) \\
&= \kappa\left(\nu \circ (\mathrm{proj}_m \circ T_{\mathbb{V},g})^{-1}\right) \\
&= \kappa\left(\nu \circ \mathrm{proj}_{\pi^{-1}(m)}^{-1}\right) = \xi(\nu)_{\pi^{-1}(m)} = (g \bullet \xi(\nu))_m.
\end{aligned}
$$

For the fourth equality we used that selecting the $m$th element from a sequence permuted by $\pi$ is equivalent to selecting the element $\pi^{-1}(m)$ of the unpermuted sequence. The second to last equality is just the definition of our smoothing scheme $\xi$. $\qquad\square$

In practice, componentwise smoothing schemes are not evaluated exactly but approximated using $S$ Monte Carlo samples $\boldsymbol{v}^{(1)}, \ldots, \boldsymbol{v}^{(S)} \in \mathbb{V}$. The $m$th component of smoothed prediction $y_n$ is generated based on some quantity of $v_m^{(1)}, \ldots, v_m^{(S)} \in \mathbb{V}$, such as their mean [75]. We can thus use the same argument as above to show that the Monte Carlo approximations are permutation equivariant: Permuting the components of the samples before applying componentwise functions is equivalent to first applying componentwise functions and then performing a permutation.

### E.2.2 Expected value smoothing scheme.

For this smoothing scheme, we assume that the intermediate and output space coincide are real-valued, i.e. $\mathbb{V} = \mathbb{Y} = \mathbb{R}^M$ We then make smoothed predictions via the expected value of the model's output distribution over $\mathbb{Y}$.

**Proposition 5.** *Let $\mathcal{B}(\mathbb{R}^M)$ be the corresponding Borel $\sigma$-algebra of $\mathbb{R}^M$. Define expected value smoothing scheme $\xi : \Delta(\mathbb{R}^M, \mathcal{B}(\mathbb{R}^M)) \to \mathbb{R}^M$ with*

$$\xi(\nu) = \int_{\mathbb{R}^M} \boldsymbol{y} \, d\nu(\boldsymbol{y})$$

*If group $\mathbb{G}$ acts on $\mathbb{R}^M$ via affine transformations, then smoothing scheme $\xi$ is equivariant.*

*Proof.* Consider an arbitrary $g \in \mathbb{G}$ with corresponding action $T_{\mathbb{Y},g}$. By change of variables and linearity of integration, we have

$$\begin{aligned}
\xi\left(\nu \circ T_{\mathbb{Y},g}^{-1}\right) &= \int_{\mathbb{R}^M} \boldsymbol{y} \, d\left(\nu \circ T_{\mathbb{Y},g}^{-1}\right)(\boldsymbol{y}) \\
&= \int_{\mathbb{R}^M} T_{\mathbb{Y},g}(\boldsymbol{y}) \, d\nu(y) \\
&= T_{\mathbb{Y},g}\left(\int_{\mathbb{R}^M} \boldsymbol{y} \, d\nu(\boldsymbol{y})\right) \\
&= g \bullet \xi(\nu).
\end{aligned}$$

$\square$

In practice, expected value smoothing is not evaluated exactly but approximated using the average of $S$ Monte Carlo samples $\boldsymbol{v}^{(1)}, \ldots, \boldsymbol{v}^{(S)} \in \mathbb{R}^M$ [75]. We can thus use the same argument as above to show that the Monte Carlo approximation is equivariant to affine transformations: Affinely transforming the samples before computing the average is equivalent to computing the average and then applying the affine transformation.

### E.2.3 Median smoothing scheme

For this smoothing scheme, we assume that the intermediate and output space coincide and are real-valued, i.e. $\mathbb{V} = \mathbb{Y} = \mathbb{R}^M$. We then make smoothed predictions via the elementwise median of the model's output distribution over $\mathbb{Y}$. As before, we write $\mathrm{proj}_m : \mathbb{R}^M \to \mathbb{R}$ for the function that maps any vector $\boldsymbol{y} \in \mathbb{R}^M$ to its $m$th component $y_m$. Thus, $\nu \circ \mathrm{proj}_m^{-1}$ is the $m$th marginal measure of $\nu$.

**Proposition 6.** *Let $\mathcal{B}(\mathbb{R}^M)$ be the Borel $\sigma$-algebra of $\mathbb{R}^M$. Let $F_m(\nu, x)$ be the cumulative distribution function for marginal measure $\nu \circ \mathrm{proj}_m^{-1}$, i.e.*

$$F_m(\nu, x) = \left(\nu \circ \mathrm{proj}_m^{-1}\right)((-\infty, x]).$$

*Let $F_m^-(\nu, p)$ and $F_m^+(\nu, p)$ be the corresponding lower and upper quantile functions, i.e.*

$$\begin{aligned}
F_m^-(\nu, p) &= \inf\{x \mid F_m(\nu, x) \geq p\} \\
F_m^+(\nu, p) &= \sup\{x \mid F_m(\nu, x) \leq p\}
\end{aligned}$$

*Define median smoothing scheme $\xi : \Delta(\mathbb{R}^M, \mathcal{B}(\mathbb{R}^M)) \to \mathbb{R}^M$ with*

$$\xi(\nu)_m = \frac{1}{2} \cdot \left[F_m^-(\nu, 1/2) + F_m^+(\nu, 1/2)\right].$$

*If group $\mathbb{G}$ acts on $\mathbb{Y}$ via elementwise linear transformations, then smoothing scheme $\xi$ is equivariant.*

*Proof.* Consider an arbitrary $g \in \mathbb{G}$. Let $T_{\mathbb{Y},g}$ be the corresponding group action with $T_{\mathbb{Y},g}(\boldsymbol{y})_m = w_m \cdot y_m + c_m$ for some $w_m, c_m \in \mathbb{R}$. Assume w.l.o.g. that $\forall m : w_m > 0$.

For the cumulative distribution functions, we have

$$F_m(\nu \circ T_{\mathbb{Y},g}^{-1}, x) = F_m(\nu, (x - c_m) / w_m).$$

For the corresponding lower and upper quantile functions, we thus have

$$F_m^-(\nu \circ T_{\mathbb{Y},g}^{-1}, p) = w_m \cdot F_m^-(\nu, p) + c_m$$
$$F_m^+(\nu \circ T_{\mathbb{Y},g}^{-1}, p) = w_m \cdot F_m^+(\nu, p) + c_m.$$

For the smoothing scheme, we thus have

$$\xi(\nu \circ T_{\mathbb{Y},g}^{-1})_m = w_m \cdot \xi(\nu)_m + c_m = (g \bullet \xi(\nu))_m.$$

$\square$

In practice, median smoothing is not evaluated exactly but approximated using the elementwise median of $S$ Monte Carlo samples $\boldsymbol{v}^{(1)}, \dots, \boldsymbol{v}^{(S)} \in \mathbb{V}$ [75]. We can thus use the same argument as above to show that the Monte Carlo approximation is equivariant to elementwise linear transformations: Linearly transforming the samples elementwise before computing the sample median is equivalent to computing the sample median and then applying the elementwise linear transformations.

### E.2.4 Center smoothing scheme

For this type of smoothing scheme, we assume that the intermediate and output space are identical, i.e. $\mathbb{V} = \mathbb{Y}$. Let $\mathbb{B}(y, r) = \{y' \in \mathbb{Y} \mid d_{\text{out}}(y, y') \leq r\}$ be the $d_{\text{out}}$ ball of radius $r$ around $y \in \mathbb{Y}$. Center smoothing makes its predictions using the center of the $d_{\text{out}}$ ball with the smallest radius among all $d_{\text{out}}$ balls with a measure of at least $\frac{1}{2}$.

**Proposition 7.** *Consider a measurable space* $(\mathbb{Y}, \mathbb{F})$*, a function* $d_{\text{out}} : \mathbb{Y} \times \mathbb{Y} \to \mathbb{R}_+$*, and assume that* $\forall y \in \mathbb{Y}, r \geq 0 : \mathbb{B}(y, r) \in \mathbb{F}$*. Define center smoothing scheme* $\xi : \Delta(\mathbb{Y}, \mathbb{F}) \to \mathbb{Y}$ *with* $\xi(\nu) = \arg\min_y r$ *s.t.* $\nu(\mathbb{B}(y, r)) \geq \frac{1}{2}$*. If* $\mathbb{G}$ *acts isometrically on* $\mathbb{Y}$*, i.e.* $\forall y, y' \in \mathbb{Y}, g \in \mathbb{G} : d_{\text{out}}(g \bullet y, g \bullet y') = d_{\text{out}}(y, y')$*, then smoothing scheme* $\xi$ *is equivariant.*

*Proof.* Consider an arbitrary $g \in \mathbb{G}$ with corresponding action $T_{\mathbb{Y},g}$. By definition of the center smoothing scheme, $\xi(\nu \circ T_{\mathbb{Y},g}^{-1})$ is

$$\left(\arg\min_y r \text{ s.t. } \left(\nu \circ T_{\mathbb{Y},g}^{-1}\right)(\mathbb{B}(y, r)) \geq \frac{1}{2}\right) = \left(\arg\min_y r \text{ s.t. } \nu\left(T_{\mathbb{Y},g}^{-1}(\mathbb{B}(y, r)) \geq \frac{1}{2})\right)\right).$$

By definition of the preimage and action $T_{\mathbb{Y},g}$, we have $T_{\mathbb{Y},g}^{-1}(\mathbb{B}(y, r)) = \{y' \mid g \bullet y' \in \mathbb{B}(y, r)\} = \{y' \mid d_{\text{out}}(y, g \bullet y') \leq r\} = \{y' \mid d_{\text{out}}(g^{-1} \bullet y, y') \leq r\} = \mathbb{B}\left(g^{-1} \bullet y, r\right)$, where the second to last equality holds because $\mathbb{G}$ acts isometrically. This shows that

$$\xi(\nu \circ T_{\mathbb{Y},g}^{-1}) = \left(\arg\min_y r \text{ s.t. } \nu\left(\mathbb{B}(g^{-1} \bullet y, r)\right) \geq \frac{1}{2}\right).$$

The optimum of this problem is $g \bullet \xi(\nu)$, because $g^{-1} \bullet (g \bullet \xi(\nu)) = \xi(\nu)$ and $\xi(\nu)$ is the optimum of the original problem without group action. $\square$

In practice, center smoothing is approximated using $S$ Monte Carlo samples $\boldsymbol{y}^{(1)}, \dots, \boldsymbol{y}^{(S)} \in \mathbb{Y}$ by selecting the sample with the smallest median distance $d_{\text{out}}$ to all other samples [93]. We can use a similar argument to the one above to show that the Monte Carlo approximation is isometry equivariant: Isometries, by definition, do not change the pairwise distances and therefore do not change which sample has the smallest median distance to all other samples.

### E.2.5 Product measures

For our discussion of product measures, note that the $\sigma$-algebra $\mathbb{D}^N$ of a measurable product space $(\mathbb{A}^N, \mathbb{D}^N)$ is not the $n$-fold Cartesian product of the $\sigma$-algebra $\mathbb{D}$. Instead, it is defined as $\left\{\left(\times_{n=1}^N \mathbb{S}_n\right) \mid \mathbb{S}_1 \in \mathbb{D}, \dots, \mathbb{S}_N \in \mathbb{D}\right\}$, i.e. the set containing all Cartesian products of measurable sets.

**Proposition 8.** *Assume that* $(\mu_{\boldsymbol{x}})_{\boldsymbol{x} \in \mathbb{A}^N}$ *is a family of product measures on the N-dimensional measurable product space* $(\mathbb{A}^N, \mathbb{D}^N)$*, i.e. there is a family of measures* $(\kappa_a)_{a \in \mathbb{A}}$ *on* $(\mathbb{A}, \mathbb{D})$ *such that* $\forall \boldsymbol{x} \in \mathbb{A}^N, \forall \left(\times_{n=1}^N \mathbb{S}_n\right) \in \mathbb{D}^N : \mu_{\boldsymbol{x}}\left(\times_{n=1}^N \mathbb{S}_n\right) = \prod_{n=1}^N \kappa_{x_n}(\mathbb{S}_n)$*. If group* $\mathbb{G}$ *acts on* $\mathbb{A}^N$ *via permutation, then the family of measures* $(\mu_{\boldsymbol{x}})_{\boldsymbol{x} \in \mathbb{X}^N}$ *is equivariant.*

*Proof.* Consider an arbitrary $g \in \mathbb{G}$ with action $T_{\mathbb{X},g} : \mathbb{A}^N \to \mathbb{A}^N$ with $T_{\mathbb{V},g}(\boldsymbol{x})_n = x_{\pi^{-1}(n)}$, where $\pi : \{1, \dots, N\} \to \{1, \dots, N\}$ is a permutation. For any $\left( \bigtimes_{n=1}^{N} \mathbb{S}_n \right) \in \mathbb{D}^N$, we have

$$\mu_{g \bullet \boldsymbol{x}} \left( \bigtimes_{n=1}^{N} \mathbb{S}_n \right) = \prod_{n=1}^{N} \kappa_{(g \bullet \boldsymbol{x})_n}(\mathbb{S}_n) = \prod_{n=1}^{N} \kappa_{x_{\pi^{-1}(n)}}(\mathbb{S}_n) = \prod_{n=1}^{N} \kappa_{x_n}(\mathbb{S}_{\pi(n)}) = \mu_{\boldsymbol{x}} \left( \bigtimes_{n=1}^{N} \mathbb{S}_{\pi(n)} \right).$$

For the second to last equality, we have just changed the iteration order of the product from $(1, \dots, N)$ to $(\pi(1), \dots, \pi(N))$. Finally, it follows from the definition of our group action and preimages that

$$\left( \bigtimes_{n=1}^{N} \mathbb{S}_{\pi(n)} \right) = \left\{ \boldsymbol{s} \in \mathbb{A}^N \mid s_1 \in \mathbb{S}_{\pi(1)}, \dots, s_N \in \mathbb{S}_{\pi(N)} \right\}$$

$$= \left\{ \boldsymbol{s} \in \mathbb{A}^N \mid s_{\pi^{-1}(1)} \in \mathbb{S}_1, \dots, s_{\pi^{-1}(N)} \in \mathbb{S}_{\pi(N)} \right\}$$

$$= \left\{ \boldsymbol{s} \in \mathbb{A}^N \mid T_{\mathbb{X},g}(\boldsymbol{s}) \in \left( \bigtimes_{n=1}^{N} \mathbb{S}_n \right) \right\}$$

$$= T_{\mathbb{X},g}^{-1} \left( \bigtimes_{n=1}^{N} \mathbb{S}_n \right),$$

and thus $\forall \boldsymbol{x} \in \mathbb{A}^N, g \in \mathbb{G} : \mu_{g \bullet \boldsymbol{x}} = \mu_{\boldsymbol{x}} \circ T_{\mathbb{X},g}^{-1}$. $\qquad\square$

### E.2.6 Isotropic Gaussian measures

**Proposition 9.** *Consider the measurable space $(\mathbb{R}^{N \times D}, \mathcal{B}(\mathbb{R}^{N \times D}))$, where $\mathcal{B}(\mathbb{R}^{N \times D})$ is the Borel $\sigma$-algebra on $\mathbb{R}^{N \times D}$. Let $(\mu_{\boldsymbol{X},\sigma})_{\boldsymbol{X} \in \mathbb{R}^{N \times D}}$ be the family of isotropic Gaussian measures with standard deviation $\sigma$ on $(\mathbb{R}^{N \times D}, \mathcal{B}(\mathbb{R}^{N \times D}))$. If a group $\mathbb{G}$ acts isometrically on $\mathbb{R}^{N \times D}$ with respect to Frobenius norm $|| \cdot ||_2$, then the family of measures is equivariant.*

*Proof.* By definition of the Gaussian measure, we have for any $\mathbb{A} \in \mathcal{B}(\mathbb{R}^{N \times D})$

$$\mu_{\boldsymbol{X},\sigma}(\mathbb{A}) = \int_{\mathbb{A}} \prod_{n=1}^{N} \prod_{d=1}^{D} \frac{1}{\sqrt{2\pi\sigma^2}} \exp\left( -\frac{1}{2\sigma^2} (Z_{n,d} - X_{n,d}^2) \right) d\boldsymbol{Z}$$

$$= \int_{\mathbb{A}} \frac{1}{\left( \sqrt{2\pi\sigma^2} \right)^{N \cdot D}} \exp\left( -\frac{1}{2\sigma^2} ||\boldsymbol{Z} - \boldsymbol{X}||_2^2 \right) d\boldsymbol{Z}.$$

By change of variables, we have

$$\left( \mu_{\boldsymbol{X},\sigma} \circ T_{\mathbb{X},g}^{-1} \right)(\mathbb{A}) = \int_{T_{\mathbb{X},g}^{-1}(\mathbb{A})} \frac{1}{\left( \sqrt{2\pi\sigma^2} \right)^{N \cdot D}} \exp\left( -\frac{1}{2\sigma^2} ||\boldsymbol{Z} - \boldsymbol{X}||_2^2 \right) d\boldsymbol{Z}$$

$$= \int_{\mathbb{A}} \left| \det\left( J_{T_{\mathbb{X},g}^{-1}} \right) \right| \frac{1}{\left( \sqrt{2\pi}\sigma \right)^{N \cdot D}} \exp\left( -\frac{1}{2\sigma^2} ||T_{\mathbb{X},g}^{-1}(\boldsymbol{Z}) - \boldsymbol{X}||_2^2 \right) d\boldsymbol{Z}$$

$$= \int_{\mathbb{A}} 1 \cdot \frac{1}{\left( \sqrt{2\pi}\sigma \right)^{N \cdot D}} \exp\left( -\frac{1}{2\sigma^2} ||\boldsymbol{Z} - T_{\mathbb{X},g}(\boldsymbol{X})||_2^2 \right) d\boldsymbol{Z}$$

$$= \mu_{g \bullet \boldsymbol{X},\sigma}(\mathbb{A}).$$

The third equality follows from $T_{\mathbb{X},g}$ being an isometry with respect to the Frobenius norm $|| \cdot ||_2$. $\qquad\square$

### E.2.7 Transformation-specific measures

In transformation-specific smoothing [46, 60, 63–65], the clean input $x \in \mathbb{X}$ is transformed via randomly sampled functions from a parametric family $(\psi_\theta)_{\theta \in \Theta}$ with $\psi_\theta : \mathbb{X} \to \mathbb{X}$ and measurable parameter space $(\Theta, \mathbb{G})$. Let $\gamma : \mathbb{G} \to \mathbb{R}_+$ be the corresponding parameter distribution. Then, transformation-specific smoothing induces a smoothing measure

$$\mu_x(\mathbb{A}) = \gamma \left( \{ \theta \in \Theta \mid f_\theta(x) \in \mathbb{A} \} \right) \tag{3}$$

on input space $(\mathbb{X}, \mathbb{D})$. [4] In the following, we show that this induced measure inherits its equivariances from the family of transformations $(\psi_\theta)_{\theta \in \Theta}$.

**Proposition 10.** *Consider a measurable space $(\mathbb{X}, \mathbb{D})$ and a parametric family of transformations $(\psi_\theta)_{\theta \in \Theta}$ with $\psi_\theta : \mathbb{X} \to \mathbb{X}$ and measurable parameter space $(\Theta, \mathbb{G})$. Let $\gamma : \mathbb{G} \to \mathbb{R}_+$ be a measure on parameter space $(\Theta, \mathbb{G})$ and consider the family of induced smoothing measures $(\mu_x)_{x \in \mathbb{X}}$, as defined in Eq. (3). If all transformations $\psi_\theta$ are equivariant to the actions of group $\mathbb{G}$, then the family of measures is equivariant.*

*Proof.* Consider an arbitrary group element $g \in \mathbb{G}$ with corresponding group action $T_{\mathbb{X}, g}$. By definition of $\mu_x$ and the equivariance of all $\psi_\theta$, we have for all $x \in \mathbb{X}$ that

$$
\begin{aligned}
\mu_{g \bullet x}(\mathbb{A}) &= \gamma\left(\{\theta \in \Theta \mid f_\theta(g \bullet x) \in \mathbb{A}\}\right) \\
&= \gamma\left(\{\theta \in \Theta \mid g \bullet f_\theta(x) \in \mathbb{A}\}\right) \\
&= \gamma\left(\left\{\theta \in \Theta \mid f_\theta(x) \in T_{\mathbb{X},g}^{-1}(\mathbb{A})\right\}\right) = \left(\mu_x \circ T_{\mathbb{X},g}^{-1}\right)(\mathbb{A}).
\end{aligned}
$$

The second-to-last equality holds because, by definition of the pre-image, $T_{\mathbb{X},g}^{-1}(\mathbb{A}) = \{x \in \mathbb{X} \mid g \bullet x \in \mathbb{A}\}$ and thus $g \bullet f_\theta(x) \in \mathbb{A} \iff f_\theta(x) \in T_{\mathbb{X},g}^{-1}(\mathbb{A})$. $\qquad\square$

### E.2.8 Sparsity-aware measures

**Proposition 11.** *Consider the measurable space $(\mathbb{X}, \mathbb{D})$, where $\mathbb{X} = \{0,1\}^{N \times D} \times \{0,1\}^{N \times N}$ is the set of all binary attributed graphs with $N$ nodes and $D$ features and $\mathbb{D} = \mathcal{P}(\mathbb{X})$ is its powerset. Let $(\mu_{\boldsymbol{X}, \boldsymbol{A}})_{(\boldsymbol{X}, \boldsymbol{A}) \in \mathbb{X}}$ be the family of sparsity-aware aware measures on $(\mathbb{X}, \mathbb{D})$, as defined by the sparsity-aware probability mass function with fixed flip probabilities $p_+^{\boldsymbol{X}}, p_-^{\boldsymbol{X}}, p_+^{\boldsymbol{A}}, p_-^{\boldsymbol{A}} \in [0,1]$ (see Eq. (18)). If group $\mathbb{G}$ acts on $\mathbb{X}$ via graph isomorphisms, then the family of measures is equivariant.*

*Proof.* Group $\mathbb{G}$ acting on $\mathbb{X}$ via graph isomorphisms means that for every $(\boldsymbol{X}, \boldsymbol{A}) \in \mathbb{X}$ and $g \in \mathbb{G}$ there is some permutation matrix $\boldsymbol{P} \in \{0,1\}^{N \times N}$ such that $g \bullet (\boldsymbol{X}, \boldsymbol{A}) = (\boldsymbol{P}\boldsymbol{X}, \boldsymbol{P}\boldsymbol{A}\boldsymbol{P}^T)$. That is, the entries of the attribute and adjacency matrix are permuted. The sparsity-aware measures are product measures, i.e. sparsity-aware smoothing perturbs each element of the attribute and adjacency matrix independently. Families of product measures are equivariant with respect to the action of groups acting via permutation, see Proposition 8. $\qquad\square$

Table 1: Suitable measures for different input domains $\mathbb{X}$, input distances $d_{\text{in}}$, and group actions.

| $\mathbb{X}$ | $d_{\text{in}}$ | Group action | Smoothing measure |
|:---:|:---:|:---:|:---:|
| $\mathbb{R}^{N \times D}$ | $\ell_1, \ell_\infty$ | Permutation, Translation | $\text{Uniform}(-\sigma, \sigma)$ |
| $\mathbb{R}^{N \times D}$ | $\ell_1, \ell_\infty$ | Permutation, Translation | $\text{Laplace}(0, \sigma)$ |
| $\mathbb{R}^{N \times D}$ | $\ell_2$ | Euclidean isometries | $\mathcal{N}(0, \sigma)$ |
| $\mathbb{Z}^{N \times D}$ | $\ell_0$ | Permutation | Discrete smoothing (see Lee et al. [80]) |
| $\{0,1\}^{N \times D} \times \{0,1\}^{N \times N}$ (Attributed graphs) | Edit cost (see Eq. (2)) | Graph isomorphism | Sparsity-aware smoothing (see Appendix E.7) |

---

[4]Assuming $\forall \mathbb{A} \in \mathbb{D} : \{\theta \mid f_\theta(x) \in \mathbb{A}\} \in \mathbb{G}$, i.e. all preimages are in the parameter $\sigma$-field $\mathbb{G}$.

Table 2: Suitable smoothing schemes for different output domains $\mathbb{Y}$, output distances $d_{\text{out}}$, and group actions (see discussion in Section 5.1). (*)For expected value and median smoothing, robustness can be certified as long as the $d_{\text{out}}$ ball $\{y' \mid d_{\text{out}}(f(x), y') \leq \epsilon\}$ is contained within a hyperrectangle (see Appendix D.2).

| $\mathbb{Y}$ | $d_{\text{out}}$ | Group action | Smoothing scheme |
|---|---|---|---|
| $\{1, \ldots, K\}^N$ | $\ell_0$ | Permutation | Elementwise majority voting |
| $\mathbb{R}^N$ | Any* | Permutation, Affine | Expected value |
| $\mathbb{R}^N$ | Any* | Permutation, Elementwise linear | Elementwise median |
| Any | Any | Isometry w.r.t. $d_{\text{out}}$ (e.g. Euclidean isometries for $\ell_2$) | Center |

# F   Graph edit distance certificates

In the following, we show how the different existing approaches for proving the robustness of graph neural networks operating on $\{0, 1\}^{N \times D} \times \{0, 1\}^{N \times N}$ w.r.t to distance $d_{\text{in}}$

$$c_{\boldsymbol{X}}^+ \cdot ||(\boldsymbol{X}' - \boldsymbol{X})_+||_0 + c_{\boldsymbol{X}}^- \cdot ||(\boldsymbol{X}' - \boldsymbol{X})_-||_0 + c_{\boldsymbol{A}}^+ \cdot ||(\boldsymbol{A}' - \boldsymbol{A})_+||_0 + c_{\boldsymbol{A}}^- \cdot ||(\boldsymbol{A}' - \boldsymbol{A})_-||_0, \quad (4)$$

with costs $c_{\boldsymbol{X}}^+, c_{\boldsymbol{X}}^-, c_{\boldsymbol{A}}^+, c_{\boldsymbol{A}}^- \in \{\infty, 1\}$ can be generalized to non-uniform costs, i.e. $c_{\boldsymbol{X}}^+, c_{\boldsymbol{X}}^-, c_{\boldsymbol{A}}^+, c_{\boldsymbol{A}}^- \in \{\infty\} \cup \mathbb{R}_+$. Combined with Proposition 2, this yields robustness guarantees w.r.t. to the graph edit distance, where the edit operations are insertion and deletion of edges and/or node attributes. Note that we also consider local budgets $\rho_1, \dots, \rho_N$, which we introduce in Appendix G.

Before making the different generalizations, we discuss how to solve Knapsack problems with local constraints, which will be used in many of the subsequent derivations.

Please note that providing an in-depth explanation and motivation for each of the certification procedures would be out of scope. Instead, we strongly recommend first reading the original papers. Our discussions are to be understood as specifying changes that have to be made relative to the original certification procedures.

## F.1   Knapsack problems with local constraints

A recurring problem in proving the robustness of graph neural networks is selecting a set of edges or attributes that have the largest effect on a classifier's prediction while complying with global and node-wise constraints on the cost of perturbations. This is a form of Knapsack problem.

**Definition 2.** *A knapsack problem with local constraints is a binary integer program of the form*

$$\max_{\boldsymbol{Q} \in \{0,1\}^{N \times M}} \sum_{n=1}^{N} \sum_{m=1}^{M} V_{n,m} \cdot Q_{n,m}$$

$$\text{s.t.} \sum_{n=1}^{N} \sum_{m=1}^{M} C_{n,m} \cdot Q_{n,m} \le \epsilon, \quad \forall n : \sum_{m=1}^{M} C_{n,m} \cdot Q_{n,m} \le \rho_n, \quad (5)$$

*with value matrix $\boldsymbol{V} \in \mathbb{R}^{N \times M}$, cost matrix $\boldsymbol{C} \in \mathbb{R}_+^{N \times M}$, global budget $\epsilon \in \mathbb{R}_+$ and local budgets $\rho_1, \dots, \rho_N \in \mathbb{R}_+$.*

Matrix $\boldsymbol{Q}$ indicates which entry of an attribute or adjacency matrix should be adversarially perturbed.

When there are only two distinct costs, i.e. we are only concerned with attribute or adjacency perturbations, the above problem can be solved exactly via dynamic programming. Alternatively, an upper bound can be found via linear relaxation, i.e. optimization over $[0, 1]^{N \times M}$.

### F.1.1   Dynamic programming

In the following, we assume that there is an index set $\mathbb{I} \subseteq \{1, \dots, N\} \times \{1, \dots, M\}$ with complement $\bar{\mathbb{I}}$ such that $\forall (n, m) \in \mathbb{I} : C_{n,m} = c^+$ and $\forall (n, m) \in \bar{\mathbb{I}} : C_{n,m} = c^-$, i.e. we only have two distinct costs. In this case, the problem from Eq. (5) can be solved in a two-step procedure, which generalizes the certification procedure for uniform costs discussed in Section 3 of [15].

The first step is to find optimal solutions per row $r$, while ignoring the global budget $\epsilon$. We pre-compute $N$ dictionaries[5] $\alpha_1, \dots, \alpha_N : \mathbb{N} \times \mathbb{N} \to \mathbb{R}_+$ and $N$ dictionaries $\beta_1, \dots, \beta_N : N \times N \to \mathcal{P}(\{1, \dots, M\})$, where $\mathcal{P}$ is the powerset. Entry $\alpha_r(i, j)$ is the optimal value of Eq. (5) if only row $\boldsymbol{Q}_r$ is non-zero, and exactly $i$ indices in $\mathbb{I}$ and $j$ indices in $\bar{\mathbb{I}}$ are non-zero, i.e.

$$\alpha_r(i, j) = \max_{\boldsymbol{Q} \in \{0,1\}^{N \times M}} \sum_{n=1}^{N} \sum_{m=1}^{M} V_{n,m} \cdot Q_{n,m}$$

$$\text{s.t.} \sum_{n,m \in \mathbb{I}} \boldsymbol{Q}_{n,m} = i, \quad \sum_{n,m \in \bar{\mathbb{I}}} \boldsymbol{Q}_{n,m} = j, \quad \boldsymbol{Q}_{-r} = \boldsymbol{0}.$$

---

[5]or sparse matrices

Entry $\beta_r(i, j)$ is the corresponding set of non-zero indices of the optimal $\boldsymbol{Q}_r$. Because all entries of value matrix $\boldsymbol{V}$ and cost matrix $\boldsymbol{C}$ are non-negative, the optimal solution is to just select the indices with the largest value $V_{n,m}$. This pre-processing step is summarized in Algorithm 1. Its complexity is in $\mathcal{O}(N \cdot ((\max_n \rho_n) / c^+) \cdot ((\max_n \rho_n) / c^-))$, i.e. it scales linearly with the number of rows and the maximum number of non-zero values in a row of $\boldsymbol{Q}$.

The second step is to combine these local optimal solutions while complying with the global budget $\epsilon$ and local budgets $\rho_1, \dots, \rho_N$. This can be achieved via dynamic programming. We create dictionaries $s_1, \dots, s_N$, with $s_r(\gamma)$ being the optimal value of Eq. (5) when only allocating budget to the first $r$ rows and incurring an overall cost of $\gamma$:

$$s_r(\gamma) = \max_{\boldsymbol{Q} \in \{0,1\}^{N \times M}} \sum_{n=1}^{N} \sum_{m=1}^{M} V_{n,m} \cdot Q_{n,m}$$

$$\text{s.t.} \sum_{n,m \in \mathbb{I}} c^+ \cdot \boldsymbol{Q}_{n,m} + \sum_{n,m \in \bar{\mathbb{I}}} c^- \cdot \boldsymbol{Q}_{n,m} = \gamma, \quad \boldsymbol{Q}_{r:} = \boldsymbol{0}.$$

The first dictionary, $s_1$, can be generated from the precomputed optimal values $\alpha_1$. After that, each dictionary $s_r$ can be generated from $s_{r-1}$, while ensuring that we remain within local budget $\rho_n$ and global budget $\epsilon$. The optimal value is given by $\max_\gamma s_N(\gamma)$. In addition, we generate dictionaries $t_1, \dots, t_N$, which aggregate the row-wise optima stored in $\beta_1, \dots, \beta_N$ such that $t_N(\gamma^*)$ with $\gamma^* = \text{argmax}_\gamma s_N(\gamma)$ gives us the non-zero entries of the optimal $\boldsymbol{Q}$. This procedure is summarized in Algorithm 2. Its complexity is in $\mathcal{O}(N \cdot (\epsilon / c^+) \cdot (\epsilon / c^-) \cdot ((\max_n \rho_n) / c^+) \cdot ((\max_n \rho_n) / c^-))$, i.e. it scales linearly with the number of rows, the maximum number of non-zero values in $\boldsymbol{Q}$, and the maximum number of non-zero values in a row of $\boldsymbol{Q}$. Note that graph neural networks are generally brittle to small adversarial perturbations, so the algorithm only needs to be executed for small $\epsilon$.

---

**Algorithm 1** Precomputation of local solutions

---

    **for** $n = 1, \dots, N$ **do**
        $\alpha_n, \beta_n \leftarrow \text{dict}(), \text{dict}()$
        $\text{best\_idx} \leftarrow \text{argsort\_desc}(\boldsymbol{V}_n)$
        $\text{best\_add\_idx} \leftarrow \text{best\_idx} \setminus \bar{\mathbb{I}}$
        $\text{best\_del\_idx} \leftarrow \text{best\_idx} \setminus \mathbb{I}$
        $\text{max\_adds} \leftarrow \lfloor \rho_n / c^+ \rfloor$
        **for** $i = 0, \dots, \text{max\_adds}$ **do**
            $\text{max\_dels} \leftarrow \lfloor (\rho_n - c^+ \cdot i) / c^- \rfloor$
            **for** $j = 0, \dots, \text{max\_dels}$ **do**
                $\alpha_n(i, j) \leftarrow \text{sum}(\{\boldsymbol{V}_{n,k} \mid k \in \text{best\_add\_idx}[:i]\})$
                $\alpha_n(i, j) \leftarrow \alpha_n(i, j) + \text{sum}(\{\boldsymbol{V}_{n,k} \mid k \in \text{best\_del\_idx}[:j]\})$
                $\beta_n(i, j) \leftarrow \text{set}(\text{best\_add\_idx}[:i]) \cup \text{set}(\text{best\_del\_idx}[:j])$
    **return** $\alpha_1, \dots, \alpha_N, \beta_1, \dots, \beta_N$

---

**Algorithm 2** Construction of the global solution from local solutions

---

$s_0, t_0 \leftarrow \text{dict}(), \text{dict}()$
$s_0(0) \leftarrow 0$
$t_0(0) \leftarrow [\,]$
**for** $n = 1, \ldots, N$ **do**
    $s_n, t_n \leftarrow \text{dict}(), \text{dict}()$
    **for** $\text{prev\_cost} \in \text{keys}(s_{n-1})$ **do**
        $b \leftarrow \min(\rho_n, \epsilon - \text{prev\_cost})$
        $\text{max\_adds} \leftarrow \lfloor b \,/\, c^+ \rfloor$
        **for** $i = 0, \ldots, \text{max\_adds}$ **do**
            $\text{max\_dels} \leftarrow \lfloor (b - c^+ \cdot i) \,/\, c^- \rfloor$
            **for** $j = 0, \ldots, \text{max\_dels}$ **do**
                $\gamma \leftarrow \text{prev\_cost} + c^+ \cdot i + c^- \cdot j$
                $v \leftarrow s_{n-1}(\text{prev\_cost}) + \alpha_n(i, j)$
                **if** $\gamma \notin \text{keys}(s_n)$ **or** $v > s_n(\gamma)$ **then**
                    $s_n(\gamma) \leftarrow v$
                    $t_n(\gamma) \leftarrow \text{concatenate}(t_{n-1}(\text{prev\_cost}), \beta(i, j))$
$\gamma^* \leftarrow \text{argmax}_{\gamma \in \text{keys}(s_N)} s_N(\gamma)$
**return** $s_N(\gamma^*), t_N(\gamma^*)$

---

### F.1.2   Linear relaxation

The above dynamic programming approach can be generalized to arbitrary cost matrices $\boldsymbol{C} \in \mathbb{R}_+^{N \times M}$. However, it may not scale because it requires iterating over all possible combinations of costs admitted by global budget $\epsilon$ and local budgets $\rho_1, \ldots, \rho_N$. A more efficient alternative is upper-bounding the optimal value of the knapsack problem in Eq. (5) by relaxing the binary matrix $\boldsymbol{Q} \in \{0, 1\}^{N \times M}$ to real values $\boldsymbol{Q} \in [0, 1]^{N \times M}$:

$$\max_{\boldsymbol{Q} \in \mathbb{R}_{N \times M}} \sum_{n=1}^{N} \sum_{m=1}^{M} V_{n,m} \cdot Q_{n,m}$$

$$\text{s.t.} \sum_{n=1}^{N} \sum_{m=1}^{M} C_{n,m} \cdot Q_{n,m} \leq \epsilon, \quad \forall n : \sum_{m=1}^{M} C_{n,m} \cdot Q_{n,m} \leq \rho_n, \tag{6}$$

$$\forall n, m : 0 \leq Q_{n,m}, \quad \forall n, m : Q_{n,m} \leq 1.$$

Intuitively, it is best to set those $Q_{n,m}$ to 1 that have the largest value-to-cost ratio, i.e. the largest $V_{n,m} \,/\, C_{n,m}$. To comply with the local and global budgets, one should first greedily select the $(n, m)$ with the largest ratio in each row $n$, until the local budgets $\rho_n$ are exhausted. One should then aggregate all these row-wise optimal indices and select those with the largest ratio, until the global budget $\epsilon$ is exhausted. This procedure is summarized in Algorithm 3.

**Algorithm 3** Optimal value of the linearly relaxed knapsack problem with local constraints (Eq. (6))

$\boldsymbol{Q}^* \leftarrow \boldsymbol{0}$          ▷ Initialize global allocations with $N \times M$ zeros.
$\boldsymbol{L} \leftarrow \boldsymbol{0}$          ▷ Initialize row-wise allocations with $N \times M$ zeros.
$\boldsymbol{R} \leftarrow \boldsymbol{V} \oslash \boldsymbol{C}$          ▷ Elementwise division. Division by 0 is defined as $\infty$.
**for** $n = 1, \ldots, N$ **do**
     best_cols $\leftarrow$ argsort_desc$(\boldsymbol{R}_n)$
     $b \leftarrow \rho_n$
     **for** $m \in$ best_cols **do**
         $L_{n,m} \leftarrow b \, / \, C_{n,m}$
         $L_{n,m} \leftarrow \min(1, L_{n,m}))$          ▷ Clip, to avoid allocating values larger than 1.
         $b \leftarrow b - C_{n,m} \cdot L_{n,m}$
best_idx $\leftarrow$ argsort_desc$(\boldsymbol{R})$
$b \leftarrow \epsilon$
**for** $(n, m) \in$ best_idx **do**
     $Q^*_{n,m} \leftarrow b \, / \, C_{n,m}$
     $Q^*_{n,m} \leftarrow \min(L_{n,m}, Q^*_{n,m})$          ▷ Clip, to avoid violating local budget constraints.
     $b \leftarrow b - C_{n,m} \cdot Q^*_{n,m}$
**return** $\sum_{n,m} V_{n,m} \cdot Q^*_{n,m}$

**Proposition 12.** *Algorithm 3 yields the optimal value of the linearly relaxed knapsack problem with local constraints, as defined in Eq. (6).*

*Proof.* To simplify the proof, we make certain assumptions about budgets $\epsilon$ and $\rho_1, \ldots, \rho_N$. If $\epsilon = 0$, then the value of $\boldsymbol{Q}$ computed by Algorithm 3 is obviously correct, because the only feasible solution that does not violate the global budget is $\boldsymbol{Q}^* = \boldsymbol{0}$. If $\rho_n = 0$ for some $n$, then the value of $\boldsymbol{Q}_n$ computed by Algorithm 3 is obviously correct, because the only feasible solution that does not violate the local budget is $\boldsymbol{Q}^*_n = \boldsymbol{0}$. We thus assume w.l.o.g. that $\epsilon > 0$ and $\forall n : \rho_n > 0$. Furthermore, we assume that the budgets do not exceed the maximum overall and row-wise cost, i.e. $\epsilon \le \sum_{n=1}^N \sum_{m=1}^M C_{n,m}$ and $\forall n : \sum_{m=1}^M C_{n,m} \le \rho_n$, since any excess budget will not have any effect on the optimal value of Eq. (6).

In the following, we prove the correctness of Algorithm 3 by verifying that the constructed solution fulfills the Karush-Kuhn-Tucket conditions. We define the Lagrangian

$$
L(\boldsymbol{Q}, \lambda, \boldsymbol{\kappa}, \boldsymbol{T}, \boldsymbol{U}) = \left( - \sum_{n=1}^N \sum_{m=1}^M V_{n,m} \cdot Q_{n,m} + \lambda \left( \sum_{n=1}^N \sum_{m=1}^M C_{n,m} \cdot Q_{n,m} - \epsilon \right) \right.
$$
$$
\left. + \sum_{n=1}^N \kappa_n \left( \sum_{m=1}^M C_{n,m} \cdot Q_{n,m} - \rho_n \right) - \sum_{n=1}^N \sum_{m=1}^M T_{n,m} \cdot Q_{n,m} + \sum_{n=1}^N \sum_{m=1}^M U_{n,m} \cdot (Q_{n,m} - 1) \right).
$$

We introduce a negative sign in the objective, because Eq. (6) is a maximization problem. Variable $\lambda \in \mathbb{R}_+$ corresponds to the global budget constraint, variable $\boldsymbol{\kappa} \in \mathbb{R}_+^N$ correspond to the local budget constraints and variables $\boldsymbol{T}, \boldsymbol{U} \in \mathbb{R}_+^{N \times M}$ correspond to the constraint that $\boldsymbol{Q}$ should be in $[0, 1]^{N \times M}$.

We claim that $\boldsymbol{Q}$, $\lambda$, $\boldsymbol{\kappa}$, $\boldsymbol{T}$ and $\boldsymbol{U}$ have the following optimal values:

- The optimal value of $\boldsymbol{Q}$ is the $\boldsymbol{Q}^*$ computed by Algorithm 3.

- The optimal value of $\lambda$ is $\lambda^* = \min_{n,m} V_{n,m} \, / \, C_{n,m}$ s.t. $Q^*_{n,m} > 0$, with $\boldsymbol{Q}^*$ computed as in Algorithm 3. That is, $\lambda^*$ is the smallest value-to-cost ratio for which some global budget is allocated.

- Define $o_n = \min_m V_{n,m} \, / \, C_{n,m}$ s.t. $L_{n,m} > 0$ with $\boldsymbol{L}$ computed as in Algorithm 3. That is, $o_n$ is the smallest value-to-cost ratio in row $n$ for which some local budget is allocated. The optimal value of $\kappa_n$ is $\kappa^*_n = \max(0, o_n - \lambda^*)$.

- The optimal value of $U_{n,m}$ is $U^*_{n,m} = \max(V_{n,m} - C_{n,m} \cdot (\lambda^* + \kappa^*_n), 0)$.

- The optimal value of $T_{n,m}$ is $T_{n,m}^* = -V_{n,m} + C_{n,m} \cdot (\lambda^* + \kappa_n^*) + U_{n,m}$, which is equivalent to $\max(0, -V_{n,m} + C_{n,m} \cdot (\lambda^* + \kappa_n^*))$.

**Stationarity.** By the definition of $\boldsymbol{T}^*$, we trivially have

$$\nabla_{Q_{n,m}} L(\boldsymbol{Q}^*, \lambda^*, \boldsymbol{\kappa}^*, \boldsymbol{T}^*, \boldsymbol{U}^*) = -V_{n,m} + C_{n,m} \cdot (\lambda^* + \kappa_n^*) - T_{n,m}^* + U_{n,m}^* = 0.$$

**Primal feasibility.** The clipping in Algorithm 3 ensures that $\forall n, m : 0 \leq Q_{n,m}^* \leq 1$ and that global budget $\epsilon$ and local budgets $\rho_1, \ldots, \rho_N$ are never exceeded.

**Dual feasibility.** Because all values $\boldsymbol{V}$ and all costs $\boldsymbol{C}$ are non-negative, the optimal values $\lambda^*, \boldsymbol{\kappa}^*$ are non-negative. The optimal values $\boldsymbol{T}^*$ and $\boldsymbol{U}^*$ are non-negative by definition.

**Complementary slackness.** We first verify complementary slackness for variable $\lambda$. Because we assume that the global budget $\epsilon$ does not exceed the maximum overall cost $\sum_{n=1}^N \sum_{m=1}^M C_{n,m}$ and all values $\boldsymbol{V}$ are non-negative, Algorithm 3 will always allocate the entire global budget, i.e.

$$\lambda^* \left( \sum_{n=1}^N \sum_{m=1}^M C_{n,m} \cdot Q_{n,m}^* - \epsilon \right) = \lambda^* \cdot 0 = 0.$$

Next, we verify complementary slackness for variable $\boldsymbol{\kappa}$. We can make a case distinction, based on the value of $\kappa_n^* = \max(0, o_n - \lambda^*)$. If $o_n \leq \lambda^*$, we have

$$\kappa_n \cdot \left( \sum_{m=1}^M C_{n,m} \cdot Q_{n,m}^* - \rho_n \right) = 0 \cdot \left( \sum_{m=1}^M C_{n,m} \cdot Q_{n,m}^* - \rho_n \right) = 0.$$

If $o_n > \lambda^*$, then – by definition – the smallest value-to-cost ratio for which budget is allocated in row $n$ is larger than the smallest value-to-cost ratio for which budget is allocated overall. Combined with our assumption that the local budget $\rho_n$ does not exceed the overall cost $\sum_{m=1}^M C_{n,m}$ in row $n$, this means that the entire local budget $\rho_n$ is used up, i.e.

$$\kappa_n^* \cdot \left( \sum_{m=1}^M C_{n,m} \cdot Q_{n,m}^* - \rho_n \right) = \kappa_n^* \cdot 0 = 0.$$

Next, we verify complementary slackness for variable $\boldsymbol{T}$. If $Q_{n,m}^* = 0$, we obviously have $T_{n,m}^* \cdot (-Q_{n,m}^*) = 0$. For $Q_{n,m}^* > 0$, recall that $T_{n,m}^* = \max(0, -V_{n,m} + C_{n,m} \cdot (\lambda^* + \kappa_n^*))$. Because we defined $\kappa_n^* = \max(0, o_n - \lambda^*)$, we have $\lambda^* + \kappa_n^* = \max(\lambda^*, o_n)$. We can thus verify $V_{n,m} \geq C_{n,m} \cdot \lambda^*$ and $V_{n,m} \geq C_{n,m} \cdot o_n$ to show that $\max(0, -V_{n,m} + C_{n,m} \cdot (\lambda^* + \kappa_n^*)) = 0$.

We defined $\lambda^*$ to be the smallest value-to-cost ratio for which $Q_{n,m}^* > 0$. We thus know that $V_{n,m} \geq C_{n,m} \cdot \lambda^*$ whenever $Q_{n,m}^* > 0$.

We defined $o_n$ to be the largest value-to-cost ratio for which $L_{n,m} > 0$. Due to the clipping in Algorithm 3, we know that $L_{n,m} \geq Q_{n,m}^*$. We thus know that, if $Q_{n,m}^* > 0$, then $L_{n,m} > 0$ and thus $V_{n,m} \geq C_{n,m} \cdot o_n$.

Combining these two results confirms that $T_{n,m}^* = 0$ if $Q_{n,m}^* > 0$ and thus $T_{n,m}^* \cdot (-Q_{n,m}^*) = 0$.

Finally, we verify complementary slackness for variable $\boldsymbol{U}$ using a similar argument. If $Q_{n,m}^* = 1$, we trivially have $U_{n,m}^* \cdot (Q_{n,m}^* - 1) = 0$. For $Q_{n,m}^* < 1$, recall that $U_{n,m}^* = \max(V_{n,m} - C_{n,m} \cdot (\lambda + \kappa_n), 0)$. We can again use that $\lambda^* + \kappa_n^* = \max(\lambda^*, o_n)$. There are two only two potential causes for $Q_{n,m}^* < 1$. The first one is that all of the local budget $\rho_n$ was used up in column $m$ of row $n$ or columns with higher value-to-cost ratio. In this case, we have $V_{n,m} \leq C_{n,m} \cdot o_n$. The other one is that all of the global budget $\epsilon$ was used up in entry $(n, m)$ or entries with higher value-to-cost ratio. In this case, we have $V_{n,m} \leq C_{n,m} \cdot \lambda^*$. Overall, this implies that $U_{n,m}^* \cdot (Q_{n,m}^* - 1) = 0 \cdot (Q_{n,m}^* - 1) = 0$.

$\square$

### F.2 Interval bound propagation for attribute perturbations

Liu et al. [12] propose to prove the robustness of $L$-layer graph convolutional networks [98] to attribute perturbations with uniform cost ($c_{\boldsymbol{X}}^+ = c_{\boldsymbol{X}}^- = 1, c_{\boldsymbol{A}}^+ = c_{\boldsymbol{A}}^- = \infty$) via interval bound

propagation. In the following, we generalize their guarantees to arbitrary attribute perturbation costs $c_{\boldsymbol{X}}^+, c_{\boldsymbol{X}}^- \in \mathbb{R}_+$.

Graph convolutional networks applied to attributes $\boldsymbol{X} \in \{0,1\}^{N \times D}$ and adjacency $\boldsymbol{A} \in \{0,1\}^{N \times N}$ are defined as

$$
\begin{aligned}
\boldsymbol{H}^{(0)} &= \boldsymbol{X} \\
\boldsymbol{Z}^{(l)} &= \tilde{\boldsymbol{A}} \boldsymbol{H}^{(l)} \boldsymbol{W}^{(l)} + \mathbf{1}(\boldsymbol{b}^{(l)})^T \quad \text{for } l = 0, \ldots, L-1 \\
\boldsymbol{H}^{(l)} &= \sigma^{(l)}(\boldsymbol{Z}^{(l-1)}) \qquad\qquad\quad \text{for } l = 1, \ldots, L,
\end{aligned}
\tag{7}
$$

where $\boldsymbol{W}^{(0)}, \ldots, \boldsymbol{W}^{(L-1)}$ are weight matrices, $\boldsymbol{b}^{(0)}, \ldots, \boldsymbol{b}^{(L-1)}$ are bias vectors, $\sigma^{(1)}, \ldots, \sigma^{(L)}$ are activation functions (here assumed to be ReLU in the first $L-1$ layers and softmax in the last layer), and $\tilde{\boldsymbol{A}}$ is the adjacency matrix after additional preprocessing steps, such as degree normalization.

Given a perturbation set $\mathbb{B}$, interval bound propagation proves the robustness of a prediction $y_n = \arg\max_k H_{n,k}^{(L)} = \arg\max_k Z_{n,k}^{(L-1)}$ by computing elementwise lower and upper bounds $\boldsymbol{R}^{(l)}$ and $\boldsymbol{S}^{(l)}$ on the pre-activation values $\boldsymbol{Z}^{(l)}$ and elementwise lower and upper bounds $\hat{\boldsymbol{R}}^{(l)}$ and $\hat{\boldsymbol{S}}^{(l)}$ on the post-activation values $\boldsymbol{H}^{(l)}$ via interval arithmetic. If $\forall k \neq y_n : R_{n,y_n}^{(L-1)} > S_{n,k}^{(L-1)}$, then the prediction $y_n$ for node $n$ is provably robust.

In our case, the perturbation set is

$$
\begin{aligned}
\mathbb{B} = \big\{ (\boldsymbol{X}', \boldsymbol{A}') \big| \boldsymbol{X} &\in \{0,1\}^{N \times D}, \boldsymbol{A} \in \{0,1\}^{N \times N}, \\
\boldsymbol{A}' &= \boldsymbol{A}, \\
c_{\boldsymbol{X}}^+ &\cdot ||(\boldsymbol{X}' - \boldsymbol{X})_+||_0 + c_{\boldsymbol{X}}^- \cdot ||(\boldsymbol{X}' - \boldsymbol{X})_-||_0 \leq \epsilon, \\
\forall n &: c_{\boldsymbol{X}}^+ \cdot ||(\boldsymbol{X}'_n - \boldsymbol{X}_n)_+||_0 + c_{\boldsymbol{X}}^- \cdot ||(\boldsymbol{X}'_n - \boldsymbol{X}_n)_-||_0 \leq \rho_n \big\},
\end{aligned}
\tag{8}
$$

with global budget $\epsilon$ and local budgets $\rho_n$.

We propose to compute the lower and upper bounds $\boldsymbol{R}^{(0)}$ and $\boldsymbol{S}^{(0)}$ for the first pre-activation values by solving a knapsack problem with local constraints (see Definition 2) via the methods discussed in Appendix F.1. We define the cost matrix $\boldsymbol{C} \in \mathbb{R}_+^{N \times D}$ to be

$$
C_{n,d} = \begin{cases} c_{\boldsymbol{X}}^+ & \text{if } X_{n,d} = 0 \\ c_{\boldsymbol{X}}^- & \text{if } X_{n,d} = 1. \end{cases}
$$

To obtain an upper bound $S_{i,j}^{(0)}$, we define the value matrix $\boldsymbol{V} \in \mathbb{R}_+^{N \times D}$ to be

$$
V_{n,d} = \begin{cases} \tilde{A}_{i,n} \cdot \max(W_{d,j}^{(0)}, 0) & \text{if } X_{n,d} = 0 \\ \tilde{A}_{i,n} \cdot \max(-W_{d,j}^{(0)}, 0) & \text{if } X_{n,d} = 1. \end{cases}
$$

That is, setting an attribute $X_{n,d}$ from 0 to 1 makes a positive contribution to upper bound $S_{i,j}^{(0)}$ if the corresponding entry of the weight matrix is positive and there is an edge between nodes $n$ and $i$. Setting an attribute $X_{n,d}$ from 1 to 0 makes a positive contribution, if the corresponding entry of the weight matrix is negative and there is an edge between nodes $n$ and $i$. We then solve the knapsack problem and add its optimal value to the original, unperturbed pre-activation value $Z_{i,j}^{(0)}$.

To obtain a lower bound $R_{i,j}^{(0)}$, we analogously define the value matrix $\boldsymbol{V} \in \mathbb{R}_+^{N \times D}$ to be

$$
V_{n,d} = \begin{cases} \tilde{A}_{i,n} \cdot \max(-W_{d,j}^{(0)}, 0) & \text{if } X_{n,d} = 0 \\ \tilde{A}_{i,n} \cdot \max(W_{d,j}^{(0)}, 0) & \text{if } X_{n,d} = 1, \end{cases}
$$

multiply the optimal value of the knapsack problem with $-1$ and then add it to the original, unperturbed pre-activation value $Z_{i,j}^{(0)}$.

After we have obtained the lower and upper bounds for the first layer's pre-activation values, we can use the same procedure as Liu et al. [12] to bound the subsequent post- and pre-activation values:

$$\hat{\boldsymbol{R}}^{(l)} = \text{ReLU}(\boldsymbol{R}^{(l-1)}) \qquad\qquad \text{for } l = 1, \ldots, L-1,$$

$$\hat{\boldsymbol{S}}^{(l)} = \text{ReLU}(\boldsymbol{S}^{(l-1)})s \qquad\qquad \text{for } l = 1, \ldots, L-1,$$

$$\boldsymbol{R}^{(l)} = (\tilde{\boldsymbol{A}}\hat{\boldsymbol{R}}^{(l)})\boldsymbol{W}_+^{(l)} + (\tilde{\boldsymbol{A}}\hat{\boldsymbol{S}}^{(l)})\boldsymbol{W}_-^{(l)} + \mathbf{1}(\boldsymbol{b}^{(l)})^T \qquad \text{for } l = 1, \ldots, L-1,$$

$$\boldsymbol{S}^{(l)} = (\tilde{\boldsymbol{A}}\hat{\boldsymbol{R}}^{(l)})\boldsymbol{W}_-^{(l)} + (\tilde{\boldsymbol{A}}\hat{\boldsymbol{S}}^{(l)})\boldsymbol{W}_+^{(l)} + \mathbf{1}(\boldsymbol{b}^{(l)})^T \qquad \text{for } l = 1, \ldots, L-1.$$

with $\boldsymbol{W}_+ = \max(\mathbf{0}, \boldsymbol{W})$ and $\boldsymbol{W}_- = \min(\mathbf{0}, \boldsymbol{W})$. If $\forall k \neq y_n : R_{n,y_n}^{(L-1)} > S_{n,k}^{(L-1)}$, then the prediction $y_n$ for node $n$ is provably robust. In our experiments, we use linear programming and not the exact dynamic programming solver for the knapsack problems (see Appendix F.1).

### F.3 Convex outer adversarial polytopes for attribute perturbations

Zügner and Günnemann [10] also propose robustness guarantees for $L$-layer graph convolutional networks (see Eq. (7) under attribute perturbations with uniform cost ($c_{\boldsymbol{X}}^+ = c_{\boldsymbol{X}}^- = 1, c_{\boldsymbol{A}}^+ = c_{\boldsymbol{A}}^- = \infty$). However, their method solves a a convex optimization problem and yields robustness guarantees that are always at least as strong as interval bound propagation. Again, we want to generalize their guarantees to arbitrary attribute perturbation costs $c_{\boldsymbol{X}}^+, c_{\boldsymbol{X}}^- \in \mathbb{R}_+$.

Let $\boldsymbol{Z}^{(L-1)}(\boldsymbol{X}, \boldsymbol{A})$ be the pre-activation values in the last layer, given attribute matrix $\boldsymbol{X} \in \{0,1\}^{N \times D}$ and adjacency matrix $\boldsymbol{A} \in \{0,1\}^{N \times N}$.

A prediction $y_n = \arg\max_k \boldsymbol{Z}^{(L-1)}(\boldsymbol{X}, \boldsymbol{A})_{n,k}$ is robust to all perturbed $\boldsymbol{X}', \boldsymbol{A}'$ from a perturbation set $\mathbb{B}$ if $\forall k \neq y_n, \forall(\boldsymbol{X}', \boldsymbol{A}') \in \mathbb{B} : \boldsymbol{Z}^{(L-1)}(\boldsymbol{X}', \boldsymbol{A}')_{n,y_n} > \boldsymbol{Z}^{(L-1)}(\boldsymbol{X}', \boldsymbol{A}')_{n,k}$. This is equivalent to showing that, for all $k \neq y_n$:

$$\min_{(\boldsymbol{X}',\boldsymbol{A}')\in\mathbb{B}} \boldsymbol{Z}^{(L-1)}(\boldsymbol{X}', \boldsymbol{A}')_{n,y_n} - \boldsymbol{Z}^{(L-1)}(\boldsymbol{X}', \boldsymbol{A}')_{n,k} > 0. \tag{9}$$

Solving these $k$ optimization problems is generally intractable due to the ReLU nonlinearities and the discrete set of perturbed graphs $\mathbb{B}$. Zügner and Günnemann [10] thus propose to make two relaxations. The first relaxation is to relax the integrality constraints on the perturbation set $\mathbb{B}$. In our case, this leads to the relaxed set

$$\begin{aligned}
\mathbb{B} = \big\{(\boldsymbol{X}', \boldsymbol{A}') \big| &\boldsymbol{X} \in [0,1]^{N \times D}, \boldsymbol{A} \in [0,1]^{N \times N}, \\
&\boldsymbol{A}' = \boldsymbol{A}, \\
&c_{\boldsymbol{X}}^+ \cdot ||(\boldsymbol{X}' - \boldsymbol{X})_+||_1 + c_{\boldsymbol{X}}^- \cdot ||(\boldsymbol{X}' - \boldsymbol{X})_-||_1 \leq \epsilon, \\
&\forall n : c_{\boldsymbol{X}}^+ \cdot ||(\boldsymbol{X}_n' - \boldsymbol{X}_n)_+||_1 + c_{\boldsymbol{X}}^- \cdot ||(\boldsymbol{X}_n' - \boldsymbol{X}_n)_-||_1 \leq \rho_n \big\}.
\end{aligned} \tag{10}$$

Note that we now use $\ell_1$ instead of $\ell_0$ norms.

The second relaxation is to relax the strict functional dependency between pre- and post-activation values $\boldsymbol{H}^{(l)} = \text{ReLU}(\boldsymbol{Z}^{(l)})$ to its convex hull:

$$\left( Z_{n,d}^{(l)}, H_{n,d}^{(l)} \right) \in \text{hull}\left( \left\{ a, \text{ReLU}(a) \in \mathbb{R} \mid R_{n,d}^{(l)} \leq a \leq S_{n,d}^{(l)} \right\} \right),$$

where $\boldsymbol{R}^{(l)}$ and $\boldsymbol{S}^{(l)}$ are elementwise lower and upper bounds on the pre-activation values $\boldsymbol{Z}^{(l)}$. We propose to use the modified interval bound propagation method introduced in Appendix F.2 to obtain these bounds.

These two relaxations leave us with an optimization problem that is almost identical to the one discussed in [10]. We just weight insertions and deletions with constants $c_{\boldsymbol{X}}^+$ and $c_{\boldsymbol{X}}^+$ instead of uniform weight 1 and have different values for the elementwise pre-activation bounds. We can thus go through the same derivations as in the proof of Theorem 4.3 in [10] while tracking the constants $c_{\boldsymbol{X}}^+$ and $c_{\boldsymbol{X}}^+$ to derive the dual of our relaxed optimization problem, which lower-bounds Eq. (9). We highlight the difference to the result for uniform costs in red. The dual problem for local budgets $\boldsymbol{\rho}$

and global budget $\epsilon$ is

$$\max_{\boldsymbol{\Omega}^{(1)},\ldots,\boldsymbol{\Omega}^{(L-1)},\boldsymbol{\kappa},\lambda} g_{\boldsymbol{\rho},\epsilon}(\boldsymbol{X},\boldsymbol{\Omega},\boldsymbol{\kappa},\lambda)$$

$$\text{s.t. } \boldsymbol{\Omega}^{(l)} \in [0,1]^{N \times h^{(l)}} \text{ for } l = L-1,\ldots,1, \tag{11}$$

$$\boldsymbol{\kappa} \in \mathbb{R}_+^N, \lambda \in \mathbb{R}_+,$$

where $h^{(l)}$ is the number of neurons in layer $l$ and

$$g_{\boldsymbol{\rho},\epsilon}(\boldsymbol{X},\boldsymbol{\Omega},\kappa,\lambda) = \sum_{l=1}^{L-1} \sum_{(n,j)\in\mathcal{I}^{(l)}} \frac{S_{n,j}^{(l)} R_{n,j}^{(l)}}{S_{n,j}^{(l)} - R_{n,j}^{(l)}} \left[\hat{\Phi}_{n,j}^{(l)}\right]_+ - \sum_{l=0}^{L-1} \mathbf{1}^T \boldsymbol{\Phi}^{(l+1)} \boldsymbol{b}^{(l)}$$

$$- \operatorname{Tr}\left[\boldsymbol{X}^T \hat{\boldsymbol{\Phi}}^{(0)}\right] - ||\boldsymbol{\Psi}||_1 - \sum_{n=1}^N \rho_n \cdot \kappa_n - \epsilon \cdot \lambda,$$

where $\mathcal{I}^{(l)}$ is the set of unstable neurons in layer $l$, i.e. $\mathcal{I}^{(l)} = \{(n,j) \mid R_{n,j}^{(l)} \leq 0 < S_{n,j}^{(l)}\}$, and

$$\Phi_j^{(L)} = \begin{cases} -1 & \text{if } j = y^* \\ 1 & \text{if } j = k \\ 0 & \text{otherwise} \end{cases}$$

$$\hat{\boldsymbol{\Phi}}^{(l)} = \tilde{\boldsymbol{A}}^T \boldsymbol{\Phi}^{(l+1)} (\boldsymbol{W}^{(l)})^T \quad \text{for} \qquad\qquad\qquad l = L-1,\ldots,0$$

$$\Phi_{n,j}^{(l)} = \begin{cases} 0 & \text{if } S_{n,j}^{(l)} \leq 0 \\ \hat{\Phi}_{n,j} & \text{if } R_{n,j}^{(l)} > 0 \\ \frac{S_{n,j}^{(l)}}{S_{n,j}^{(l)} - R_{n,j}^{(l)}} \left[\hat{\Phi}_{n,j}^{(l)}\right]_+ + \Omega_{n,j}^{(l)} \left[\hat{\Phi}_{n,j}^{(l)}\right]_- & \text{if } (n,j) \in \mathcal{I}^{(l)} \end{cases} \quad \text{for } l = L-1,\ldots,1$$

$$\Psi_{n,d} = \begin{cases} \max\left(\Delta_{n,d} - c_{\boldsymbol{X}}^+ \cdot (\kappa_n + \lambda), 0\right) & \text{if } X_{n,d} = 0 \\ \max\left(\Delta_{n,d} - c_{\boldsymbol{X}}^- \cdot (\kappa_n + \lambda), 0\right) & \text{if } X_{n,d} = 1 \end{cases}$$

$$\Delta_{n,d} = \left[\hat{\boldsymbol{\Phi}}^{(0)}\right]_+ \cdot (1 - X_{n,d}) - \left[\hat{\boldsymbol{\Phi}}^{(0)}\right]_- \cdot X_{n,d}.$$

The indexing changes are necessary because we consider $L$ and not $(L-1)$-layer networks. The sign changes are necessary because we define the clipping of negative values differently, i.e. $\boldsymbol{X}_- = \min(\boldsymbol{X}, 0)$ and not $\boldsymbol{X}_- = -\min(\boldsymbol{X}, 0)$. Aside from that, the only difference are the costs $c_{\boldsymbol{X}}^+$ and $c_{\boldsymbol{X}}^-$ in the definition of $\boldsymbol{\Psi}$.

Because this is a dual problem, any choice of $\boldsymbol{\Omega}$, $\boldsymbol{\kappa}$ and $\lambda$ yields a lower bound on Eq. (9). But using optimal parameters yields a tighter bound.

**Proposition 13.** *Define value matrix $\boldsymbol{V} = \boldsymbol{\Delta}$ and cost matrix $\boldsymbol{C} = c_{\boldsymbol{X}}^+ \cdot [1 - \boldsymbol{X}]_+ + c_{\boldsymbol{X}}^- \cdot [\boldsymbol{X}]_+$. Then,*

- *The optimal value of $\lambda$ is $\lambda^* = \min_{n,m} V_{n,m} / C_{n,m}$ s.t. $Q_{n,m}^* > 0$, with $\boldsymbol{Q}^*$ computed as in Algorithm 3. That is, $\lambda^*$ is the smallest value-to-cost ratio for which some global budget is allocated.*

- *Define $o_n = \min_m V_{n,m} / C_{n,m}$ s.t. $L_{n,m} > 0$ with $\boldsymbol{L}$ computed as in Algorithm 3. That is, $o_n$ is the smallest value-to-cost ratio in row $n$ for which some local budget is allocated. The optimal value of $\kappa_n$ is $\kappa_n^* = \max(0, o_n - \lambda^*)$.*

*Proof.* For any fixed $\boldsymbol{\Omega}^{(1)},\ldots,\boldsymbol{\Omega}^{(L-1)}$, we can go through the same derivations as in the proofs of Theorems 4.3 and 4.4 in [10] – while keeping track of constants $c_{\boldsymbol{X}}^+, c_{\boldsymbol{X}}^-$ – to show that the above optimization problem is up to an additive constant equivalent to

$$\min_{\lambda \in \mathbb{R}_+, \boldsymbol{\kappa} \in \mathbb{R}_+^N, \boldsymbol{U} \in \mathbb{R}_+^{N \times D}} \sum_{n,d} U_{n,d} + \sum_{n=1}^N \rho_n \cdot \kappa_n + \epsilon \cdot \lambda$$

$$\text{s.t. } U_{n,d} \geq V_{n,d} - C_{n,d} \cdot \kappa_n - C_{n,d} \cdot \lambda$$

with $\boldsymbol{V} = \boldsymbol{\Delta}$ and $\boldsymbol{C} = c_{\boldsymbol{X}}^+ \cdot [1 - \boldsymbol{X}]_+ + c_{\boldsymbol{X}}^- \cdot [\boldsymbol{X}]_+$.

By standard construction, the dual of the above linear program is

$$\max_{\boldsymbol{Q} \in \mathbb{R}^{N \times M}} \sum_{n=1}^{N} \sum_{m=1}^{M} V_{n,m} \cdot Q_{n,m}$$

$$\text{s.t.} \sum_{n=1}^{N} \sum_{m=1}^{M} C_{n,m} \cdot Q_{n,m} \leq \epsilon, \quad \forall n : \sum_{m=1}^{M} C_{n,m} \cdot Q_{n,m} \leq \rho_n,$$

$$\forall n, m : 0 \leq Q_{n,m}, \quad \forall n, m : Q_{n,m} \leq 1.$$

This is exactly the linearly relaxed knapsack problem with local constraints from Eq. (6). During our proof of Proposition 12 via Karush-Kuhn-Tucket conditions, we have already shown that $\lambda^*$ and $\boldsymbol{\kappa}^*$ are the optimal values of the dual (here: primal) variables. $\square$

Since we have optimal values for variables $\lambda$ and $\boldsymbol{\kappa}$ of the optimization problem in Eq. (11), we only need to choose values for $\boldsymbol{\Omega}^{(1)}, \ldots, \boldsymbol{\Omega}^{(L-1)}$. They can be either be optimized via gradient ascent or set to some constant value. In our experiments, we choose

$$\Omega_{n,j}^{(l)} = \frac{S_{n,j}^{(l)}}{S_{n,j}^{(l)} - S_{n,j}^{(l)}},$$

as suggested in [10]. As discussed in [10], the efficiency of the certification procedure can be further improved by slicing the attribute and adjacency matrix to only contain nodes that influence the hidden representation of node $n$ in each layer $l$.

### F.4 Bilinear programming for adjacency perturbations

Zügner and Günnemann [13] derive guarantees for the robustness of graph convolutional networks to edge deletions ($c_{\boldsymbol{X}}^+ = c_{\boldsymbol{X}}^- = c_{\boldsymbol{A}}^+ = \infty, c_{\boldsymbol{A}}^- = 1$). Generalizing these guarantees to arbitrary costs $c_{\boldsymbol{A}}^- \in \mathbb{R}_+$ does not require any additional derivations. Multiplying the cost by a factor $k$ is equivalent to dividing global budget $\epsilon$ and local budgets $\rho_1, \ldots, \rho_n$ by factor $k$.

### F.5 Linearization and dualization for adjacency perturbations

Jin et al. [15] derive robustness guarantees for graph classifiers consisting of a 1-layer graph convolutional network followed by a linear layer and mean pooling under adjacency perturbations with uniform cost ($c_{\boldsymbol{X}}^+ = c_{\boldsymbol{X}}^- = \infty, c_{\boldsymbol{A}}^+ = c_{\boldsymbol{A}}^- = 1$). In the following, we want to generalize their guarantees to arbitrary $c_{\boldsymbol{A}}^+$ and $c_{\boldsymbol{A}}^-$.

In fact, the authors propose two different approaches. The first one involves linearization of the neural network and the formulation of a Lagrange dual problem to enforce symmetry of the perturbed adjacency matrix. The second one combines Fenchel biconjugation with convex outer adversarial polytopes to derive a lower bound that is then optimized via conjugate gradients. While the second objective is sound, its optimization via conjugate gradient may be numerically unstable, which is why an entirely different solver is used in practice.[6] In order to not mispresent the contents of [15], we focus on the first approach, which offers similarly strong guarantees (see Fig.4 in [15]).

The considered graph classification architecture is

$$F(\boldsymbol{X}, \boldsymbol{A}) = \sum_{n=1}^{N} \text{ReLU} \left( \boldsymbol{D}^{-1} \tilde{\boldsymbol{A}} \boldsymbol{X} \boldsymbol{W} \right) \boldsymbol{U} \, / \, N,$$

where $\tilde{\boldsymbol{A}} = \boldsymbol{A}$ is the adjacency matrix after introducing self-loops, i.e. setting all diagonal entries to 1, $\boldsymbol{D} \in \mathbb{R}^{N \times N}$ with $\boldsymbol{D}_{i,i} = \boldsymbol{1}^T \tilde{\boldsymbol{A}}_i$ is the diagonal degree matrix of $\tilde{\boldsymbol{A}}$, $\boldsymbol{W} \in \mathbb{R}^{D \times H}$ are the graph convolution weights and $\boldsymbol{U} \in \mathbb{R}^{H \times K}$ are the linear layer weights for $K$ classes.

---

[6]See https://github.com/RobustGraph/RoboGraph/blob/master/robograph/attack/cvx_env_solver.py

As before, robustness for a specific prediction $y = \arg\max_k F(\boldsymbol{X}, \boldsymbol{A})_k$ under a perturbation set $\mathbb{B}$ can be proven by showing that the classification margin is positive, i.e.

$$\forall k \neq y : \min_{(\boldsymbol{X}', \boldsymbol{A}') \in \mathbb{B}} F(\boldsymbol{X}', \boldsymbol{A}')_y - F(\boldsymbol{X}', \boldsymbol{A}')_k > 0.$$

For this specific architecture, this is equivalent to showing that, for all $k \neq y$,

$$\min_{(\boldsymbol{X}, \boldsymbol{A}) \in \mathbb{B}} \sum_{n=1}^{N} f_{n,k}(\boldsymbol{X}', \boldsymbol{A}') > 0$$

with $f_{n,k}(\boldsymbol{X}', \boldsymbol{A}') = \left( \mathbf{1}^T \tilde{A}'_n \right)^{-1} \text{ReLU}\left( (\tilde{\boldsymbol{A}}'_n)^T \boldsymbol{X}' \boldsymbol{W} \right) (\boldsymbol{U}_{:,y} - \boldsymbol{U}_{:,k}) \, / \, N.$

Note that we could move the degree of node $n$ out of the nonlinearity because it is only a scalar factor. Our perturbation set is

$$\begin{aligned}
\mathbb{B} = \big\{ (\boldsymbol{X}', \boldsymbol{A}') \big| & \boldsymbol{X} \in \{0,1\}^{N \times D}, \boldsymbol{A} \in \{0,1\}^{N \times N}, \\
& \boldsymbol{X}' = \boldsymbol{X}, \\
& c_{\boldsymbol{A}}^+ \cdot ||(\boldsymbol{X}' - \boldsymbol{X})_+||_0 + c_{\boldsymbol{A}}^- \cdot ||(\boldsymbol{X}' - \boldsymbol{X})_-||_0 \leq \epsilon, \\
& \forall n : c_{\boldsymbol{A}}^+ \cdot ||(\boldsymbol{X}'_n - \boldsymbol{X}_n)_+||_0 + c_{\boldsymbol{A}}^- \cdot ||(\boldsymbol{X}'_n - \boldsymbol{X}_n)_-||_0 \leq \rho_n \big\}.
\end{aligned} \tag{12}$$

**Linearization.** Unless one wants to find the worst-case perturbation for each $f_{n,k}$ via brute-forcing (which is a viable approach for very small budgets and discussed in [15]), the first step of the certification procedure is to find $N$ linear models that lower-bound the nodewise functions $f_{n,k}$ for all possible perturbed inputs. That is, we want to find vectors $\boldsymbol{q}^{(1)}, \ldots, \boldsymbol{q}^{(N)} \in \mathbb{R}^N$ and scalars $b^{(1)}, \ldots, b^{(N)} \in \mathbb{R}^K$ such that

$$\forall_{(\boldsymbol{X}', \boldsymbol{A}') \in \mathbb{B}} : f_{n,k}(\boldsymbol{X}', \boldsymbol{A}') \geq \underline{f_{n,k}}(\boldsymbol{X}', \boldsymbol{A}') \tag{13}$$

$$\text{with } \underline{f_{n,k}}(\boldsymbol{X}', \boldsymbol{A}') = \left( \mathbf{1}^T \tilde{A}'_n \right)^{-1} \left( (\tilde{\boldsymbol{A}}'_n)^T \boldsymbol{q}^{(n)} + b^{(n)} \right). \tag{14}$$

Like in [15], we use the linear bounds from CROWN [133].[7] Computing these bounds requires elementwise lower and upper bounds $\boldsymbol{r}^{(1)}, \ldots, \boldsymbol{r}^{(N)}$ and $\boldsymbol{s}^{(1)}, \ldots, \boldsymbol{s}^{(N)}$ on the pre-activation values $\boldsymbol{z}^{(n)} = ((\tilde{\boldsymbol{A}}'_n)^T \boldsymbol{X}' \boldsymbol{W})$ with $\boldsymbol{z}^{(n)} \in \mathbb{R}^H$.

For our new perturbation set, we propose to compute these elementwise lower and upper bounds by solving knapsack problems with local constraints via the algorithms discussed in Appendix F.1. To bound the the pre-activation values $\boldsymbol{z}^{(n)}$ of the $n$th node, we define the cost matrix $\boldsymbol{C} \in \mathbb{R}_+^{1 \times N}$ as

$$C_{1,m} = \begin{cases} \infty & \text{if } m = n \\ c_{\boldsymbol{A}}^+ & \text{if } m \neq n \wedge A_{n,m} = 0 \\ c_{\boldsymbol{A}}^- & \text{if } m \neq n \wedge A_{n,m} = 1. \end{cases}$$

The infinite cost for diagonal elements ensures that they are not adversarially perturbed. Afterall, there is no benefit to attacking elements that are anyway overwritten by self-loops. The cost matrix has shape $1 \times N$, because in every row $n$, there are only $N$ edges that can be perturbed.

To obtain an upper bound $s_h^{(n)}$, we define the value matrix $\boldsymbol{V} \in \mathbb{R}_+^{1 \times N}$ to be

$$V_{1,m} = \begin{cases} \max\left( \boldsymbol{X}_m^T \boldsymbol{W}_{:,h}, 0 \right) & \text{if } A_{n,m} = 0 \\ \max\left( -\boldsymbol{X}_m^T \boldsymbol{W}_{:,h}, 0 \right) & \text{if } A_{n,m} = 1. \end{cases}$$

We then solve the knapsack problem, which tells us how much the pre-activation value can change under $\mathbb{B}$, and then add this optimal value to the original, unperturbed pre-activation value $z_h^{(n)}$.

To obtain a lower bound $r_h^{(N)}$, we analogously define the value matrix $\boldsymbol{V} \in \mathbb{R}_+^{1 \times N}$ to be

$$V_{1,m} = \begin{cases} \max\left( -\boldsymbol{X}_m^T \boldsymbol{W}_{:,h}, 0 \right) & \text{if } A_{n,m} = 0 \\ \max\left( \boldsymbol{X}_m^T \boldsymbol{W}_{:,h}, 0 \right) & \text{if } A_{n,m} = 1. \end{cases}$$

---

[7]These linear bounds are referred to as doubleL in [15].

multiply the optimal value of the knapsack problem with $-1$ and then add it to the original, unperturbed pre-activation value $z_h^{(N)}$.

**Nodewise guarantees.** The next step is to use the constructed linear lower bounds $\underline{f}_{1,k}, \ldots, \underline{f}_{N,k}$ to lower-bound the value of functions $f_{1,k}, \ldots, f_{N,k}$ for each possible number of perturbations admitted by local budgets $\rho_1, \ldots, \rho_N$. Note that we now need to take the degree normalization into account.

For our new perturbation set, we propose to use Algorithm 1 to perform these precomputations. We set the cost parameters to $c^+ = c_{\boldsymbol{A}}^+$ and $c^- = c_{\boldsymbol{A}}^-$. To prevent perturbations of the diagonal entries that are overwritten with self-loops, we set the index set parameters $\mathbb{I}$ and $\bar{\mathbb{I}}$ to $\mathbb{I} = \{(1, m) \mid \boldsymbol{A}_{n,m} = 0\} \cup \{1, n\}$ and $\bar{\mathbb{I}} = \{(1, m) \mid \boldsymbol{A}_{n,m} = 1\} \cup \{1, n\}$. We set the global and local budget parameter of the algorithm to $\epsilon$ and $\rho_n$, respectively. Because Algorithm 1 is defined to solve a maximization and not a minimization problem, we define the value matrix $\boldsymbol{V} \in \mathbb{R}^{1 \times N}$ to be

$$V_{1,m} = \begin{cases} -q_m^{(n)} & \text{if } A_{n,m} = 0 \\ q_m^{(n)} & \text{if } A_{n,m} = 1. \end{cases}$$

That is, setting $A_{n,m}$ from 0 to 1 has a large value if $q_m^{(n)}$ is a negative number. Setting $A_{n,m}$ from 1 to 0 has a large value if $q^{(n)}$ if a large positive number. Algorithm 1 yields a dictionary $\alpha_n$, where $\alpha_n(i, j)$ indicates how much the value of $\underline{f}_{n,k}(\boldsymbol{X}, \boldsymbol{A})$ changes when optimally inserting $i$ and deleting $j$ edges. It further yields a dictionary $\beta_n$, where $\beta_n(i, j)$ contains the optimal set of edges to perturb in row $n$ of the adjacency matrix. The last thing we need to do is to add them to the unperturbed values $\underline{f}_{n,k}(\boldsymbol{X}, \boldsymbol{A})$ while accounting for the fact that perturbing edges also influences the degree and thus the degree normalization. We propose to do so via Algorithm 4

---

**Algorithm 4** Computation of $-1 \cdot \min_{\boldsymbol{X}', \boldsymbol{A}' \in \mathbb{B}} \underline{f}_{n,k}$ from the precomputed worst-case changes without degree normalization stored in dictionary $\alpha_n$.

---

$\text{max\_adds} \leftarrow \lfloor \rho_n \, / \, c^+ \rfloor$
**for** $i = 0, \ldots, \text{max\_adds}$ **do**
    $\text{max\_dels} \leftarrow \lfloor (\rho_n - c^+ \cdot i) \, / \, c^- \rfloor$
    **for** $j = 0, \ldots, \text{max\_dels}$ **do**
        $\text{old\_degree} \leftarrow \mathbf{1}^T \tilde{A}_n$
        $\text{new\_degree} \leftarrow \text{old\_degree} + i - j$
        $\text{new\_degree} \leftarrow \max(1, \min(\text{new\_degree}, N))$       $\triangleright$ Between 1 and $N$, due to self-loop.
        $\alpha_n(i, j) \leftarrow \alpha_n(i, j) - \text{old\_degree} \cdot \underline{f}_{n,k}(\boldsymbol{X}, \boldsymbol{A})$
        $\alpha_n(i, j) \leftarrow \alpha_n(i, j) \, / \, \text{new\_degree}$
**return** $\alpha_n$

---

Note that we compute the negative value of the lower bound, because all discussed algorithms are designed for maximization problems.

**Combining the nodewise guarantees.** Now we have, for each node $n$ and every possible number of insertions $i$ and deletions $j$, the negative of a lower bound on $f_{n,k}$, which is stored in $\alpha_n(i, j)$. The last step is to combine these nodewise guarantees while complying with global budget $\epsilon$. To this end, we can just reuse Algorithm 2 with $c^+ = c_{\boldsymbol{A}}^+$ and $c^- = c_{\boldsymbol{A}}^-$. It yields a sequence $t$ of length $N$, whose $n$th element is the set of the worst indices to perturb in row $n$ of the adjacency matrix. It further yields a lower bound on the negative classification margin $-1 \cdot (F(\boldsymbol{X}', \boldsymbol{A}')_y - F(\boldsymbol{X}', \boldsymbol{A}')_k)$ for a specific $k$. If after performing the above procedure for all $k \neq y$ all classification margins are positive, then the prediction is provably robust.

**Enforcing symmetry via dualization.** An important aspect of the discussion in [15] is that one may want to introduce the additional constraint $\boldsymbol{A}' = \boldsymbol{A}'^T$ when proving robustness for symmetric graphs.

The authors show that

$$\min_{(mX, A') \in \mathbb{B}} \sum_{n=1}^{N} \underline{f_{n,k}}(X', A') \quad \text{s.t. } A' = A'^T$$

$$\geq \max_{\Lambda \in \mathbb{R}^{N \times N}} \min_{(mX, A') \in \mathbb{B}} \sum_{n=1}^{N} \underline{f_{n,k}}(X', A') + \text{Tr}((\Lambda^T - \Lambda)A'),$$

with dual variable $\Lambda \in \mathbb{R}^{N \times N}$. Note that $\text{Tr}((\Lambda^T - \Lambda)A) = \sum_{n=1}^{N} \sum_{m=1}^{M} (\Lambda^T - \Lambda)_{n,m}^T \cdot A'_{n,m}$.

The inner optimization problem can be solved exactly via the "nodewise guarantees" and "combining the nodewise guarantees" steps above, after replacing each parameter vector $q^{(n)}$ of the linearized models $\underline{f_{n,k}}$ defined in Eq. (14) with $q^{(n)} + \left(1^T \tilde{A}'_n\right) \cdot (\Lambda - \Lambda^T)_n$.

Because the inner optimization problem is solved exactly, dual variable $\Lambda$ can be optimized via gradient ascent on $\text{Tr}((\Lambda^T - \Lambda)A')$ (Danskin's theorem).

### F.6 Policy iteration for adjacency perturbations

Bojchevski and Günnemann [11] derive robustness certificates for models where the predictions are a linear function of the (personalized) PageRank. Specifically, they consider the following architecture called $\pi$-PPNP:

$$Y = \text{softmax}(\Pi H), \quad H_{v,:} = f_\theta(X_{v,:}), \quad \Pi = (1 - \alpha)(I_N - \alpha D^{-1} A)^{-1}$$

where $X$ the feature matrix of the graph, $f$ is a neural network with parameter $\theta$, $H_{v,:}$ the prediction for node $v$ and $\Pi$ the personalized PageRank matrix with teleport probability $\alpha$ (i.e. $\Pi_{v,:}$ is the personalized PageRank of node $v$). Further note that $I_N$ is the identity matrix and $D$ is the degree matrix $D_{ii} = \sum_j = A_{ij}$ of a graph $G = (\mathcal{V}, \mathcal{E})$ with nodes $\mathcal{V}$ and edges $\mathcal{E}$.

Their certificates are against adjacency perturbations with uniform costs ($c_A^+ = c_A^- = 1$) and we will generalize their certificate to arbitrary costs for edge insertion and deletion. Let the following set denote all admissible perturbed graphs under costs $c_A^+$ for edge insertion and $c_A^-$ for edge deletion:

$$\begin{aligned}
\mathcal{Q}_\mathcal{F} = \{(\mathcal{V}, \tilde{\mathcal{E}} := \mathcal{E}_f \cup \mathcal{F}_+) \,|\, &\mathcal{F}_+ \in \mathcal{P}(\mathcal{F}), \\
&c_A^+ \cdot |\tilde{\mathcal{E}} \setminus \mathcal{E}| + c_A^- \cdot |\mathcal{E} \setminus \tilde{\mathcal{E}}| \leq \epsilon, \\
&c_A^+ \cdot |\tilde{\mathcal{E}}^v \setminus \mathcal{E}^v| + c_A^- \cdot |\mathcal{E}^v \setminus \tilde{\mathcal{E}}^v| \leq \rho_v, \forall v\}
\end{aligned} \tag{15}$$

where $\mathcal{E}_f$ is a set of fixed edges that cannot be modified, $\mathcal{F} \subseteq \mathcal{V} \times \mathcal{V}$ the set of fragile edges that can be modified. Here, $\mathcal{F}_+ \subseteq \mathcal{F}$ denotes the set of edges that are included in the perturbed graph (analogously $\mathcal{F}_- \subseteq \mathcal{F}$ the set of edges that are not included anymore in the perturbed graph). Here, $\epsilon$ denotes the global budget and $\rho_v$ the local budget for node $v$.

We assume a fixed graph $G$, set of fragile edges $\mathcal{F}$, global budget $B$ and local budgets $b_v$. Note that since $\pi$-PPNP separates prediction from propagation we can further assume fixed model logits $H$. Following Bojchevski and Günnemann [11] we further define the worst-case margin between classes $y_t$ and $c$ under any perturbed graph $\tilde{G} \in \mathcal{Q}_\mathcal{F}$ (Problem 1 of Bojchevski and Günnemann [11]):

$$m_{y_t,c}^*(t) = \min_{\tilde{G} \in \mathcal{Q}_\mathcal{F}} m_{y_t,c}(t) = \min_{\tilde{G} \in \mathcal{Q}_\mathcal{F}} \pi_{\tilde{G}}(e_t)^T (H_{:,y_t} - H_{:,c}) \tag{16}$$

where $e$ is the unit vector and $\pi_{\tilde{G}}(e_t)$ the personalized PageRank vector of target node $t$ under perturbed graph $\tilde{G}$. We want to show that a specific target node $t$ is certifiably robust w.r.t. the logits $H$ and the set $\mathcal{Q}_\mathcal{F}$. This is the case if $m_{y_t,*}^*(t) = \min_{c \neq y_t} m_{y_t,c}^*(t) > 0$. In the following we will derive guarantees for this under local and global adversarial budgets, respectively.

**Local constraints only.** Bojchevski and Günnemann [11] phrase the problem of Equation 16 as a more general average cost infinite horizon Markov decision process and present a policy iteration algorithm that solves it in polynomial time. Following their derivations we also define $r = -(H_{:,y_t} - H_{:,c})$, where $r_v$ denotes the rewards a random walker gets for visiting node $v$ [11]. The following adapted policy iteration computes the worst-case graph $\tilde{G} \in \mathcal{Q}_\mathcal{F}$ under arbitrary costs:

---

**Algorithm 5** Policy Iteration with local budgets under arbitrary costs

---
**Require:** Graph $G = (\mathcal{V}, \mathcal{E})$, reward $\boldsymbol{r}$, set of fixed edges $\mathcal{E}_f$, fragile edges $\mathcal{F}$, local budgets $b_v$ and
    costs $c_{\boldsymbol{A}}^+$, for edge insertion and $c_{\boldsymbol{A}}^-$ for edge deletion
 1: Initialization: arbitrary $\mathcal{W}_0 \subseteq \mathcal{F}$, $\boldsymbol{A}^G$ corresponding to $G$
 2: **while** $\mathcal{W}_k \neq \mathcal{W}_{k-1}$ **do**
 3:     Solve $(\boldsymbol{I}_N - \alpha \boldsymbol{D}^{-1}\boldsymbol{A})\boldsymbol{x} = \boldsymbol{r}$ for $\boldsymbol{x}$ where $\boldsymbol{A}_{ij} = 1 - \boldsymbol{A}_{ij}^G$ if $(i,j) \in \mathcal{W}_k$
 4:     $l_{ij} \leftarrow (1 - 2\boldsymbol{A}_{ij}^G)(\boldsymbol{x}_j - \frac{\boldsymbol{x}_i - \boldsymbol{r}_i}{\alpha})$ for all $(i,j) \in \mathcal{F}$
 5:     **for** $v \in \mathcal{V}$ **do**
 6:         $\boldsymbol{m} \leftarrow \arg\max_{\boldsymbol{m}} \sum_{(v,j) \in F^+ \cup F^-} m_j l_{v,j}$ s.t. $\sum_{(v,j) \in F^-} c_{\boldsymbol{A}}^- m_j + \sum_{(v,j) \in F^+} c_{\boldsymbol{A}}^+ m_j \leq$
   $\rho_v, m_j \in \{0, 1\}$
 7:         $\mathcal{L}_v \leftarrow \{(v,j) \in \mathcal{F} \mid \boldsymbol{m}_j = 1\}$
 8:     $\mathcal{W}_k \leftarrow \bigcup_v \mathcal{L}_v$
 9:     $k \leftarrow k + 1$
    **return** $\mathcal{W}_k$

---

Line 6 of the policy iteration requires us to find those fragile edges with the largest score $l$ under the local budget $\rho_v$. This is an instance of the Knapsack problem, which we discussed earlier. In practice we solve this using dynamic programming, specifically we call Algorithm 2 for row-vector $\boldsymbol{V}_{1,j} \leftarrow l_{v,j}$, $\epsilon \leftarrow \infty$ and $\rho_1 \leftarrow \rho_v$. Finally, note that Bojchevski and Günnemann [11] show the correctness of the policy iteration for arbitrary sets of admissible perturbed graphs $\mathcal{Q}_{\mathcal{F}}$, thus the correctness of Algorithm 5 follows from their proof. In particular, the additional costs for insertion and deletion does not change the fact that we can model the problem as a Markov decision process.

**Local and global constraints.** Lastly, for local and global constraints we can use the same auxiliary graph as introduced by Bojchevski and Günnemann [11]. In particular, the additional constraints are only additional enrichments of the Linear Program resulting from the auxiliary graph, yielding a quadratically constrained linear program. To account for additional costs, we have to replace constraint (4f) of their QCLP with the following constraints that also considers additional costs:

$$\sum_{(i,j) \in \mathcal{F}} c_{\boldsymbol{A}}^- \cdot [(i,j) \in \mathcal{E}]\beta_{i,j}^0 + c_{\boldsymbol{A}}^+ \cdot [(i,j) \notin \mathcal{E}]\beta_{i,j}^1 \leq \epsilon$$

### F.7 Sparsity-aware randomized smoothing for attribute and adjacency perturbations

Bojchevski et al. [14] present robustness certificates for sparse data based on randomized smoothing. Their main idea is to introduce a sparsity-aware smoothing distribution that preserves the sparsity of the underlying data distribution. Such sparsity-aware certificates for discrete data are currently state-of-the-art for certifying robustness of GNNs against structure and attribute perturbations.

Bojchevski et al. [14] derive robustness guarantees only under uniform costs ($c_{\boldsymbol{A}}^+ = c_{\boldsymbol{A}}^- = c_{\boldsymbol{X}}^+ = c_{\boldsymbol{X}}^- = 1$). Here we make use of our findings (Appendix E) and generalize their certificate to graph edit distances under arbitrary costs for insertion and deletion. Specifically for sparse graphs we model adversaries that perturb nodes by adding ones (flip $0 \rightarrow 1$) to or deleting ones (flip $1 \rightarrow 0$) of the adjacency or feature matrix. Since adversaries can perturb both the features $\boldsymbol{X}$ and edges $\boldsymbol{A}$ of a graph $G = (\boldsymbol{A}, \boldsymbol{X})$ we consider the threat model $\mathcal{B}_\epsilon(G) = \{\tilde{G} \mid \delta(G, \tilde{G}) \leq \epsilon\}$ with

$$
\begin{aligned}
\delta(G, \tilde{G}) = & c_{\boldsymbol{A}}^+ \sum_i \sum_j \mathbb{I}(\tilde{\boldsymbol{A}}_{ij} = \boldsymbol{A}_{ij} + 1) + c_{\boldsymbol{A}}^- \sum_i \sum_j \mathbb{I}(\tilde{\boldsymbol{A}}_{ij} = \boldsymbol{A}_{ij} - 1) \\
& + c_{\boldsymbol{X}}^+ \sum_i \sum_j \mathbb{I}(\tilde{\boldsymbol{X}}_{ij} = \boldsymbol{X}_{ij} + 1) + c_{\boldsymbol{X}}^- \sum_i \sum_j \mathbb{I}(\tilde{\boldsymbol{X}}_{ij} = \boldsymbol{X}_{ij} - 1)
\end{aligned}
\tag{17}
$$

where $\mathbb{I}$ is an indicator function and $c_{\boldsymbol{A}}^+, c_{\boldsymbol{A}}^-, c_{\boldsymbol{X}}^+, c_{\boldsymbol{X}}^-$ the corresponding costs for addition and deletion.

**Sparsity-aware smoothing distribution.** Bojchevski et al. [14] propose a family of smoothing distributions that preserves sparsity of the underlying data. Applied to graphs, the distribution of randomly perturbed graphs $(\tilde{\boldsymbol{X}}, \tilde{\boldsymbol{A}})$ given clean graph $(\boldsymbol{X}, \boldsymbol{A})$ is defined by the probability mass

function $Q : \{0,1\}^{N \times D} \times \{0,1\}^{N \times N} \to [0,1]$ with

$$Q(\tilde{\boldsymbol{X}}, \tilde{\boldsymbol{A}}) = \prod_{i,j} q_{\boldsymbol{X}}(\tilde{\boldsymbol{X}}_{i,j}) \prod_{i,j} q_{\boldsymbol{A}}(\tilde{\boldsymbol{A}}_{i,j}) \tag{18}$$

and elementwise functions $q_{\boldsymbol{X}}, q_{\boldsymbol{A}} : \{0,1\} \to [0,1]$ with

$$q_{\boldsymbol{X}}(\tilde{\boldsymbol{X}}_{i,j}) = \begin{cases} (p_{\boldsymbol{X}}^+)^{1-\boldsymbol{X}_{ij}} (p_{\boldsymbol{X}}^-)^{\boldsymbol{X}_{ij}} & \text{if } \tilde{\boldsymbol{X}}_{i,j} \neq \boldsymbol{X}_{i,j} \\ 1 - (p_{\boldsymbol{X}}^+)^{1-\boldsymbol{X}_{ij}} (p_{\boldsymbol{X}}^-)^{\boldsymbol{X}_{ij}} & \text{otherwise} \end{cases}$$

$$q_{\boldsymbol{A}}(\tilde{\boldsymbol{A}}_{i,j}) = \begin{cases} (p_{\boldsymbol{A}}^+)^{1-\boldsymbol{A}_{ij}} (p_{\boldsymbol{A}}^-)^{\boldsymbol{A}_{ij}} & \text{if } \tilde{\boldsymbol{A}}_{i,j} \neq \boldsymbol{A}_{i,j} \\ 1 - (p_{\boldsymbol{A}}^+)^{1-\boldsymbol{A}_{ij}} (p_{\boldsymbol{A}}^-)^{\boldsymbol{A}_{ij}} & \text{otherwise} \end{cases}$$

where parameters $p_{\boldsymbol{X}}^+, p_{\boldsymbol{X}}^-, p_{\boldsymbol{A}}^+, p_{\boldsymbol{A}}^- \in [0,1]$ specify the probability of randomly inserting or deleting attributes and edges, respectively. Note that by using different probabilities to flip $0 \to 1$ with probability $p^+$ and $1 \to 0$ with probability $p^-$, the smoothing distribution allows to preserve sparsity of the data especially for $p^+ \ll p^-$.

**Graph edit distance certificates.** The certificate of Bojchevski et al. [14] guarantees robustness against the threat model $\mathcal{B}_{r_{\boldsymbol{A}}^+, r_{\boldsymbol{A}}^-, r_{\boldsymbol{X}}^+, r_{\boldsymbol{X}}^-}(G)$, which bounds the perturbations individually by having separate radii for addition $r^+$ and deletion $r^-$. To certify robustness under graph edit distance with arbitrary costs and given budget $\epsilon$, we have to certify robustness with respect to all balls $\mathcal{B}_{r_{\boldsymbol{A}}^+, r_{\boldsymbol{A}}^-, r_{\boldsymbol{X}}^+, r_{\boldsymbol{X}}^-}(G)$ with

$$c_{\boldsymbol{A}}^+ \cdot r_{\boldsymbol{A}}^+ + c_{\boldsymbol{A}}^- \cdot r_{\boldsymbol{A}}^- + c_{\boldsymbol{X}}^+ \cdot r_{\boldsymbol{X}}^+ + c_{\boldsymbol{X}}^- \cdot r_{\boldsymbol{X}}^- \leq \epsilon.$$

Finally note that in practice we do not have to consider all balls since if we can certify one radius the classifier is also robust for smaller radii [14]. Therefore the number of combinations one has to consider reduces significantly in practice.

# G   Local budgets and local robustness

As discussed in Section 4, the domain $\mathbb{X}$ may be composed of $N$ distinct elements, i.e. $\mathbb{X} = \mathbb{A}^N$ for some set $\mathbb{A}$. Similarly, the task may require making $M$ distinct predictions, i.e. the co-domain is $\mathbb{Y} = \mathbb{B}^M$ for some set $\mathbb{B}$. In certain tasks, like node classification, it is common to enforce local distance constraints on each of the $N$ elements of a perturbed input $x'$ and investigate the robustness of some subset of prediction elements $\mathbb{T} \subseteq \{1, \ldots, M\}$. In the following, we discuss how to generalize our definition of robustness (see Definition 1) to enforce such local budget constraints and quantify such local robustness.

For this discussion, recall that a model is $(\mathbb{G}, d_{\text{in}}, d_{\text{out}}, \epsilon, \delta)$-equivariant-robust if

$$\left( \max_{x' \in \mathbb{X}} \max_{g \in \mathbb{G}} d_{\text{out}}(f(x), g^{-1} \bullet f(g \bullet x')) \text{ s.t. } d_{\text{in}}(x, x') \leq \epsilon \right) \leq \delta.$$

**Local budgets.**   Local budget constraints can be easily accounted for by replacing the original optimization domain $\{g \bullet x' \mid g \in \mathbb{G}, x' \in \mathbb{X}, d_{\text{in}}(x, x') \leq \epsilon\}$ in the above definition with

$$\left\{ g \bullet x' \mid g \in \mathbb{G}, x' \in \mathbb{A}^N, d_{\text{in}}(x, x') \leq \epsilon, \forall n : d_{\text{loc}}(x_n, x'_n) \leq \rho_n \right\},$$

with global distance $d_{\text{in}} : \mathbb{A}^N \to \mathbb{R}_+$, global budget $\epsilon$, local distance $d_{\text{loc}} : \mathbb{A} \to \mathbb{R}_+$, and local budgets $\rho_1, \ldots, \rho_N \in \mathbb{R}_+$.

**Local robustness.**   Quantifying the robustness of some subset of prediction indices $\mathbb{T} \subseteq \{1, \ldots, M\}$ requires a function $d_{\text{out}} : \mathbb{B}^{|\mathbb{T}|} \times \mathbb{B}^{|\mathbb{T}|} \to \mathbb{R}_+$. We need to be careful about where we introduce the indexing to pass the original $M$-dimensional predictions into $d_{\text{out}}$. It is not correct to measure output distance using $d_{\text{out}}(f(x)_{\mathbb{T}}, g^{-1} \bullet f(g \bullet x')_{\mathbb{T}})$, i.e. it is not correct to index before reverting the effect of group action $g \in \mathbb{G}$. Afterall, group $\mathbb{G}$ may act differently on $\mathbb{B}^M$ and $\mathbb{B}^{|\mathbb{T}|}$. For example, rotating a point cloud around its center of mass and then subsampling it is not the same as rotating around the center of mass of the subsampled point cloud. Therefore, $d_{\text{out}}(f(x)_{\mathbb{T}}, g^{-1} \bullet f(g \bullet x')_{\mathbb{T}})$ may be very large, even if $f$ is perfectly equivariant and not affected by the small perturbation $(x' - x)$. Instead, we need to measure output distance using

$$d_{\text{out}}(f(x)_{\mathbb{T}}, (g^{-1} \bullet f(g \bullet x'))_{\mathbb{T}}).$$

Combining local budgets and local robustness leads to the following definition:

**Definition 3.**  *Consider a ground truth function $y : \mathbb{X} \to \mathbb{Y}$ with $\mathbb{X} = \mathbb{A}^N$ and $\mathbb{Y} = \mathbb{B}^N$ for some sets $\mathbb{A}, \mathbb{B}$. Further consider input distance function $d_{\text{in}} : \mathbb{A}^N \times \mathbb{A}^N \to \mathbb{R}_+$, local distance function $d_{\text{loc}} : \mathbb{A} \times \mathbb{A} \to \mathbb{R}_+$, a set of output indices $\mathbb{T} \subseteq \{1, \ldots, M\}$ and an output distance function $d_{\text{out}} : \mathbb{B}^{|T|} \times \mathbb{B}^{|T|} \to \mathbb{R}_+$. Assume that $y$ is equivariant with respect to the action of group $\mathbb{G}$. Then, a prediction $f(x)$ for clean input $x \in \mathbb{X}$ is $(\mathbb{G}, d_{\text{in}}, d_{\text{out}}, \epsilon, \boldsymbol{\rho}, \delta, \mathbb{T})$-equivariant-robust if*

$$\max_{x' \in \mathbb{X}} \max_{g \in \mathbb{G}} d_{\text{out}} \left( f(x)_{\mathbb{T}}, (g^{-1} \bullet f(g \bullet x'))_{\mathbb{T}} \right)$$

$$\text{s.t. } d_{\text{in}}(x, x') \leq \epsilon, \quad \forall n : d_{\text{loc}}(x_n, x'_n) \leq \rho_n,$$

*is less than or equal to $\delta$.*

Similar to what we discussed in Section 5, using an equivariant model $f$ means that $d_{\text{out}}\left( f(x)_{\mathbb{T}}, (g^{-1} \bullet f(g \bullet x'))_{\mathbb{T}} \right) = d_{\text{out}}\left( f(x)_{\mathbb{T}}, f(x')_{\mathbb{T}} \right)$ and we recover the traditional notion of adversarial robustness with local constraints for the subset of predictions $\mathbb{T}$.

# H  Definition of robustness for non-compact sets

For our discussion in Section 4, we assumed compactness of all optimization domains, so that minima and maxima always exist. In particular, we assumed that the set $\{x' \in \mathbb{X} \mid d_{\text{in}}(x, x') \leq \epsilon\}$ is compact for all $\epsilon \in \mathbb{R}_+$ and thus contains all its limit points. If this is not the case, the minimum $\min_{g \in \mathbb{G}} d_{\text{in}}(x, g \bullet x')$ may not exist for certain groups $\mathbb{G}$ and perturbed inputs $x' \in \mathbb{X}$. In this case, the action-induced distance needs to be defined as $\hat{d}_{\text{in}}(x, x') = \inf_{g \in \mathbb{G}} d_{\text{in}}(x, g \bullet x')$.

Furthermore, our discussion on perturbation models in Section 4.1 may no longer hold. We may have $\{x' \mid \hat{d}_{\text{in}}(x, x') \leq \epsilon\} \supsetneq \{g \bullet x' \mid g \in \mathbb{G}, d_{\text{in}}(x, x') \leq \epsilon\}$, since the l.h.s. set also contains perturbed inputs $x' \in \mathbb{X}$ that can be mapped arbitrarily close to a $d_{\text{in}}$-ball of radius $\epsilon$, whereas the r.h.s. set only contains perturbed inputs that can be mapped into the interior of the ball (via the inverse group element $g^{-1}$). To ensure equality, we also need to include these limit points. This leads us to the following definition:

**Definition 4.** *Assume that ground truth function $y : \mathbb{X} \to \mathbb{Y}$ is equivariant with respect to the action of group $\mathbb{G}$. Further define the set of limit points*

$$\mathbb{L} = \{g \bullet x' \mid \inf_{h \in \mathbb{G}} d_{\text{in}}(x, h \bullet x') = \epsilon \ \wedge \ g \in \arg\inf_{g \in \mathbb{G}} d_{\text{in}}(x, g \bullet x')\}.$$

*Then, a prediction $f(x)$ for clean input $x \in \mathbb{X}$ is $(\mathbb{G}, d_{\text{in}}, d_{\text{out}}, \epsilon, \delta)$-equivariant-robust if*

$$(\max_{x' \in \mathbb{X}} \max_{g \in \mathbb{G}} d_{\text{out}}(f(x), g^{-1} \bullet f(g \bullet x')) \ \text{ s.t. } \ d_{\text{in}}(x, x') \leq \epsilon \vee x' \in \mathbb{L}) \leq \delta.$$

One could also constrain $x'$ to be in the closure of $\{x' \in \mathbb{X} \mid d_{\text{in}}(x, x') \leq \epsilon\}$. This would however correspond to a potentially stronger notion of robustness, since the closure may also contain limit points that are not in $\mathbb{L}$. The model may thus need to be robust to a larger set of perturbed inputs.

Note that all our discussions in Section 5 also apply to this notion of robustness.

# I  Definition of robustness for non-isometric group actions

For our discussion in Section 4, we assumed that group $\mathbb{G}$ acts isometrically on input space $\mathbb{X}$, i.e. $\forall x, x' \in \mathbb{X}, \forall g \in \mathbb{G} : d_{\text{in}}(g \bullet x, g \bullet x') = d_{\text{in}}(x, x')$. If this is not the case, one could also try to define an action-induced distance as follows:

$$\hat{d}_{\text{in}}(x, x') = \min_{g, h \in \mathbb{G}} d_{\text{in}}(h \bullet x, g \bullet x'),$$

i.e. try to minimize the distance between clean input $x$ and perturbed input $x'$ by transforming both via group actions. However, we evidently have $\min_{g, h \in \mathbb{G}} d_{\text{in}}(h \bullet x, g \bullet x') \leq \min_{g \in \mathbb{G}} d_{\text{in}}(x, g \bullet x')$. We can thus distinguish two cases

**Case 1.** Both action-induced distances are identical for all $x, x' \in \mathbb{X}$ (which is always the case when $\mathbb{G}$ acts isometrically). In this case, introducing the second group element is redundant.

**Case 2.** There is a pair $x, x' \in \mathbb{X}$ such that $\min_{g, h \in \mathbb{G}} d_{\text{in}}(h \bullet x, g \bullet x') < \min_{g \in \mathbb{G}} d_{\text{in}}(x, g \bullet x')$. In this case, Desideratum 3 from Section 4 would be violated, i.e. the action-induced distance $\hat{d}_{\text{in}}$ would not optimally preserve the original distance $d_{\text{in}}$.

Nevertheless, one could conceivably define robustness for equivariant tasks with non-isometric group actions as follows:

$$(\max_{x' \in \mathbb{X}} \max_{g, h \in \mathbb{G}} d_{\text{out}}(h^{-1} \bullet f(h \bullet x), g^{-1} \bullet f(g \bullet x')) \ \text{ s.t. } \ d_{\text{in}}(x, x') \leq \epsilon) \leq \delta.$$

All our discussions in Section 5 also apply to this notion of robustness.

However, this notion would be qualitatively different from Definition 1: Rather than requiring robustness robustness for a specific clean prediction $f(x)$, one would require robustness for the entire set of predictions $\{f(h \bullet x) \mid h \in \mathbb{G}\}$. As such, it is arguably not as good of a generalization of the classic notion of adversarial robustness. If one opts to use this notion of robustness for certification, one should at least state it explicitly. Afterall, being explicit about the underlying semantics of tasks and certificates is a main focus of this work.

Note that typically considered group actions (translation, rotation, reflection, permutation, etc.) act isometrically on the typically considered input spaces (e.g. Euclidean spaces), meaning these considerations should usually not be relevant in practice.

# J  Relation to transformation-specific robustness

As discussed in Section 2, works on transformation-specific robustness are concerned with robustness to unnoticeable parametric transformations, such as small rotations or translations. More formally, consider a parametric function $\psi_\theta : \mathbb{X} \to \mathbb{X}$ with parameter $\theta \in \Theta$. A model is considered robust to transformation-specific attacks if $\max_{\theta \in \Theta} d_{\text{out}}(f(x), f(\psi_\theta(x))) \leq \delta$ for some small $\delta \in \mathbb{R}_+$. Usually, $d_{\text{out}}$ is defined as the 0–1-loss $\mathbb{1}[y \neq y']$ [56–65]. The underlying assumption is that the transformations $\psi_\theta$ do not change the ground truth label of an input.

The key differences to our proposed notion of robustness are that (1) transformation-specific robustness does not consider that there are transformations for which predictions explicitly need to change and (2) works on transformation-specific usually do not consider unstructured perturbations like camera noise which are only constrained by a distance function $d_{\text{in}}$.

In the following, we demonstrate that transformation-specific robustness can nevertheless be framed as a special case of our proposed notion of robustness — or relaxations thereof. We can distinguish three cases, depending on the structure of $\{\psi_\theta \mid \theta \in \Theta\}$.

**Case 1.** In the first case, the family of transformations $\{\psi_\theta \mid \theta \in \Theta\}$ forms a group $\mathbb{G}$ with composition as the group operator ($\psi_\theta \cdot \psi_{\theta'} = \psi_\theta \circ \psi_{\theta'}$) and application of the transformation as the group action ($\psi_\theta \bullet x = \psi_\theta(x)$).[8] In this case, the above definition of transformation-specific robustness can be reformulated as $\max_{g \in \mathbb{G}} d_{\text{out}}(f(x), f(g \bullet x)) \leq \delta$. Assuming that our distance function fulfills $d_{\text{in}}(x, x') = 0 \iff x = x'$, this is equivalent to

$$\max_{x' \in \mathbb{X}} \max_{g \in \mathbb{G}} d_{\text{out}}(f(x), f(g \bullet x')) \leq \delta \quad \text{s.t. } d_{\text{in}}(x, x') \leq 0. \tag{19}$$

We observe that this is a special of our notion of robustness (see Definition 1), where the adversary has a budget of $\epsilon = 0$ (i.e. can only apply group actions) and the task is group invariant.

**Case 2.** In the second case, the set of transformations is restricted to a subset $\mathbb{H}$ of $\mathbb{G}$ that is not a proper group. For example, one may restrict an adversary to small translations instead of arbitrary translations (see, e.g., [57]). In this case, transformation-specific robustness can be framed as

$$\max_{x' \in \mathbb{X}} \max_{g \in \mathbb{H}} d_{\text{out}}(f(x), f(g \bullet x')) \leq \delta \quad \text{s.t. } d_{\text{in}}(x, x') \leq 0,$$

which is a relaxation of Eq. (19) since $\mathbb{H} \subset \mathbb{G}$.

**Case 3.** In the third case, the family $\{\psi_\theta \mid \theta \in \Theta\}$ does not have a group structure or $\psi_\theta(x)$ is not a proper group action (e.g. due to interpolation artifacts after rotation of an image). In this case, we can choose an arbitrary $\epsilon \in \mathbb{R}$ and define an arbitrary $d_{\text{in}}(x, x')$ such that $\{x' \mid d_{\text{in}}(x, x') \leq \epsilon\} = \{\psi_\theta(x) \mid \theta \in \Theta\}$. That is, the distance between $x$ and $x'$ is smaller than $\epsilon$ if $x'$ can be obtained via a transformation of $x$. With this choice of $d_{\text{in}}$, transformation-specific robustness is equivalent to

$$\max_{x' \in \mathbb{X}} d_{\text{out}}(f(x), f(x')) \leq \delta \quad \text{s.t. } d_{\text{in}}(x, x') \leq \epsilon.$$

This is an instance of classic adversarial robustness, which is a special case of our proposed notion of robustness (no equivariance).

---

[8] Such a family of transformations is referred to as "resolvable" in transformation-specific robustness literature.

