# OpenReview forum: "Provable Adversarial Robustness for Group Equivariant Tasks: Graphs, Point Clouds, Molecules, and More"
_NeurIPS.cc/2023/Conference — NeurIPS 2023 poster_

### Official Review · Reviewer_nMMh · 2023-07-03

**Soundness:** 3 good
**Presentation:** 2 fair
**Contribution:** 2 fair
**Rating:** 5
**Confidence:** 3

**Summary:**

This paper studies the robustness certification for equivariant tasks. Specifically, the adversarial robustness with group equivariance is defined in terms of  the input and output distance. Then equivariance-preserving randomized smoothing is introduced as well as various smoothing schemes. Experiments show that robustness guarantees can be obtained via equivariance preserving randomized smoothing.

**Strengths:**

- The motivation of studying robustness with equivariance is clear and the problem is well formulated.

- The theory part looks sound and detailed and the paper is well organized.

**Weaknesses:**

- There are not enough baselines to show that the equivariance can improve the provable robustness as claimed on line 8 in the abstract. It is expected to compare with some previous works to back up the claim that prior work underestimates the robustness as line 399 states.

- If I understand correctly,  the permutation invariant task is a special case of equivariant task as the experiment of point cloud classification shows. Therefore, transformation specific robustness work [1, 2, 3, 4] can be involved like image/point cloud classification against 2D/3D rotation or translation. It will be of great interest to discuss the connection between these resolvable/non-resolvable transformations and group equivariance.

- Each experiment needs more justifications for choosing different smoothing schemes and measure types.

- The notations in section 3 regarding measure-theoretic randomized smoothing are a bit confusing and not easy to follow. It is better to explain each domain notation ahead and explicitly.


---

[1] Li, Linyi, et al. "Tss: Transformation-specific smoothing for robustness certification." *Proceedings of the 2021 ACM SIGSAC Conference on Computer and Communications Security*. 2021.

[2] Chu, Wenda, Linyi Li, and Bo Li. "Tpc: Transformation-specific smoothing for point cloud models." *International Conference on Machine Learning*. PMLR, 2022.

[3] Hu, Hanjiang, et al. "Robustness Certification of Visual Perception Models via Camera Motion Smoothing." *Conference on Robot Learning*. PMLR, 2022.

[4] Hao, Zhongkai, et al. "Gsmooth: Certified robustness against semantic transformations via generalized randomized smoothing." *International Conference on Machine Learning*. PMLR, 2022.

**Questions:**

Summary of rating:

I am most concerned with the first bullet point in Weaknesses where more experiments are expected to support the claims. I would be happy to raise my rating if it is well addressed

**Limitations:**

Yes

---

> ### Author Rebuttal · Authors · 2023-08-09
>
> Thank you for your review!
>
> ## Baselines and the deficiencies of prior work
> We discuss this aspect in detail in the global rebuttal comment above.
>
> In short: We did not mean to say that existing certification procedures had some weakness that needed to be resolved in order to obtain stronger guarantees.
> What we meant to express is that **prior work does not adequately interpret the robustness guarantees  computed by their proposed certification procedures**.
>
> Prior work uses methods like convex relaxation to prove that an equivariant model's prediction is (almost) constant within a certified region $B = \\{x' \mid d_\mathrm{in}(x,x') \leq \epsilon\\}$. One of our main arguments in Section 5 is that, under our sound notion of robustness,  this implies that the model is robust (but not constant) within  region $\hat{B} = \\{x' \mid \hat{d}_\mathrm{in}(x,x') \leq \epsilon\\} \supseteq B$. Here, $\hat{d}\_\mathrm{in}$ is the action-induced distance from Section 4. Depending on $d\_\mathrm{in}$ and the equivariance group, the region $\hat{B}$ may be significantly larger than the region $B$ reported in prior work (see visual example in global comment above). Importantly, this does not depend on a specific certification procedure or baseline.
>
> **Baselines:**
> Our paper is not primarily about which certification method we use, but about what their computed  guarantees actually mean.
> However:
> * For GNNs, we evaluate every existing certificate for insertion/removal of edges/attributes. This corresponds to the $(c^+=1 = c^-)$ lines in Fig. 5, Fig. 6 and Appendix A.3. We further develop novel methods for non-uniform cost.
> * There is no prior work for models with continuous equivariances like DimeNet++. This is why we had to develop equivariance-preserving randomized smoothing in Section 5.1.
> * Similarly, there is no method outside randomized smoothing that can prove robustness for DGCNN under $\ell_2$ norm threats.
>
> During the rebuttal period, we became aware of "3DVerifier" [1], which is specifically designed for the PointNet architecture and $\ell_2$ perturbations. They claim a higher average certified $\ell_2$ radius of $0.959$ (see Table 2 in [1]). This is actually great for us. Combined with our novel theoretical insights (Proposition 2), we can use this method to provide even stronger guarantees against correspondence distance attacks.
>
> Again, please note that our focus is not on advocating for a specific certification method, but on providing semantically meaningful guarantees. We propose equivariance-preserving smoothing as a general-purpose method for cases where no specialized ones are available.
>
> We will update Sections 5.0 and 5.1 of the camera-ready to further clarify this for future readers and use 3DVerifier to showcase stronger guarantees for PointNet.
>
> ## Transformation-specific robustness for (non-)resolvable transformations and group equivariance.
> We discuss this in detail in the global comment.
> In short:
> 1.) If the transformations have a group structure, and the task is invariant w.r.t. this group, then transformation-specific robustness is equivalent to ($G$, $d_\mathrm{in}$, $d_\mathrm{out}, \epsilon, \delta)$ robustness with adversarial budget $\epsilon=0$ -- or a relaxation thereof.
> 2.) Otherwise, it can be thought of as guaranteeing classic adversarial robustness w.r.t. a distance function $d_\mathrm{in}(x,x')$ that determines whether $x'$ is a transformed version of $x$.
>
> **Concerning (non)-resolvable transformations**
> Resolvability is similar to the closedness of groups. A family of functions is resolvable if composing two functions yields another function from the family. A group is closed if combining two group elements yields a new element from the group. The group operator is basically the same as the functions' composition operator. However, groups impose additional constraints on what transformations are to be considered (invertibility, existence of an identity element).
>
> Overall, our paper does indeed deal with resolvable transformations with a group structure (e.g. rotation). Unlike work on transformation-specific robustness, we consider adversaries that are not limited to such transformations, but can perform arbitrary perturbations within some distance $\epsilon$. In addition, we consider the entire group, whereas transformation-specific robustness usually considers some small range of parameters (e.g. rotation angles). We further account for how the model's predictions should change under specific perturbations (equivariance), whereas transformation-specific robustness assumes them to remain constant (invariance).
>
> We will include this and all papers you mentioned in Section 2 of the camera-ready.
>
> ## Justification for smoothing schemes and measures
> Having to choose smoothing schemes and measures is inherent to randomized smoothing. In prior work, the scheme is dictated by the co-domain and the output distance. The measure is dictated by the domain and input distance (see Fig. 1 of [2]).
> In equivariance-preserving smoothing we additionally need to consider their equivariances, which we specify in Section 5.1.
>
> We created two tables that show which scheme and measure is to be used when (see global rebuttal pdf above). These should facilitate the practical application of equivariance-preserving randomized smoothing and will be included in the camera ready.
>
> ## Notation in Section 3
> We agree that our explanation of a randomized smoothing was a little too terse. We will use the higher page limit in the camera ready to provide an explanation of the involved symbols before explaining randomized smoothing.
>
> ---
>
> We hope that we addressed all your comments to your satisfaction.
> Please let us know if you have any further questions during the discussion period.
>
> ---
>
> [1] Mu et al. "3DVerifier: Efficient Robustness Verification for 3D Point Cloud Models". Machine Learning 2022 (Springer)
> [2] Yang et al. "Randomized smoothing of all shapes and sizes". ICML 2020

---

> > ### Comment · Reviewer_nMMh · 2023-08-11
> >
> > Thanks for your reply. Although the experiment is not comprehensive with enough baselines, I think the paper is significant in problem formulation and theoretical framework. Therefore, I raise my rating to borderline accept.

---

### Official Review · Reviewer_4EgB · 2023-07-06

**Soundness:** 4 excellent
**Presentation:** 3 good
**Contribution:** 4 excellent
**Rating:** 7
**Confidence:** 4

**Summary:**

The paper investigates adversarial robustness for group equivariant tasks. To this end, the authors propose a novel notion of adversarial robustness:
An attacker aims to find a perturbation $x'$ "close" (up to some distance constraint in the input space) to a natural input $x$. For this perturbation, they aim to find the worst-case perturbation within the input constraint and over the group of symmetries over the input. By a model equivariant to the underlying group, this reduces to classical adversarial robustness.

Further, the paper shows that randomized smoothing, a popular framework for making models provably adversarially robust, (and many of its variants) can fulfill this notion of robustness when the underlying model, distribution, measure, and smoothing scheme are equivariant.
Several instantiations of this approach are showcased on a variety of tasks, including graphs, point clouds, and molecules.


**Strengths:**

- Well written and easy to read
- The proposed definition for adversarial robustness is a) novel, b) a natural extension of the classical definition, and c) well motivated
- The section on the different variants of randomized smoothing and their relation to the problem is extensive
- I appreciate the combination of semantic preserving change (here the group action) and perceptual noise in the set of admissible perturbations and think this is a promising direction for the field.

**Weaknesses:**

- No investigation of the robustness notion directly, i.e., no attacks against undefended models where performed to showcase the issue
- Some further disambiguation from related work might be helpful to the reader (see below)


**Questions:**

- It seems that the described notion of adversarial robustness requires knowing the inherent equivalences in the data. However, for sufficiently complex data many of the underlying symmetries are not known. Could the notion of robustness be applied in such cases?
- In the conclusion the authors state equivariant robustness guarantees for non-equivariant models as a future direction. It seems that this is the approach already taken in [53,50,77 (Appendix B.5)]. Thus, while these target a fundamentally different setting, could you comment on the difference and relation to these works, particularly when viewed through the novel notion of robustness?


**Limitations:**

Limitations are adequately discussed in the paper.

---

> ### Author Rebuttal · Authors · 2023-08-09
>
> Thank you for your review!
>
> Please note that we cannot update the manuscript during the rebuttal period. We will however make several changes for the camera ready version, some of which are already integrated into the one-page pdf attached to the global comment above.
>
> ## Adversarial attacks
> As you mention, for "a model equivariant to the underlying group, this [the novel notion of robustness] reduces to classical adversarial robustness", which is one of our key theoretical results. Please note that this result applies to both provable robustness and adversarial attacks. Attacks on equivariant models for graphs and point clouds under the classic notion of robustness has already been extensively studied [1, 2].
>
> In fact, **even when the model's equivariances do not match those of the task, any classic adversarial attack is a valid adversarial attack under our notion of robustness**.
>
> Recall the optimization problem from Eq. 1:
> $\max_{x' \in X} \max_{g \in G} d_\mathrm{out}(f(x), g^{-1} \circ f(g \circ x'))$ s.t. $d_\mathrm{in}(x, x') \leq \epsilon$.
> A classic adversarial attack is equivalent to constraining group element $g$ to be the identity element $e$. Thus, any classic adversarial attacks is a valid adversarial attack chosen from some more constrained search space.
>
> Nevertheless, conducting some adversarial attacks of our own is a great opportunity to showcase that our novel notion of robustness is not just some theoretical construct, but a practical property that can be empirically studied. You can find our results for adversarial attacks against PointNet (using a single-step gradient attack) and GCN (using the method from Section 4.4 of [3]) in the pdf attached to the global comment above.
>
> Based on your comment we will
> * explicitly discuss the relation between classic adversarial attacks and our notion of robustness
> * and include the aforementioned experiments (including a full experimental setup) in the camera-ready version of our submission.
>
> ## Unknown equivariances
> We would first like to point out that assuming certain equivariances to be known is not a specific limitation of our work. In fact, the entire field of geometric machine learning is centered around building models implementing specific known equivariances.
>
> How unknown equivariances fit into our framework is however a very interesting question.
>
> Firstly: The notion of **robustness does not require knowledge of the underlying equivariances** (as long as they have a group structure). It is just a property of the model and task that is either fulfilled or not fulfilled. If one had a guarantee that the model's and the task's equivariances match, one could even derive robustness guarantees using classic methods. However: Without specifying the "$G$" part of  $(G, d_\mathrm{in}, d_\mathrm{out}, \epsilon, \delta)$-equivariance-robustness, it would not really be possible to interpret what these robustness guarantees specifically mean.  Making robustness guarantees for equivariant tasks meaningful and interpretable is, in a sense, the exact goal of our paper.
>
> Secondly: One could even generalize our notion of robustness to **unknown equivariances without a group structure**. We could simply define $\max_{x' \in X} \max_{\theta \in \Theta} d_\mathrm{out}(f(x), \psi_{Y,\theta}^{-1} (f( \psi_{X,\theta}(x') )))$, where $\psi_{X,\cdot}$ and $\psi_{Y,\cdot}$ are (unknown) parametric functions in the input and output space.
> The same argument as above would apply. But again: Without knowledge of the equivariances, the robustness properties would not really be interpretable.
>
> Based on your comment, we will mention that
> * our definition does not assume the equivariances to be known,
> * but knowing the equivariances is necessary to provide interpretable robustness gurantees  .
>
> in Section 6 of the camera-ready version.
>
>
> ## Transformation-specific robustness viewed through our notion of robustness
> These methods can indeed be seen as a first step towards provable robustness for non-equivariant models applied to equivariant tasks (for the special case of group invariance). Thank you for pointing out this connection.
>
> We discuss this relation in detail in the global rebuttal comment above. In short:
> 1.) If the transformations have a group structure, and the task is known to be invariant w.r.t. this group, then transformation-specific robustness is equivalent to ($G$, $d_\mathrm{in}$, $d_\mathrm{out}, \epsilon, \delta)$ robustness with adversarial budget $\epsilon=0$ -- or a relaxation thereof.
> 2.) Otherwise, it can be thought of as guaranteeing classic adversarial robustness, i.e. $G = \\{e\\}$,  w.r.t. a distance function $d_\mathrm{in}(x,x')$ that determines if $x'$ is a transformed version of $x$.
>
> There are two key **differences to our notion of robustness**.
> Firstly, we do not constrain the adversary to only applying a specific type of (group-structured) transformation. Instead, they can perform arbitrary perturbations within some distance $\epsilon$ of the original input. Secondly, we do account for how the model's predictions should change under transformations of its inputs (equivariance), whereas transformation-specific robustness always requires the predictions to remain constant (invariance).
>
> Based on your comment, we will include this discussion in the camera ready version of our paper.
>
> ---
>
> We hope that we could answer all your questions to your satisfaction. Please let us know if you have any further questions during the discussion period.
>
> ---
>
>
> [1] Naderi et al. "Adversarial Attacks and Defenses on 3D Point Cloud Classification: A Survey". arxiv 2023
> [2] Jin et al. "Adversarial Attacks and Defenses on Graphs: A Review and Empirical Study". SIGKDD Explorations 2020
> [3] Zügner et al. "Certifiable robustness of graph convolutional networks under structure perturbations". KDD 2019

---

> > ### Comment · Reviewer_4EgB · 2023-08-11
> > **Reply**
> >
> > I thank the authors for their reply. I'm delighted by the suggested discussions to be include.
> > Currently I do not have any further questions.

---

### Official Review · Reviewer_jBnt · 2023-07-07

**Soundness:** 2 fair
**Presentation:** 2 fair
**Contribution:** 2 fair
**Rating:** 5
**Confidence:** 1

**Summary:**

In this paper, it propose  equivariance-perserving randomized smoothing to provide the robustness of the GNN model. The proof demonstrates the provably robustness of the proposed smoothing method.

**Strengths:**

1. The theoretic guarantee of this paper is strong. And the proposed random smoothing methods can be applied on multiple tasks including the point cloud classification and the force field prediction.

**Weaknesses:**

1. The experiment of Figure 4 can contain more geometric GNN such as Spherenet (https://arxiv.org/abs/2102.05013), and GemNet,(https://arxiv.org/abs/2106.08903) to demonstrate.
2. The math part of this paper is a little hard to follow. Maybe some some figures or concrete descriptions on how to apply the proposed smooth techniques can help people understand better.

**Questions:**

What is the model settings of the DimeNet++ in Figure 4? What is the performance compared to reported results?

**Limitations:**

Is there any experiments showing the performance of model when no smoothing techniques are used? I think it can provide the reason why such smoothing technique is needed.

---

> ### Author Rebuttal · Authors · 2023-08-09
>
> Thank you for your helpful review, based on which we will add additional information to the camera ready version, as detailed in our discussion below.
>
> ## Experiments with different architectures
> Following your comment, we performed experiments with additional geometric models for molecule data. Thus far, we added SchNet (one of the best known baselines in the field) and PaiNN (another geometric GNN), as implemented in PyTorch geometric and SchNetPack, respectively.
>
> You can find these results in the rebuttal pdf attached to the global comment above.
> We observe that PaiNN and SchNet offer very similar (but not identical) provable robustness w.r.t.  registration distance attacks.
>
> For the camera ready version we will also include experiments with SphereNet and GemNet.
>
> ## DimeNet++ model settings
> As mentioned in Appendix B.2.2 we use the default parameters specified in the original DimeNet and DimeNet++ papers. The parameters are also included in the file `seml/configs/force_fields/train_dimenet_pp_md17.yaml` provided with the supplementary material.
>
> We use 4 layers with 128 hidden channels. The triplet embedding size is 64. The basis embedding size is 8. The output embedding size is 256. The number of basis functions is 8 (bilinear), 7 (spherical), 6 (radial). The cutoff radius for graph construction is $5 \mathring{A}$. The number of residual layers before the skip connection is $1$. The number of residual layers after the skip connection is $2$.
>
> Based on your comment, we will explicitly include these parameters (and the parameters for any additional models we investigate) in Appendix 2 of our submission.
>
> ## SchNet and PaiNN model settings
> For SchNet, we use 128 hidden channels and 128 filters. We set the number of interaction blocks to 6 and use 50 Gaussians. The cutoff distance is $10 \mathring{A}$, with a maximum number of $32$ neighbors. We use addition as our global readout function.
>
> For PaiNN, we use 128 embedding dimensions in 3 interaction blocks (without shared interaction or filter weights). We use $20$ Gaussian radial basis functions and a cutoff distance of $5 \mathring{A}$.
>
> Again, these parameters (and details on the training procedure) will be added to Appendix 2 of our submission.
>
> ## Model performance with and without smoothing
> As suggested by you, we also performed experiments without randomized smoothing.
> Recall that we had eight sub-datasets "aspirin", "uracil", "ethanol", "benzene", "napthalene", "toluene", "salicylic acid", "malonaldehyde".
>
> The average force MAE of DimeNet++ was:
> * 0.344, 0.182, 0.176, 0.158, 0.095, 0.111, 0.227, 0.304 (without smoothing)
> * 0.369, 0.201, 0.181, 0.192, 0.104, 0.122, 0.238, 0.299 (with smoothing)
>
> The average force MAE of SchNet was:
> * 0.870, 0.322, 0.263, 0.244, 0.250, 0.289, 0.449, 0.445 (without smoothing)
> * 0.931, 0.371, 0.285, 0.260, 0.285, 0.321, 0.496, 0.449  (with smoothing)
>
> The average force MAE of PaiNN was:
> * 0.450, 0.247, 0.295, 0.168, 0.167, 0.188, 0.312, 0.423 (without smoothing)
> * 0.516, 0.285, 0.372, 0.197, 0.192, 0.229, 0.367, 0.493 (with smoothing)
>
> We observe that the accuracy of the model decreases as we apply smoothing noise.
> This is expected and inherent to all randomized smoothing methods. We essentially trade off accuracy for provable robustness. We found $\sigma=1 \mathrm{fm}$ to be a sweet spot where the errors are still very similar to the baseline but we can effectively prove that the adversarial change of the model's predictions is small compared to the test error.
>
> Combined with the evaluation of our randomized smoothing certificates above, we can conclude that DimeNet++ and PaiNN offer a better accuracy-robustness tradeoff than SchNet.
>
> ## Comparison to the reported results in  DimeNet++/SchNet/PaiNN papers
> DimeNet++ is actually not evaluated on MD17 in the original paper. However, the reported MAEs for DimeNet  (which DimeNet++ is a slightly tweaked, more efficient version of), are
> * 0.499, 0.301, 0.230, 0.187, 0.215, 0.216, 0.374, 0.383
>
> We observe that the DimeNet++ MAEs are slightly lower, which is consistent with the claims in the DimeNet++ paper.
>
> The numbers reported for SchNet and PaiNN in the original PaiNN paper are
> * 1.35, 0.56, 0.39, -, 0.58, 0.57, 0.85, 0.66
> * 0.371, 0.140, 0.230, -, 0.083, 0.102, 0.209, 0.319
>
> We observe that our numbers for SchNet are slightly better and our numbers for PaiNN are slightly worse than in the PaiNN paper (which may be expected). However, all numbers are in the expected range of $[0.05, 1.0]$.
>
> ## More detailed description on how to apply randomized smoothing
> We agree that, due to the 9 page limit for the initial submission, the description is currently somewhat terse and directed at an audience already familiar with randomized smoothing.
>
> In essence, all randomized smoothing methods are the same. We just need to choose a smoothing distribution based on the input domain and distance, sample from it and feed the results into our model.
> Then we need to choose a smoothing scheme based on the output domain and distance to compute some simple statistic of the random predictions (e.g. mean or median).
> This may have been somewhat obscured by our measure-theoretic formalization of the procedure.
>
> Based on your comment, we will make the following changes to the ten-page camera ready version:
> * Provide a less dense explanation of randomized smoothing in Background Section 3
> * Provide pseudocode for the overall randomized smoothing procedure
> * Provide a table specifying when to use which smoothing distribution
> * Provide a table specifying when to use which smoothing scheme
> * Provide pseudocode for the certification procedure of the different smoothing schemes (currently, we only reference the original papers for this)
>
> We already included the tables in the rebuttal pdf above.
>
> ---
> We hope that we answered all questions to your satisfaction.
> Please let us know if there is anything else you would like to discuss during the second part of the rebuttal period.

---

### Official Review · Reviewer_t1fA · 2023-07-07

**Soundness:** 4 excellent
**Presentation:** 3 good
**Contribution:** 2 fair
**Rating:** 4
**Confidence:** 3

**Summary:**

Authors note that the definition of adversarial robustness needs to be adjusted in cases when the problem is equivalent - for example, graph perturbations that are large in terms of the number of removed/added edges might actually be small if we take into account graph isomorphism (eg if a large graph perturbation ends up not change the graph topology much). Effectively, authors propose (Proposition 1) to change to notion of the distance between inputs x and x’ in the definition of adversarial robustness to equal the minimal distance between x and all elements of the equivalence class of x’ (eg all permutations of x’ if the input domain is graphs, or all translations and rotations of x’ if the problem is translation and rotation equivariant = use set registration distance between x and x’ to define adversarial robustness of the translation+rotation equivariant task). Then authors argue that symmetries in the output domain should also be accounted for (Def 1) which is often hard, but can be skipped entirely (Proposition 2) if the model itself is equivariant by construction. Then authors argue that the notion of smoothing distribution can (and should) take into account the updated notion of distance between points and that this definition allows that to theoretically justify several existing equivariant smoothing schemes and define more generalized notions of robustness (e.g. Eq 2). Authors note that in many cases computing such distances is computationally hard (e.g. NP-hard when working with permutation equivariance groups). Authors measure their updated equivariance-aware adversarial robustness of PointNet and DGCNN on permutation-invariant problem of ModelNet40 classification, of DimeNet++ on rigid-transformation-equivariant regression on MD17, and on graph convolutional DNN on permutation-equivariant node classification with several notions of graph edit distance (weight of deletion/insertion).

**Strengths:**

The paper is relatively easy to follow and sound. The motivation makes sense. Authors provide a formal generalization of the intuitive notions of how to measure distances between inputs in problems that have natural symmetries and having a shared language might be useful as a reference for future research.

**Weaknesses:**

My main concern with this paper is that it proposes a shared general formalization for the intuitive procedure that people have been already using extensively across many tasks with symmetries (eg rigid-aligned error in pose estimation), but this formalization that does not appear to bring much value beyond defining “shared language” and identifying “generalized knobs” that one can tweak to get slightly varied definitions of adversarial robustness. While reading the paper, I was hoping that authors would provide some kind of surprising practically useful result that makes use of their generalized definitions (eg a general algorithm that lets us compute such distances efficiently), but that did not happen - all presented results appear to be mathematizations of observations we already take for granted (eg if the model is equivariant by construction, there’s no need to invert its outputs because they are already “canonical”). One good example of a paper that worked with symmetries, introduced some simple new general definitions, and proved something surprising and practically useful from that point is “Training Generative Adversarial Networks with Limited Data” by Karras et al (see Appendix C).

I might be wrong, so I would appreciate input from other reviewers.


**Questions:**

See weaknesses.

**Limitations:**

Authors acknowledge that their approach is applicable only to equivariant models.

---

> ### Author Rebuttal · Authors · 2023-08-09
>
> Thank you for your review.
>
> In the following, we would like to highlight the ways in which our results are both surprising and practically useful. We would further like to illustrate this by constrasting them with those of Karras et al. Then, we would like to call attention to the fact that the results of Section 4 are not primarily "a shared language", but contributions in themselves. Finally, we would like to emphasize two key technical and useful contributions.
>
> ## Surprising practically useful results
> We would strongly argue that "It is possible to prove the robustness of neural networks to perturbations bounded by semantically meaningful, extensively studied and computationally hard (even NP-hard) distance functions" is in itself a very surprising result.
>
> To go into more detail: Section 4 leaves us with a novel notion of robustness that requires optimization over a set bounded by potentially NP-hard functions. As such, developing a general algorithm that lets us compute such distances efficiently is not really realistic and would have implications beyond machine learning.
>
> There are two paths one could take:
> * The first one would be the classic certification approach of relaxing the problem. This is essentially the approach taken by recent work on robustness w.r.t. graph optimal transport. For instance, [27] essentially replaces the discrete permutation matrices corresponding to symmetric group $S_N$ with a continuous matrix defining a soft correspondence between nodes in two graphs. The resulting guarantees would be pessimistic bounds on the actual robustness.
> * The second one is the path we take: We identify that, nowadays, equivariant tasks are solved with equivariant models. This allows us to completely eliminate the hard, group-theoretic aspects without any  pessimistic relaxation.
>
> Ignoring the surprise factor: It cannot be denied that this is a practically useful result that resolves an inherent limitation of recent work, which invested  significant  effort into a suboptimal solution to the practically relevant problem of in-/equivariance aware robustness certification.
>
> ## Concerning Karras et al.
> Thank you for pointing out the interesting connection to the work of Karras et al., which we will also include in the related work section of the paper. This offers a great opportunity to highlight the practical usefulness of our novel "equivariance-preserving randomized smoothing" framework.
>
> The results from Appendix C of Karrass et al. show for different transformations $T$ and all  distributions:
> $\mu = \mu' \iff T \mu = T \mu'$.
> A discriminator can thus learn (properties of) the original data distribution. This allows the generator to learn the original distribution while benefitting from a less overfitting discriminator enabled by data augmentation.
>
> The results from Appendix D of our work show for different transformations $T_g$ (corresponding to the push-forward of group actions) and  families of smoothing distributions $(\mu_x)\_{x \in X}$:
> $\mu_{g \bullet x} = \mu_{g \bullet x'} \implies T_g \mu_x = T_g \mu_{x'}$.
> An in-/equivariant base model thus recovers the original smoothing distributions. This allows the smoothed model to appropriately model the equivariant ground truth while benefitting from  provable robustness enabled by randomized smoothing and Proposition 2.
>
> Because robustness is a practically useful property, both works make a strikingly similar "amount" of practically useful contribution.
>
> ## "Shared language"
> Our goal in Section 4 is not (primarily) to define shared language so that members of the trustworthy and geometric ML communities can collaborate.
>
> We first identify deficiencies of directly transplanting common notions of robustness from the image domain onto equivariant domains (Section 1). We then develop a sound notion of adversarial robustness from the perspective of semantics-aware robustness, i.e. defining robustness with knowledge about the ground-truth (Section 4).
>
> We do not deny that similar equivariance errors appear in other fields (see "special cases" in Section 4.3). But the contribution is justifying, in a principled manner, why this notion of robustness should be be worked on by trustworthy machine learning researchers.
>
> ## Observations we already take for granted
> We understand that certain intermediate steps may be less exciting for some readers with a background in geometric machine learning (which are not our primary audience, see above).
>
> However, we believe that our findings are crucial for the trustworthy ML community, which thus far fails to appropriately address key aspects of extending their methods to geometric machine learning tasks and models.
>
> ## Additional technical contributions
> Finally, we  want to highlight two key technical and practical contributions.
>
> Even though Proposition 2 eliminates some of the hard group-theoretic aspects, we still need to prove that our equivariant model is robust under the classic notion of robustness. Prior to this work, there simply existed no specialized method to do so for many practically relevant models (e.g. those with continuous equivariances).
> We identify that randomized smoothing is a promising candidate, but may invalidate Proposition 2.
> We thus develop the novel framework of equivariance-preserving smoothing to enable model-agnostic certification under our notion of robustness.
>
> We further derive multiple non-probabilistic certification procedures for GNNs under perturbations with non-uniform cost. This was not possible before and offers new insights into the robustness properties of this important class of equivariant models (see Section 5.2 and Appendix E).
>
> We will update the beginning of Section 5.1 to better emphasize these contributions.
>
> ---
>
> We hope that we could convince you that our submission is a meaningful contribution to the field of trustworthy machine learning.
>
> Please let us know if you have any additional questions during the discussion period.

---

### Official Review · Reviewer_y2Vs · 2023-07-08

**Soundness:** 3 good
**Presentation:** 2 fair
**Contribution:** 3 good
**Rating:** 6
**Confidence:** 2

**Summary:**

This paper presents several contributions to the field of adversarial robustness for group equivariant tasks. The authors propose a sound concept of adversarial robustness for these tasks and demonstrate that using equivariant models can facilitate achieving provable robustness. They also prove that various randomized smoothing approaches preserve equivariance and extend existing robustness certificates for graph and node classification from $l_0$ perturbations to graph edit distance perturbations with user-specified costs. The paper underscores the importance of reevaluating adversarial robustness in equivariant tasks for future research in robust and geometric machine learning. Extensive experiments were conducted in various settings to validate the claims of the proposed method.

**Strengths:**

- The paper presents a novel definition of adversarial robustness for group equivariant tasks, which is a significant contribution to the field. The $\epsilon-\delta$ definition is interesting.
- The paper provides theoretical proofs for the soundness of their proposed notion.
- The proposed method is applicable to a wide range of tasks, including graphs, point clouds, molecules, and more.
- The paper includes experimental results that validate the theoretical claims, providing empirical evidence for the effectiveness of the proposed method.

**Weaknesses:**

I think this paper is innovative and sound in general, and just want to list a few things to address below:
- While the proposed definition of robustness in this paper specifically tailors toward group equivariance tasks, I am wondering how it can address the limitations and problems with the previously used definition. In figure 1 and figure 2, the authors give two examples related to graph isomorphism and rotation – how would the flaws in the old notion of robustness cause issues in applications related to those scenarios? For instance, one potential setting of graph node classification attacks is making imperceptible insertions/deletions to the recommendation graphs (example attacks in [1] and [2]). How will this new notion of robustness affect the consideration of attacks when grounded to those specific tasks/applications?

[1] Daniel Zügner, Amir Akbarnejad, and Stephan Günnemann. Adversarial attacks on neural networks for graph data. In Proceedings of the 24th ACM SIGKDD International Conference on Knowledge Discovery & Data Mining, pages 2847–2856, 2018.
[2] Daniel Zügner and Stephan Günnemann. Adversarial attacks on graph neural networks via meta learning. arXiv preprint arXiv:1902.08412, 2019.

**Questions:**

Although the paper mainly focuses on certified robustness, will this new notion of adversarial robustness for group equivariant tasks have any implications for the design of new adversarial attacks & defenses? If so, can you discuss some potential directions for proposing new attacks & defenses with respect to this definition?

**Limitations:**

The authors adequately addressed the limitations.

---

> ### Author Rebuttal · Authors · 2023-08-09
>
> Thank you for your review. We are glad to hear that you find our paper sound and innovative.
>
> We are not quite sure if we interpreted your first question correctly, so please let us know if you would like any additional clarification during the second part of the rebuttal period.
>
> ## Flaws of classic robustness exemplified in Figures 1 and 2.
> The goal of provably robust machine learning is to provide a robustness certificate, i.e. a guarantee that a model's prediction cannot be adversarially attacked (similar to how a high test accuracy can be thought of as certifying good generalization to unseen data -- but provably).
>
> The shown flaws would cause this certificate to either oversell (Figure 1) or undersell (Figure 2) the actual robustness of the model.
>
> In Fig. 1, a method operating under the classic notion of robustness might prove that a prediction does not change with up to $\epsilon=7$ edge perturbations and thus claim it to be very robust. However, this claim could even be made about a model whose prediction is changed by something as simple as representing exactly the same graph via an equivalent adjacency matrix. It thus has little practical value.
>
> In Fig. 2, a method operating under the classic notion of robustness would deem a model non-robust if it rotates its prediction as the input image rotates. However, this equivariance (learned or enforced) is exactly what is needed to solve the task with high accuracy. Thus, existing certificates would force model developers to either prioritize accuracy or provable robustness -- even though they are not actually add odds with each other.
>
> Our proposed notion of robustness resolves these flaws by accounting for equivariance, so that meaningful robustness certificates can be provided without causing wrong incentives for model developers.
>
> ## How the novel notion affects the considerations of attacks (particularly on graphs)
> We can distinguish two cases.
>
> When **the model equivariances and the task equivariances match**, then the model is equivariant-robust if and only if it is classically robust. We can thus reuse any existing method to disprove robustness, i.e. perform an adversarial attack. In particular, this applies to graph neural networks, where one can continue using Nettack or meta learning.
>
> Furthermore, existing attacks are even valid adversarial attacks when **the model and task equivariance do not match**. Recall the optimization problem from Eq. 1:
> $\max_{x' \in X} \max_{g \in G} d_\mathrm{out}(f(x), g^{-1} \circ f(g \circ x'))$ s.t. $d_\mathrm{in}(x, x') \leq \epsilon$.
> A classic adversarial attack is equivalent to constraining group element $g$ to be the identity element $e$. Thus, any classic adversarial attacks is a valid (but not necessarily optimal) adversarial attack chosen from some more constrained search space.
>
> Based on the feedback of Reviewer 5, we have also conducted some adversarial attacks of our own (see global rebuttal pdf).
>
> ## Implications / Potential directions for attacks and defenses
> A direct implication of the discussion above is that one can continue using existing methods -- at least as baseline. There are however interesting directions for developing improved methods that are specifically tailored towards equivariant settings.
>
> **Concerning attacks:**
> One interesting direction is developing adversarial attacks for scenarios where the model and task equivariances do not match (e.g. vision transformers). As discussed above, existing attacks are in such cases only a pessimistic bound on the model's actualy vulnerability. The challenge here is finding good ways of  optimizing both the adversarial noise and the group action -- a first attempt could be made via alternating optimization.
>
> Another interesting direction is using our knowledge about equivariances to reduce the search space for adversarial noise. Finding an attack essentially requires optimization over $\hat{B} = \\{g \bullet x \mid g \in G, x \in B\\}$, where $B$ is a classic perturbation set $B = \\{x' \mid d_\mathrm{in}(x,x') \leq \epsilon\\}$, such as a  $\ell_p$ ball. There may be a smaller set $S \subset B$ which gets mapped to the same $\hat{B}$ by group $G$. Consider for example Fig. 3d in the pdf attached to the global comment above. Instead of by a ball, the same set could also be generated by a line that is diagonally translated. Thus, we could find $x'$ via optimization over a one-dimensional space.
>
> **Concerning defenses:**
> A natural direction is using newly developed attacks for adversarial training. Stronger and more efficient attacks allow the trained model to be more robust without overly impacting the computational cost of the training procedure.
>
> Another interesting direction direction is actively using a model's equivariances as a defense mechanism. A first example of this is actually discussed in [1]. There, the inputs of a permutation invariant model are randomly shuffled to disrupt gradient-based attacks without changing the model's prediction.
>
> ---
>
> Based on your comment and those of other reviewers, we will add a section on attacks and defenses to the camera ready version of our paper (after Section 5). We also conducted some initial experiments (see Reviewer 5).
>
> We hope that we could address all your questions to your satisfaction.
> Please let us know if you have any additional comments or questions during the second part of the rebuttal period.
>
> ---
>
> [1] Zhang et al. "The Art of Defense: Letting Networks Fool the Attacker". IEEE Transactions on Information Forensics and Security.

---

> > ### Comment · Reviewer_y2Vs · 2023-08-20
> > **Response to Authors**
> >
> > Dear Authors,
> >
> > Thank you for the comprehensive rebuttal and the clarifications provided. At present, I have no further questions and am willing to raise my rating to weak accept.
> >
> > Best regards,
> > Reviewer y2Vs

---

### Official Review · Reviewer_kw3s · 2023-07-25

**Soundness:** 4 excellent
**Presentation:** 4 excellent
**Contribution:** 3 good
**Rating:** 7
**Confidence:** 4

**Summary:**

This paper provides a new insight into adversarial robustness for equivariant tasks (like graph classification) where we need to separate the perturbations that occur according the allowable transformations (that actually form a group) - and where we'd like to have the same output disregarding the norm of perturbation - and harmful perturbations, where we need to care about its norm.

**Strengths:**

The novel work stating the very important question: what is the objective for the adversarial robustness for inputs which are equivariant according to some group of transformations?
Although the task seems very vague, the authors answered it in a very elegant manner by:
- introducing the new distance $\hat{d}_{in}$ which should be invariant w.r.t. to the group transformations
- introducing two properties of this new distance (upper bound as usual distance $d_{in}$ + being the max of such distances)
- and finally proving the Proposition 1 that shows that $\hat{d}_{in}$ is actually min among usual distances w.r.t. group transformations for one of its input.
Based on it, the authors came to a definition of adversarial robustness for group equivariant tasks, which can be shrunk to the usual definition of adversarial robustness when we don't have the group properties, or compatible with other group-invariant distances like Hausdorff and Chamfer ones.
Moreover, they proved the following (Proposition 2): if a model is equivariant w.r.t. to the group transformations, then the new definition it is equivalent to being robust in the usual sense. As a result, we just need to prove the usual robustness (for which we have a number of techniques like Certified Smoothing) for an equivariant model.

**Weaknesses:**

During the discussion in "Related Work / Transformation-specific robustness" (line 88), and even more seriously during "Product measures" (lines 287-288), it makes sense to mention the work [1] where the interesting usecase was studied:
- multiplicative group of the "gamma correction" image transformations
- Rayleigh distribution to serve as a smoothing distribution
Would be interesting to know how it fits into the proposed framework.

Additionally, how the proposed framework is dealing with interpolation error (e.g., [1], [2]) which is very non-trivial and is out of group properties, is not mentioned in the paper.


[1] Muravev, Nikita, and Aleksandr Petiushko. "Certified robustness via randomized smoothing over multiplicative parameters of input transformations." Proceedings of the Thirty-First International Joint Conference on Artificial Intelligence, 3366-3372, 2022.
[2] Marc Fischer, Maximilian Baader, and Martin Vechev. Certified defense to image transformations via randomized smoothing. Advances in Neural information processing systems, 33: 8404–8417, 2020.

**Questions:**

The main question is why do the authors provide multiple experimentation results if according to their theoretic results we can just re-use any existing methods provided for usual adversarial robustness? We just don't need to implement anything.

**Limitations:**

According to the text, there is a statement: "prior work drastically underestimates the strength of the adversary and actually proves robustness for significantly larger sets of perturbed inputs" (e.g., lines 398-400).
Is there any :
- empirical proof for it?
- theoretical consideration reasons for it?

Probably the answer was somehow blurred inside the text, would love to hear the clear explanation for it.

---

> ### Author Rebuttal · Authors · 2023-08-09
>
> Thank you for the thorough review of our submission and pointing out the interesting connection to smoothing over multiplicative transformation parameters.
>
> ## How randomized smoothing over multiplicative parameters fits into the proposed framework
> We provide an in-depth discussion of transformation-specific robustness through the lens of our novel notion of robustness in the global rebuttal comment above.
>
> **In short:**
> 1.) If transformations have a group structure, and the task is known to be invariant w.r.t. this group, then transformation-specific robustness is equivalent to ($G$, $d_\mathrm{in}$, $d_\mathrm{out}, \epsilon, \delta)$ robustness with adversarial budget $\epsilon=0$ -- or a relaxation thereof.
> 2.) Otherwise, it can be thought of as guaranteeing classic adversarial robustness, i.e. $G = \\{e\\}$,  w.r.t. a distance function $d_\mathrm{in}(x,x')$ that determines if $x'$ is a transformed version of $x$.
>
> **Concerning randomized smoothing of transformation parameters**:
> An interesting connection here is that transformation-specific smoothing can actually be an instance of equivariance-preserving smoothing. The smoothed model may simply inherit equivariances from the transformation: If $\forall g \in G, \forall \theta, \forall x : \psi_\theta(g \bullet x) = g \bullet \psi_\theta(x)$, then the smoothed model $\xi(\psi_\beta(x))$ with random parameters $\beta$ will be equivariant w.r.t. group $G$ -- assuming an appropriate smoothing scheme $\xi$ (e.g. center smoothing).
>
> Concerning **multiplicative transformations:**
> The Gamma transformation is a multiplicative group. In fact, any multiplicatively composable transformation inherits its group structure from the multiplicative group $(R_+,\cdot)$ if we define the group operator as $\psi_\alpha \bullet \psi_\beta = \psi_{\alpha \cdot \beta}$.
> Thus, if our task is invariant w.r.t. to group $(\psi_\theta)$, we are in case 1.) above.
>
> If our task is not specifically invariant w.r.t. $(\psi_\theta)$, we are in case 2.). As discussed above, we can use the method from [1] as a form of equivariance-preserving smoothing.
> Multiplicative transformations actually have many practically interesting equivariances. For example, scaling of point clouds is multiplicatively composable and also rotation and permutation equivariant. So we can use the results in [1] together with our work to prove the robustness of models for rotation-invariant point cloud classification to adversarial scaling.
>
> Concerning **Rayleigh smoothing:**
> Thus far, we have not been able to discern whether specifically using Rayleigh parameter noise offers us any additional equivariances that are not achievable with other parameter distributions. But we will try to look further into it.
>
> Based on your feedback, we will expand our discussion of transformation-specific smoothing and specifically [1] in the camera-ready version.
>
> ## How interpolation errors fit into the framework
> Interpolation errors are primarily an issue with digital images, because they are the result of rasterizing the continuous image signal and quantizing the color values.
> Using the terminology from [1], this means that certain transformations are non-composable. Using the terminology from our paper, this means that the transformations are not a group. Since we are focusing on group equivariance, such transformations are not covered by our framework.
>
> Please note that this is an inherent problem of the image domain and not specifically a limitation of our work. This is also why equivariant models for images are typically derived for continuous image signals (see, e.g., [2, 3]).
> Domains where equivariant models are actually used in practice (graphs, point clouds, molecules ...) do not suffer from this problem. For example, PointNet is perfectly permutation invariant, DimeNet++ is perfectly rotation equivariant and GCNs are perfectly ismorphism equivariant.
>
> Based on your comment, we will include this discussion in Section 6 of the camera ready.
>
> ## Why we provide experimental results
> You are right. Our rigorous derivations show that existing certification methods for equivariant models can be reused for the proposed notion of robustness.
> The reason we conducted new experiments is that they offer novel insights:
> 1.) Firstly, there is no prior work on proving the robustness of models with continuous equivariances. Our equivariance-preserving randomized smoothing approach can thus be seen as a baseline for future work.
> 2.) Secondly, we derive new graph guarantees with non-uniform cost for insertion and deletion (see Section 5.2). Their experimental evaluation provides more detailed insights into the robustness of graph neural network, which were impossible to obtain with existing certification methods.
>
> ## How prior work underestimates the strength of the adversary
> We discuss this point in the global rebuttal comment. In short: What we meant to say is that prior work uses their method to claim  robustness within a set $B = \\{x' \mid d_\mathrm{in}(x,x')\leq \epsilon\\}$. Our work shows that, under a  sound notion of robustness, the same models are actually robust (but not constant) in $\hat{B} =  \\{x' \mid \hat{d}\_\mathrm{in}(x,x')\leq \epsilon\\} \supseteq B$ with action-induced distance $\hat{d}_\mathrm{in}$. This certified region may be significantly larger, depending on the distance function and group (see Fig. 3 in the pdf attached to the global comment).
>
> Based on your comment, we will make this more explicit in the camera-ready version of our paper.
>
> ---
>
> Please let us know if you have any additional questions.
>
> ---
>
> [1] Nikita Murarev and Aleksandr Petiushko. “Certified robustness via randomized smoothing over multiplicative parameters of input transformations.” IJCAI 2022.
> [2] Maurice Weiler and Gabriele Cesa. "General E(2) - Equivariant Steerable CNNs". NeurIPS 2019.
> [3] Gabriele Cesa, Leon Lang, and Maurice Weiler "A Program to Build E(N)-Equivariant Steerable CNNs". ICLR 2022.

---

> > ### Comment · Reviewer_kw3s · 2023-08-13
> > **Thanks!**
> >
> > I thank the authors for their reply. I'm grateful for the suggested discussions to be included. Currently I do not have any further questions.
> > My main concern still is that the group structure is nice in theory but in practice it is usually broken by multiple real considerations (like interpolation), so would be nice somehow to make "almost" group structure research in the future :)

---

### Author Rebuttal · Authors · 2023-08-09

While we have already personally responded to each of the reviewers insightful comments and questions, there are two points that appeared multiple times.

We would like to use this comment to discuss them in detail. We also attached a pdf with additional figures and tables. We further list prospective changes for the camera ready version at the end of this comment.

The comments asked for further clarification concerning:
* How transformation-specific robustness fits into our proposed notion of robustness.
* In what way prior work underestimates the strength of the adversary.
---
## Transformation-specific robustness
Consider a parametric function $\psi_\theta : X \rightarrow X$ with parameter $\theta \in \Theta$.
A model is robust to transformation-specific attacks if $\max_\mathrm{\theta \in \Theta} d_\mathrm{out}(f(x), f(\psi_\theta(x))) \leq \delta$ for some small $\delta$. We can distinguish two cases:

**Transformations with a group structure:**
The first case is that the set of functions $\\{\psi_\theta \mid \theta \in \Theta\\}$ forms a group $G$, with function composition $\circ$ being the group operator and group action $\bullet$ being the application of the function (e.g. rotation).

In this case, the above problem can be reformulated as
$\max_\mathrm{g \in G} d_\mathrm{out}(f(x), f(g \bullet x))$.
This optimization problem is equivalent to $\max_{x' \in X} \max_\mathrm{g \in G} d_\mathrm{out}(f(x), f(g \bullet x'))$ under the constraint $x' = x$.
We can  reformulate this constraint using any formal distance function $d_\mathrm{in}(x,x')$ as $d_\mathrm{in}(x,x') \leq 0$.
We can thus express transformation-specific robustness as
$(\max_{x' \in X} \max_\mathrm{g \in G} d_\mathrm{out}(f(x), f(g \bullet x'))$ s.t. $d_\mathrm{in}(x,x') \leq \epsilon) \leq \delta$ with adversarial budget $\epsilon = 0$.

We observe that this is a special of our notion of robustness, where the adversary has a budget of 0 (i.e. can only apply group actions) and the task is group invariant.
However, in practice, methods for proving transformation-specific robustness are only applied to some subset of $G$ (e.g. a small range of rotation angles). Their guarantees can thus be thought of as a relaxation of this special case.

**Transformations without a group structure:**
If the parametric function $\psi_\theta$ does not have a group structure, we can define any distance function $d_\mathrm{in}(x,x')$ such that $\\{x' \mid d_\mathrm{in}(x,x') \leq \epsilon\\} =
\\{\psi_\theta(x) \mid \theta \in \Theta\\}$. That is, perturbed inputs are closer than $\epsilon$ iff they are the result of applying the function to $x$ with some choice of parameter.

With this choice of distance function, transformation-specific robustness is equivalent to
$(\max_{x' \in X}  d_\mathrm{out}(f(x), f(g \bullet x')$ s.t. $d_\mathrm{in}(x,x') \leq \epsilon) \leq \delta$.
This is an instance of classic adversarial robustness, which is a special case of our notion of robustness (no equivariance).

## How existing works underestimate the strength of the adversary
One of our key results (Proposition 2) is that classic robustness certificates for equivariant models are in fact valid for our novel notion of robustness.
What we meant to say is that, by definition, being robust under our notion of robustness means being robust within a potentially much larger region.

**Importantly, there are no technical deficiencies with the  existing robustness certification methods that exist for specific equivariant model architectures. The problem is with how their provided robustness guarantees have been interpreted thus far.**

Prior work guarantees that a model’s prediction is (almost) constant within certified region $B = \\{x' \mid d_\mathrm{in}(x,x') \leq \epsilon \\}$.
As discussed in Section 4, this means that the model is actually robust (but not necessarily constant) within certified region $\hat{B} = \\{x' \mid \hat{d}_\mathrm{in}(x,x') \leq \epsilon\\}$, where $\hat{d}\_\mathrm{in}$ is the action-induced distance (e.g. the graph edit distance).

All we meant to say is that region $\hat{B}$ may be significantly larger than original region $B$, depending on original input distance $d_\mathrm{in}$, budget $\epsilon$ and equivariance group $G$.

**Visual example (see Fig. 3 in attached pdf):**
Assume clean input $x = (\sqrt{2} \ \sqrt{2})^T \in R^2$. Futher assume that $d_\mathrm{in}$ is the $l_2$ distance and $\epsilon=1$, i.e. we have a certified circle of radius $1$ around $x$.

If we assume the model and task to be equivariant w.r.t. to rotation group $\mathit{SO}(2)$, then the model is equivariant-robust within an annulus ("donut") around the origin.
It's area is $\pi \cdot (3^2 - 1^2)$, which is significantly larger than the original $\pi \cdot 1^2$

If we additionally assume translation equivariance, then the certified region even has an infinite area,  unlike the original small circle. Furthermore, it contains an $l_2$ ball centered at $x$ with a radius of $3$ (compared to the original $1$).

---

## Changelog
Based on the reviewer's valuable feedback, we have decided to make the following changes to the camera ready version of our paper (note that this year's review process does not allow changes during the rebuttal):
* Expand discussion of transformation-specific smoothing in Section 2 and Appendix (see above)
* Discuss image interpolation errors in Section 6
* Further clarify the  "existing methods underestimate the strength of the adversary" comment (see above)
* Add discussion of attacks/defenses (Reviewer 2/5)
* Discuss GAN training with data augmentation in Section 2 (Reviewer 3)
* Add experiments on SchNet / PaiNN / GemNet / SphereNet (see Reviewer 4 and rebuttal pdf)
* Provide more detailed instructions on applying randomized smoothing, including tables (see Review 6 and rebuttal pdf)
* Add experiments on adversarial attacks (Reviewer 5 and rebuttal pdf)
* Mention that we assume equivariance to be known (Reviewer 5)

---

### Decision · Program_Chairs · 2023-09-21

**Decision:**

Accept (poster)

**Comment:**

The authors study adversarial robustness for group equivariant tasks by proposing a novel notion of adversarial robustness. This problem is important and well motivated. The reviewers highly appreciated the novelty and the elegance of the method proposed. All the reviewers found the work relevant, interesting, with meaningful theoretical and empirical results. Therefore I do recommend acceptance of this paper.